# Invariance Principle Meets Information Bottleneck for Out-of-Distribution Generalization

**Kartik Ahuja**[†]      **Ethan Caballero**[*][†]      **Dinghuai Zhang**[*][†]

**Jean-Christophe Gagnon-Audet** [†]      **Yoshua Bengio** [†]      **Ioannis Mitliagkas**[†]

**Irina Rish**[†]

## Abstract

The invariance principle from causality is at the heart of notable approaches such as invariant risk minimization (IRM) that seek to address out-of-distribution (OOD) generalization failures. Despite the promising theory, invariance principle-based approaches fail in common classification tasks, where invariant (causal) features capture all the information about the label. Are these failures due to the methods failing to capture the invariance? Or is the invariance principle itself insufficient? To answer these questions, we revisit the fundamental assumptions in linear regression tasks, where invariance-based approaches were shown to provably generalize OOD. In contrast to the linear regression tasks, we show that for linear classification tasks we need much stronger restrictions on the distribution shifts, or otherwise OOD generalization is impossible. Furthermore, even with appropriate restrictions on distribution shifts in place, we show that the invariance principle alone is insufficient. We prove that a form of the *information bottleneck* constraint along with invariance helps address key failures when invariant features capture all the information about the label and also retains the existing success when they do not. We propose an approach that incorporates both of these principles and demonstrate its effectiveness in several experiments.

## 1 Introduction

Recent years have witnessed an explosion of examples showing deep learning models are prone to exploiting shortcuts (spurious features) (Geirhos et al., 2020; Pezeshki et al., 2020) which make them fail to generalize out-of-distribution (OOD). In Beery et al. (2018), a convolutional neural network was trained to classify camels from cows; however, it was found that the model relied on the background color (e.g., green pastures for cows) and not on the properties of the animals (e.g., shape). These examples become very concerning when they occur in real-life applications (e.g., COVID-19 detection (DeGrave et al., 2020)).

To address these out-of-distribution generalization failures, invariant risk minimization (Arjovsky et al., 2019) and several other works were proposed (Ahuja et al., 2020; Pezeshki et al., 2020; Krueger et al., 2020; Robey et al., 2021; Zhang et al., 2021). The invariance principle from causality (Peters et al., 2015; Pearl, 1995) is at the heart of these works. The principle distinguishes predictors that only rely on the causes of the label from those that do not. The optimal predictor that only focuses on the causes is invariant and min-max optimal (Rojas-Carulla et al., 2018; Koyama and Yamaguchi, 2020; Ahuja et al., 2021) under many distribution shifts but the same is not true for other predictors.

---

[*]Equal contribution.

[†]Mila - Quebec AI Institute, Université de Montréal. Correspondence to: kartik.ahuja@mila.quebec.

35th Conference on Neural Information Processing Systems (NeurIPS 2021).

**Our contributions.** Despite the promising theory, invariance principle-based approaches fail in settings (Aubin et al., 2021) where invariant features capture all information about the label contained in the input. A particular example is image classification (e.g., cow vs. camel) (Beery et al., 2018) where the label is a deterministic function of the invariant features (e.g., shape of the animal), and does not depend on the spurious features (e.g., background). To understand such failures, we revisit the fundamental assumptions in linear regression tasks, where invariance-based approaches were shown to provably generalize OOD. We show that, in contrast to the linear regression tasks, OOD generalization is significantly harder for linear classification tasks; we need much stronger restrictions in the form of support overlap assumptions[3] on the distribution shifts, or otherwise it is not possible to guarantee OOD generalization under interventions on variables other than the target class. We then proceed to show that, even under the right assumptions on distribution shifts, the invariance principle is insufficient. However, we establish that *information bottleneck* (IB) constraints (Tishby et al., 2000), together with the invariance principle, provably works in both settings – when invariant features completely capture the information about the label and also when they do not. (Table 1 summarizes our theoretical results presented later). We propose an approach that combines both these principles and demonstrate its effectiveness on linear unit tests (Aubin et al., 2021) and on different real datasets.

| Task | Invariant features capture label info | Support overlap invariant features | Support overlap spurious features | **OOD generalization guarantee** ($\mathcal{E}_{tr} \to \mathcal{E}_{all}$) | | | |
|------|------|------|------|------|------|------|------|
| | | | | ERM | IRM | IB-ERM | IB-IRM |
| Linear Classification | Full/Partial | No | Yes/No | \multicolumn Impossible for any algorithm to generalize OOD [Thm2] | | | |
| | Full | Yes | No | ✗ | ✗ | ✓ | ✓ [Thm3,4] |
| | Partial | Yes | No | ✗ | ✗ | ✗ | ✓ [Appendix] |
| | Full | Yes | Yes | ✓ | ✓ | ✓ | ✓ [Thm3,4] |
| | Partial | Yes | Yes | ✗ | ✓ | ✗ | ✓ |
| Linear Regression | Full | No | No | ✓ | ✓ | ✓ | ✓ |
| | Partial | No | No | ✗ | ✓ | ✗ | ✓ [Thm4] |

Table 1: Summary of the new and existing results (Arjovsky et al., 2019; Rosenfeld et al., 2021). IB-ERM (IRM): information bottleneck - empirical (invariant) risk minimization ERM (IRM).

## 2 OOD generalization and invariance: background & failures

**Background.** We consider a supervised training data $D$ gathered from a set of training environments $\mathcal{E}_{tr}$: $D = \{D^e\}_{e \in \mathcal{E}_{tr}}$, where $D^e = \{x_i^e, y_i^e\}_{i=1}^{n^e}$ is the dataset from environment $e \in \mathcal{E}_{tr}$ and $n^e$ is the number of instances in environment $e$. $x_i^e \in \mathbb{R}^d$ and $y_i^e \in \mathcal{Y} \subseteq \mathbb{R}^k$ correspond to the input feature value and the label for $i^{th}$ instance respectively. Each $(x_i^e, y_i^e)$ is an i.i.d. draw from $\mathbb{P}^e$, where $\mathbb{P}^e$ is the joint distribution of the input feature and the label in environment $e$. Let $\mathcal{X}^e$ be the support of the input feature values in the environment $e$. The goal of OOD generalization is to use training data $D$ to construct a predictor $f : \mathbb{R}^d \to \mathbb{R}^k$ that performs well across many unseen environments in $\mathcal{E}_{all}$, where $\mathcal{E}_{all} \supset \mathcal{E}_{tr}$. Define the risk of $f$ in environment $e$ as $R^e(f) = \mathbb{E}\big[\ell(f(X^e), Y^e)\big]$, where for example $\ell$ can be 0-1 loss, logistic loss, square loss, $(X^e, Y^e) \sim \mathbb{P}^e$, and the expectation $\mathbb{E}$ is w.r.t. $\mathbb{P}^e$. Formally stated, our goal is to use the data from training environments $\mathcal{E}_{tr}$ to find $f : \mathbb{R}^d \to \mathcal{Y}$ to minimize

$$\min_f \max_{e \in \mathcal{E}_{all}} R^e(f). \tag{1}$$

So far we did not state any restrictions on $\mathcal{E}_{all}$. Consider binary classification: without any restrictions on $\mathcal{E}_{all}$, no method can reduce the above objective ($\ell$ is 0-1 loss) to below one. Suppose a method outputs $f^*$; if $\exists\, e \in \mathcal{E}_{all} \setminus \mathcal{E}_{tr}$ with labels based on $1 - f^*$, then it achieves an error of one. Some assumptions on $\mathcal{E}_{all}$ are thus necessary. Consider how $\mathcal{E}_{all}$ is restricted using invariance for linear regressions (Arjovsky et al., 2019).

**Assumption 1.** *Linear regression structural equation model (SEM). In each $e \in \mathcal{E}_{all}$*

$$\begin{aligned} Y^e &\leftarrow w_{\mathsf{inv}}^* \cdot Z_{\mathsf{inv}}^e + \epsilon^e, \quad Z_{\mathsf{inv}}^e \perp \epsilon^e, \quad \mathbb{E}[\epsilon^e] = 0, \mathbb{E}\big[|\epsilon^e|^2\big] \leq \sigma_{\mathsf{sup}}^2 \\ X^e &\leftarrow S(Z_{\mathsf{inv}}^e, Z_{\mathsf{spu}}^e) \end{aligned} \tag{2}$$

*where $w_{\mathsf{inv}}^* \in \mathbb{R}^m$, $Z_{\mathsf{inv}}^e \in \mathbb{R}^m$, $Z_{\mathsf{spu}} \in \mathbb{R}^o$, $S \in \mathbb{R}^{d \times (m+o)}$, $S$ is invertible ($m + o = d$). We focus on invertible $S$ but several results extend to non-invertible $S$ as well (see Appendix).*

---

[3]Support is the region where the probability density for continuous random variables (probability mass function for discrete random variables) is positive. Support overlap refers to the setting where train and test distribution maybe different but share the same support. We formally define this later in Assumption 5.

Assumption 1 states how $Y^e$ and $X^e$ are generated from latent invariant features $Z_{\text{inv}}^e$ [4], latent spurious features $Z_{\text{spu}}^e$ and noise $\epsilon^e$. The *relationship between label and invariant features is invariant, i.e.,* $w_{\text{inv}}^*$ *is fixed* across all environments. However, the distributions of $Z_{\text{inv}}^e$, $Z_{\text{spu}}^e$, and $\epsilon^e$ are allowed to change arbitrarily across all the environments. Suppose $S$ is identity. If we regress only on the invariant features $Z_{\text{inv}}^e$, then the optimal solution is $w_{\text{inv}}^*$, which is independent of the environment, and the error it achieves is bounded above by the variance of $\epsilon^e$ ($\sigma_{\text{sup}}^2$). If we regress on the entire $Z^e$ and the optimal predictor places a non-zero weight on $Z_{\text{spu}}^e$ (e.g., $Z_{\text{spu}}^e \leftarrow Y^e + \zeta^e$), then this predictor fails to solve equation (1) ($\exists\, e \in \mathcal{E}_{all}$, $Z_{\text{spu}}^e \to \infty$, error $\to \infty$, see Appendix for details). Also, not only regressing on $Z_{\text{inv}}^e$ is better than on $Z^e$, it can be shown that it is optimal, i.e., it solves equation (1) under Assumption 1 and achieves a value of $\sigma_{\text{sup}}^2$ for the objective in equation (1).

**Invariant predictor.** Define a linear representation map $\Phi : \mathbb{R}^{r \times d}$ (that transforms $X^e$ as $\Phi(X^e)$) and define a linear classifier $w : \mathbb{R}^{k \times r}$ (that operates on the representation $w \cdot \Phi(X^e)$). We want to search for representations $\Phi$ such that $\mathbb{E}[Y^e | \Phi(X^e)]$ is invariant (in Assumption 1 if $\Phi(X^e) = Z_{\text{inv}}^e$, then $\mathbb{E}[Y^e | \Phi(X^e)]$ is invariant). We say that a data representation $\Phi$ elicits an invariant predictor $w \cdot \Phi$ across the set of training environments $\mathcal{E}_{tr}$ if there is a predictor $w$ that simultaneously achieves the minimum risk, i.e., $w \in \arg\min_{\tilde{w}} R^e(\tilde{w} \cdot \Phi)$, $\forall e \in \mathcal{E}_{tr}$. The main objective of IRM is stated as

$$\min_{w \in \mathbb{R}^{k \times r}, \Phi \in \mathbb{R}^{r \times d}} \frac{1}{|\mathcal{E}_{tr}|} \sum_{e \in \mathcal{E}_{tr}} R^e(w \cdot \Phi) \quad \text{s.t. } w \in \arg\min_{\tilde{w} \in \mathbb{R}^{k \times r}} R^e(\tilde{w} \cdot \Phi), \ \forall e \in \mathcal{E}_{tr}. \tag{3}$$

Observe that if we drop the constraints in the above which search only over invariant predictors, then we get the standard empirical risk minimization (ERM) (Vapnik, 1992) (assuming all the training environments occur with equal probability). In all our theorems, we use 0-1 loss for binary classification $\mathcal{Y} = \{0, 1\}$ and square loss for regression $\mathcal{Y} = \mathbb{R}$. For binary classification, the output of the predictor is given as $\mathsf{I}(w \cdot \Phi(X^e))$, where $\mathsf{I}(\cdot)$ is the indicator function that takes 1 if the input is $\geq 0$ and 0 otherwise, and the risk is $R^e(w \cdot \Phi) = \mathbb{E}\big[|\mathsf{I}(w \cdot \Phi(X^e)) - Y^e|\big]$. For regression, the output of the predictor is $w \cdot \Phi(X^e)$ and the corresponding risk is $R^e(w \cdot \Phi) = \mathbb{E}\big[(w \cdot \Phi(X^e) - Y^e)^2\big]$. We now present the main OOD generalization result from Arjovsky et al. (2019) for linear regressions.

**Theorem 1.** *(Informal) If Assumption 1 is satisfied,* $\mathsf{Rank}[\Phi] > 0$, $|\mathcal{E}_{tr}| > 2d$, *and* $\mathcal{E}_{tr}$ *lie in a linear general position (a mild condition on the data in* $\mathcal{E}_{tr}$, *defined in the Appendix), then each solution to equation* (3) *achieves OOD generalization (solves equation (1),* $\nexists\, e \in \mathcal{E}_{all}$ *with risk* $> \sigma_{\text{sup}}^2$*).*

Despite the above guarantees, IRM has been shown to fail in several cases including linear SEMs in (Aubin et al., 2021). We take a closer look at these failures next.

**Understanding the failures: fully informative invariant features vs. partially informative invariant features (FIIF vs. PIIF).** We define properties salient to the datasets/SEMs used in the OOD generalization literature. Each $e \in \mathcal{E}_{all}$, the distribution $(X^e, Y^e) \sim \mathbb{P}^e$ satisfies the following properties. a) $\exists$ a map $\Phi^*$ (linear or not), which we call an *invariant feature map*, such that $\mathbb{E}\big[Y^e | \Phi^*(X^e)\big]$ is the same for all $e \in \mathcal{E}_{all}$ and $Y^e \not\perp \Phi^*(X^e)$. These conditions ensure $\Phi^*$ maps to features that have a finite predictive power and have the same optimal predictor across $\mathcal{E}_{all}$. For the SEM in Assumption 1, $\Phi^*$ maps to $Z_{\text{inv}}^e$. b) $\exists$ a map $\Psi^*$ (linear or not), which we call *spurious feature map*, such that $\mathbb{E}\big[Y^e | \Psi^*(X^e)\big]$ is not the same for all $e \in \mathcal{E}_{all}$ and $Y^e \not\perp \Psi^*(X^e)$ for some environments. $\Psi^*$ often creates a hindrance in learning predictors that only rely on $\Phi^*$. Note that $\Psi^*$ should not be a transformation of some $\Phi^*$. For the SEM in Assumption 1, suppose $Z_{\text{spu}}^e$ is anti-causally related to $Y^e$, then $\Psi^*$ maps to $Z_{\text{spu}}^e$ (See Appendix for an example).

In the colored MNIST (CMNIST) dataset (Arjovsky et al., 2019), the digits are colored in such a way that in the training domain, color is highly predictive of the digit label but this correlation being spurious breaks down at test time. Suppose the invariant feature map $\Phi^*$ extracts the uncolored digit and the spurious feature map $\Psi^*$ extracts the background color. Ahuja et al. (2021) studied two variations of the colored MNIST dataset, which differed in the way final labels are generated from original MNIST labels (corrupted with noise or not). They showed that the IRM exhibits good OOD generalization (50% improvement over ERM) in anti-causal-CMNIST (AC-CMNIST, original data from Arjovsky et al. (2019)) but is no different from ERM and fails in covariate shift-CMNIST (CS-CMNIST). In AC-CMNIST, the invariant features $\Phi^*(X^e)$ (uncolored digit) are *partially informative* about the label, i.e., $Y \not\perp X^e | \Phi^*(X^e)$, and color contains information about label not contained

---

[4]In many examples in the literature, invariant features are causal, but not always (Rosenfeld et al., 2021).

| Fully informative invariant features (FIIF) | Partially informative invariant features (PIIF) |
|---|---|
| $\forall e \in \mathcal{E}_{all}, Y^e \perp X^e \| \Phi^*(X^e)$ | $\exists\, e \in \mathcal{E}_{all}\ Y^e \not\perp X^e \| \Phi^*(X^e)$ |
| **Task: classification** | **Task: classification or regression** |
| Example 2/2S, CS-CMNIST | Example 1/1S, Example 3/3S, AC-CMNIST |
| SEM in Assumption 2 | SEM in Rosenfeld et al. (2021) |
| **ERM and IRM fail** | **ERM fails, IRM succeeds sometimes** |
| Theorem 3,4 (This paper) | Theorem 9, 5.1 (Arjovsky et al., 2019; Rosenfeld et al., 2021) |

Table 2: Categorization of OOD evaluation datasets and SEMs. Example 1/1S, 2/2S, 3/3S from (Aubin et al., 2021), AC-CMNIST(Arjovsky et al., 2019), CS-CMNIST(Ahuja et al., 2021).

in the uncolored digit. On the other hand in CS-CMNIST, invariant features are *fully informative* about the label, i.e., $Y \perp X^e \| \Phi^*(X^e)$, i.e., they contains all the information about the label that is contained in input $X^e$. Most human labelled datasets have fully informative invariant features; the labels (digit value) only depend on the invariant features (uncolored digit) and spurious features (color of the digit) do not affect the label. [5] In the rare case, when the humans are asked to label images in which the object being labelled itself is blurred, humans can rely on spurious features such as the background making such a data representative of PIIF setting. In Table 2, we divide the different datasets used in the literature based on informativeness of the invariant features. We observe that when the invariant features are fully informative, both IRM and ERM fail but only in classification tasks and not in regression tasks (Ahuja et al., 2021); this is consistent with the linear regression result in Theorem 1, where IRM succeeds regardless of whether $Y^e \perp X^e \| Z_{\text{inv}}^e$ holds or not. Motivated by this observation, we take a closer look at the classification tasks where invariant features are fully informative.

## 3 OOD generalization theory for linear classification tasks

**A two-dimensional example with fully informative invariant features.** We start with a 2D classification example (based on Nagarajan et al. (2021)), which can be understood as a simplified version of the CS-CMNIST dataset (Ahuja et al., 2021), Example 2/2S of Aubin et al. (2021), where both IRM and ERM fail. The example goes as follows. In each training environment $e \in \mathcal{E}_{tr}$

$$Y^e \leftarrow \mathsf{I}\Big(X_{\text{inv}}^e - \frac{1}{2}\Big),\ \text{where } X_{\text{inv}}^e \in \{0,1\} \text{ is Bernoulli}\Big(\frac{1}{2}\Big),$$

$$X_{\text{spu}}^e \leftarrow X_{\text{inv}}^e \oplus W^e,\ \text{where } W^e \in \{0,1\} \text{ is Bernoulli}\big(1-p^e\big) \text{ with selection bias } p^e > \frac{1}{2}, \tag{4}$$

where Bernoulli$(a)$ takes value 1 with probability $a$ and 0 otherwise. Each training environment is characterized by the probability $p^e$. Following Assumption 1, we assume that the labelling function does not change from $\mathcal{E}_{tr}$ to $\mathcal{E}_{all}$, thus the relation between the label and the invariant features does not change. Assume that the distribution of $X_{\text{inv}}^e$ and $X_{\text{spu}}^e$ can change arbitrarily. See Figure 1a) for a pictorial representation of this example illustrating the gist of the problem: there are many classifiers with the same error on $\mathcal{E}_{tr}$ while only the one identical to the labelling function $\mathsf{I}(X_{\text{inv}}^e - \frac{1}{2})$ generalizes correctly OOD. Define a classifier $\mathsf{I}(w_{\text{inv}}x_{\text{inv}} + w_{\text{spu}}x_{\text{spu}} - \frac{1}{2}(w_{\text{inv}} + w_{\text{spu}}))$. Define a set of classifiers $\mathcal{S} = \{(w_{\text{inv}}, w_{\text{spu}}) \text{ s.t. } w_{\text{inv}} > |w_{\text{spu}}|\}$. Observe that all the classifiers in $\mathcal{S}$ achieve a zero classification error on the training environments. However, only classifiers for which $w_{\text{spu}} = 0$ solve the OOD generalization (eq. (1)). With $\Phi$ as the identity, it can be shown that all the classifiers $\mathcal{S}$ form an invariant predictor (satisfy the constraint in equation (3) over all the training environments when $\ell$ is the 0-1 loss). Observe that increasing the number of training environments to infinity does not address the problem, unlike with the linear regression result discussed in Theorem 1 (Arjovsky et al., 2019), where it was shown that if the number of environments increases linearly in the dimension of the data, then the solution to IRM also solves the OOD generalization (eq. (1)). [6] We use the above example to construct general SEMs for linear classification when the invariant features are fully informative. We follow the structure of the SEM from Assumption 1 in our construction.

---

[5]The deterministic labelling case was referred as realizable problems in (Arjovsky et al., 2019).

[6]Please note that this example illustrates certain important facets in a very simple fashion; only in this example a max-margin classifier can solve the problem but not in general. (Further explanation in the Appendix).

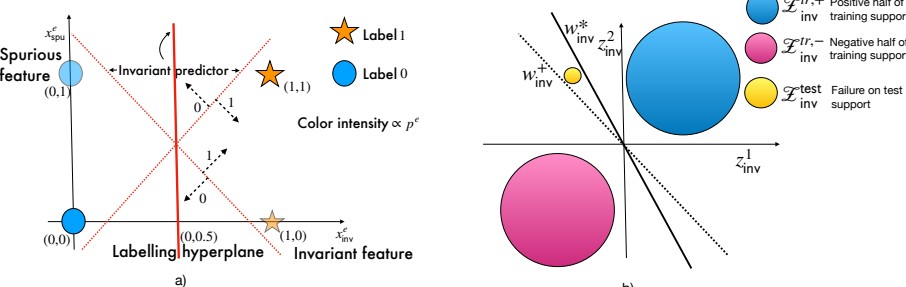

a)
b)

Figure 1: a) 2D classification example illustrating multiple invariant predictors: Most of these predictors rely on spurious features and each of them achieve zero error across all $\mathcal{E}_{tr}$, b) illustration of the impossibility result. If latent invariant features in the training environments are separable, then there are multiple equally good candidates that could have generated the data, and the algorithm cannot distinguish between these.

**Assumption 2.** *Linear classification structural equation model (FIIF). In each $e \in \mathcal{E}_{all}$*

$$Y^e \leftarrow I\big(w_{\mathsf{inv}}^* \cdot Z_{\mathsf{inv}}^e\big) \oplus N^e, \qquad N^e \sim \mathsf{Bernoulli}(q), q < \frac{1}{2}, \qquad N^e \perp (Z_{\mathsf{inv}}^e, Z_{\mathsf{spu}}^e), \tag{5}$$

$$X^e \leftarrow S\big(Z_{\mathsf{inv}}^e, Z_{\mathsf{spu}}^e\big),$$

*where $w_{\mathsf{inv}}^* \in \mathbb{R}^m$ with $\|w_{\mathsf{inv}}^*\| = 1$ is the labelling hyperplane, $Z_{\mathsf{inv}}^e \in \mathbb{R}^m$, $Z_{\mathsf{spu}}^e \in \mathbb{R}^o$, $N^e$ is binary noise with identical distribution across environments, $\oplus$ is the XOR operator, $S$ is invertible.*

If noise level $q$ is zero, then the above SEM covers linearly separable problems. See Figure 2a) for the directed acyclic graph (DAG) corresponding to this SEM. From the DAG observe that $Y^e \perp X^e | Z_{\mathsf{inv}}^e$, which implies that the invariant features are fully informative. Contrast this with a DAG that follows Assumption 1 shown in Figure 2b), where $Y^e \not\perp X^e | Z_{\mathsf{inv}}^e$ and thus the invariant features are not fully informative. If $\mathcal{E}_{all}$ follows the SEM in Assumption 2 and suppose the distribution of $Z_{\mathsf{inv}}^e$, $Z_{\mathsf{spu}}^e$ can change arbitrarily, then it can be shown that only a classifier identical to the labelling function $I(w_{\mathsf{inv}}^* \cdot Z_{\mathsf{inv}}^e)$ can solve the OOD generalization (eq. (1)); such a classifier achieves an error of $q$ (noise level) in all the environments. As a result, if for a classifier we can find $e \in \mathcal{E}_{all}$ that follows Assumption 2 where the error is greater than $q$, then such a classifier does not solve equation (1). Now we ask – what are the minimal conditions on training environments $\mathcal{E}_{tr}$ to achieve OOD generalization when $\mathcal{E}_{all}$ follow Assumption 2? To achieve OOD generalization for linear regressions, in Theorem 1, it was required that the number of training environments grows linearly in the dimension of the data. However, there was no restriction on the support of the latent invariant and latent spurious features, and they were allowed to change arbitrarily from train to test (for further discussion on this, see the Appendix). Can we continue to work with similar assumptions for the SEM in Assumption 2 and solve the OOD generalization (eq. (1))? We state some assumptions and notations to answer that. Define the support of the invariant (spurious) features $Z_{\mathsf{inv}}^e$ ($Z_{\mathsf{spu}}^e$) in environment $e$ as $\mathcal{Z}_{\mathsf{inv}}^e$ ($\mathcal{Z}_{\mathsf{spu}}^e$).

**Assumption 3.** *Bounded invariant features.* $\cup_{e \in \mathcal{E}_{tr}} \mathcal{Z}_{\mathsf{inv}}^e$ *is a bounded set.*[7]

**Assumption 4.** *Bounded spurious features.* $\cup_{e \in \mathcal{E}_{tr}} \mathcal{Z}_{\mathsf{spu}}^e$ *is a bounded set.*

**Assumption 5.** *Invariant feature support overlap.* $\forall e \in \mathcal{E}_{all}, \mathcal{Z}_{\mathsf{inv}}^e \subseteq \cup_{e' \in \mathcal{E}_{tr}} \mathcal{Z}_{\mathsf{inv}}^{e'}$

**Assumption 6.** *Spurious feature support overlap.* $\forall e \in \mathcal{E}_{all}, \mathcal{Z}_{\mathsf{spu}}^e \subseteq \cup_{e' \in \mathcal{E}_{tr}} \mathcal{Z}_{\mathsf{spu}}^{e'}$

Assumption 5 (6) states that the support of the invariant (spurious) features for unseen environments is the same as the union of the support over the training environments. It is important to note that support overlap does not imply that the distribution over the invariant features does not change. We now define a margin that measures how much the is training support of invariant features $Z_{\mathsf{inv}}^e$ separated by the labelling hyperplane $w_{\mathsf{inv}}^*$. Define Inv-Margin $= \min_{z \in \cup_{e \in \mathcal{E}_{tr}} \mathcal{Z}_{\mathsf{inv}}^e} \mathsf{sgn}\big(w_{\mathsf{inv}}^* \cdot z\big)\big(w_{\mathsf{inv}}^* \cdot z\big)$. This margin only coincides with the standard margin in support vector machines when the noise level $q$ is 0 (linearly separable) and $S$ is identity. If Inv-Margin $> 0$, then the labelling hyperplane $w_{\mathsf{inv}}^*$ separates the support into two halves (see Figure 1b)).

---

[7]A set $\mathcal{Z}$ is bounded if $\exists M < \infty$ such that $\forall z \in \mathcal{Z}, \|z\| \leq M$.

**Assumption 7.** *Strictly separable invariant features.* Inv-Margin $> 0$.

Next, we show the importance of support overlap for invariant features.

**Theorem 2.** *Impossibility of guaranteed OOD generalization for linear classification. Suppose each $e \in \mathcal{E}_{all}$ follows Assumption 2. If for all the training environments $\mathcal{E}_{tr}$, the latent invariant features are bounded and strictly separable, i.e., Assumption 3 and 7 hold, then every deterministic algorithm fails to solve the OOD generalization (eq. (1)), i.e., for the output of every algorithm $\exists\, e \in \mathcal{E}_{all}$ in which the error exceeds the minimum required value $q$ (noise level).*

The proofs to all the theorems are in the Appendix. We provide a high-level intuiton as to why invariant feature support overlap is crucial to the impossibility result. In Figure 1b), we show that if the support of latent invariant features are strictly separated by the labelling hyperplane $w^*_{\mathsf{inv}}$, then we can find another valid hyperplane $w^+_{\mathsf{inv}}$ that is equally likely to have generated the same data. There is no algorithm that can distinguish between $w^*_{\mathsf{inv}}$ and $w^+_{\mathsf{inv}}$. As a result, if we use data from the region where the hyperplanes disagree (yellow region Figure 1b)), then the algorithm fails.

**Significance of Theorem 2.** We showed that without the support overlap assumption on the invariant features, OOD generalization is impossible for linear classification tasks. This is in contrast to linear regression in Theorem 1 (Arjovsky et al., 2019), where even in the absence of the support overlap assumption, guaranteed OOD generalization was possible. Applying the above Theorem 2 to the 2D case (eq. (4)) implies that we cannot assume that the support of invariant latent features can change, or else that case is also impossible to solve.

Next, we ask what further assumptions are minimally needed to be able to solve the OOD generalization (eq. (1)). Each classifier can be written as $\bar{w} \cdot X^e = \bar{w} \cdot S(Z^e_{\mathsf{inv}}, Z^e_{\mathsf{spu}}) = \tilde{w}_{\mathsf{inv}} \cdot Z^e_{\mathsf{inv}} + \tilde{w}_{\mathsf{spu}} Z^e_{\mathsf{spu}}$. If $\tilde{w}_{\mathsf{spu}} \neq 0$, then the classifier $\bar{w}$ is said to rely on spurious features.

**Theorem 3.** *Sufficiency and Insufficiency of ERM and IRM. Suppose each $e \in \mathcal{E}_{all}$ follows Assumption 2. Assume that a) the invariant features are strictly separable, bounded, and satisfy support overlap, b) the spurious features are bounded (Assumptions 3-5, 7 hold).*

● *Sufficiency: If the spurious features satisfy support overlap (Assumption 6 holds), then both ERM and IRM solve the OOD generalization problem (eq. (1)). Also, there exist solutions to ERM and IRM solutions that rely on the spurious features and still achieve OOD generalization.*

● *Insufficiency: If spurious features do not satisfy support overlap, then both ERM and IRM fail at solving the OOD generalization problem (eq. (1)). Also, there exist no such classifiers that rely on spurious features and also achieve OOD generalization.*

**Significance of Theorem 3.** From the first part, we learn that if the support overlap is satisfied for both the invariant features and the spurious features, then either ERM or IRM can solve the OOD generalization (eq. (1)). Interestingly, in this case we can have classifiers that rely on the spurious features and yet solve the OOD generalization (eq. (1)). For the 2D case (eq. (4)) this case implies that the entire set $\mathcal{S}$ solves the OOD generalization (eq. (1)). From the second part, we learn that if support overlap holds for invariant features but not for spurious features, then the ideal OOD optimal predictors rely only on the invariant features. In this case, methods like ERM and IRM continue to rely on spurious features and fail at OOD generalization. For the above 2D case (eq. (4)) this implies that only the predictors that rely only on $X^e_{\mathsf{inv}}$ in the set $\mathcal{S}$ solve the OOD generalization (eq. (1)).

To summarize, we looked at SEMs for classification tasks when invariant features are fully informative, and find that the support overlap assumption over invariant features is necessary. Even in the presence of support overlap for invariant features, we showed that ERM and IRM can easily fail if the support overlap is violated for spurious features. This raises a natural question – Can we even solve the case with the support overlap assumption only on the invariant features? We will now show that the information bottleneck principle can help tackle these cases.

## 4 Information bottleneck principle meets invariance principle

**Why the information bottleneck?** The information bottleneck principle prescribes to learn a representation that compresses the input $X$ as much as possible while preserving all the relevant information about the target label $Y$ (Tishby et al., 2000). Mutual information $I(X; \Phi(X))$ is used to measure information compression. If representation $\Phi(X)$ is a deterministic transformation of $X$, then in principle we can use the entropy of $\Phi(X)$ to measure compression (Kirsch et al., 2020). Let

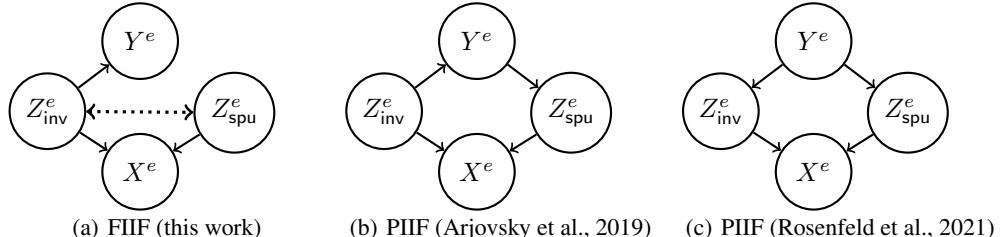

(a) FIIF (this work)    (b) PIIF (Arjovsky et al., 2019)    (c) PIIF (Rosenfeld et al., 2021)

Figure 2: Comparison of the DAG from Assumption 2 (fully informative invariant features) vs. DAGs from Rosenfeld et al. (2021); Arjovsky et al. (2019) (partially informative invariant features).

us revisit the 2D case (eq. (4)) and apply this principle to it. Following the second part of Theorem 3, where ERM and IRM failed, assume that invariant features satisfy the support overlap assumption, but make no such assumption for the spurious features. Consider three choices for $\Phi$: identity (selects both features), selects invariant feature only, selects spurious feature only. The entropy of $H(\Phi(X^e))$ when $\Phi$ is the identity is $H(p^e) + \log(2)$, where $H(p^e)$ is the Shannon entropy in Bernoulli($p^e$). If $\Phi$ selects the invariant/spurious features only, then $H(\Phi(X^e)) = \log(2)$. Among all three choices, the one that has the least entropy and also achieves zero error is the representation that focuses on the invariant feature. We could find the OOD optimal predictor in this example just by using information bottleneck. Does it mean the invariance principle isn't needed? We answer this next.

**Why invariance?** Consider a simple classification SEM. In each $e \in \mathcal{E}_{tr}$, $Y^e \leftarrow X_{\text{inv}}^{1,e} \oplus X_{\text{inv}}^{2,e} \oplus N^e$ and $X_{\text{spu}}^e \leftarrow Y^e \oplus V^e$, where all the random variables involved are binary valued, noise $N^e, V^e$ are Bernoulli with parameters $q$ (identical across $\mathcal{E}_{tr}$), $c^e$ (varies across $\mathcal{E}_{tr}$) respectively. If $c^e < q$, then in $\mathcal{E}_{tr}$ predictions based on $X_{\text{spu}}^e$ are better than predictions based on $X_{\text{inv}}^{1,e}, X_{\text{inv}}^{2,e}$. If both $X_{\text{inv}}^{1,e}, X_{\text{inv}}^{2,e}$ are uniform Bernoulli, then these features have a higher entropy than $X_{\text{spu}}^e$. In this case, the information bottleneck would bar using $X_{\text{inv}}^{1,e}, X_{\text{inv}}^{2,e}$. Instead, we want the model to focus on $X_{\text{inv}}^{1,e}$, $X_{\text{inv}}^{2,e}$ and not on $X_{\text{spu}}^e$. Invariance constraints encourage the model to focus on $X_{\text{inv}}^{1,e}, X_{\text{inv}}^{2,e}$. In this example, observe that invariant features are partially informative unlike the 2D case (eq. (4)).

**Why invariance and information bottleneck?** We have illustrated through simple examples when the information bottleneck is needed but not invariance and vice-versa. We now provide a simple example where both these constraints are needed at the same time. This example combines the 2D case (eq. (4)) and the example we highlighted in the paragraph above: $Y^e \leftarrow X_{\text{inv}}^e \oplus N^e$, $X_{\text{spu}}^{1,e} \leftarrow X_{\text{inv}}^e \oplus W^e$, and $X_{\text{spu}}^{2,e} \leftarrow Y^e \oplus V^e$. In this case, the invariance constraint does not allow representations that use $X_{\text{spu}}^{2,e}$ but does not prohibit representations that rely on $X_{\text{spu}}^{1,e}$. However, information bottleneck constraints on top ensure that representations that only use $X_{\text{inv}}^e$ are used. We now describe an objective [8] that combines both these principles:

$$\min_{w,\Phi} \sum_{e \in \mathcal{E}_{tr}} h^e(w \cdot \Phi) \quad \text{s.t.} \quad \frac{1}{|\mathcal{E}_{tr}|} \sum_{e \in \mathcal{E}_{tr}} R^e(w \cdot \Phi) \leq r^{\text{th}}, \ w \in \arg\min_{\tilde{w} \in \mathbb{R}^{k \times r}} R^e(\tilde{w} \cdot \Phi), \forall e \in \mathcal{E}_{tr}, \quad (6)$$

where $h^e$ in the above is a lower bounded differential entropy defined below and $r^{\text{th}}$ is the threshold on the average risk. Typical information bottleneck based optimization in neural networks involves minimization of the entropy of the representation output from a certain hidden layer. For both analytical convenience and also because the above setup is a linear model, we work with the simplest form of bottleneck which directly minimizes the entropy of the output layer. Recall the definition of differential entropy of a random variable $X$, $h(X) = -\mathbb{E}_X[\log d\mathbb{P}_X]$ and $d\mathbb{P}_X$ is the Radon-Nikodym derivative of $\mathbb{P}_X$ with respect to Lebesgue measure. Because in general differential entropy has no lower bound, we add a small independent noise term $\zeta$ (Kirsch et al., 2020) to the classifier to ensure that the entropy is bounded below. We call the above optimization information bottleneck based invariant risk minimization (IB-IRM). In summary, *among all the highly predictive invariant predictors we pick the ones that have the least entropy*. If we drop the invariance constraint from the above optimization, we get information bottleneck based empirical risk minimization (IB-ERM). In the above formulation and following result, we assume that $X^e$ are continuous random variables; the results continue to hold for discrete $X^e$ as well (See Appendix for details).

**Theorem 4.** *IB-IRM and IB-ERM vs. IRM and ERM*

---

[8]Results extend to alternate objective with information bottleneck constraints and average risk as objective.

• **Fully informative invariant features (FIIF).** *Suppose each $e \in \mathcal{E}_{all}$ follows Assumption 2. Assume that the invariant features are strictly separable, bounded, and satisfy support overlap (Assumptions 3,5 and 7 hold). Also, for each $e \in \mathcal{E}_{tr}$ $Z^e_{spu} \leftarrow AZ^e_{inv} + W^e$, where $A \in \mathbb{R}^{o \times m}$, $W^e \in \mathbb{R}^o$ is continuous, bounded, and zero mean noise. Each solution to IB-IRM (eq. (6), with $\ell$ as 0-1 loss, and $r^{th} = q$), and IB-ERM solves the OOD generalization (eq. (1)) but ERM and IRM (eq.(3)) fail.*

• **Partially informative invariant features (PIIF).** *Suppose each $e \in \mathcal{E}_{all}$ follows Assumption 1 and $\exists~e \in \mathcal{E}_{tr}$ such that $\mathbb{E}[\epsilon^e Z^e_{spu}] \neq 0$. If $|\mathcal{E}_{tr}| > 2d$ and the set $\mathcal{E}_{tr}$ lies in a linear general position (a mild condition defined in the Appendix), then each solution to IB-IRM (eq. (6), with $\ell$ as square loss, $\sigma^2_\epsilon < r^{th} \leq \sigma^2_Y$, where $\sigma^2_Y$ and $\sigma^2_\epsilon$ are the variance in the label and noise across $\mathcal{E}_{tr}$) and IRM (eq.(3)) solves OOD generalization (eq. (1)) but IB-ERM and ERM fail.*

**Significance of Theorem 4 and remarks.** In the first part (FIIF), IB-ERM and IB-IRM succeed without assuming support overlap for the spurious features, which was crucial for success of ERM and IRM in Theorem 3. This establishes that support overlap of spurious features is not a necessary condition. Observe that when invariant features are fully informative, IB-ERM and IB-IRM succeed, but when invariant features are partially informative IB-IRM and IRM succeed. In real data settings, we do not know if the invariant features are fully or partially informative. Since IB-IRM is the only common winner in both the settings, it would be pragmatic to use it in the absence of domain knowledge about the informativeness of the invariant features. In the paragraph preceding the objective in equation (6), we discussed examples where both the IB and IRM constraints were needed at the same time. In the Appendix, we generalize that example and show that if we change the assumptions in linear classification SEM in Assumption 2 such that the invariant features are partially informative, then we see the joint benefit of IB and IRM constraints. At this point, it is also worth pointing to a result in Rosenfeld et al. (2021), which focused on linear classification SEMs (DAG shown in Figure 2c) with partially informative invariant features. Under the assumption of complete support overlap for spurious and invariant features, authors showed IRM succeeds.

### 4.1 Proposed approach

We take the three terms from the optimization in equation (6) and create a weighted combination as

$$\sum_e \Big(R^e(\Phi) + \lambda \|\nabla_{w,w=1.0} R^e(w \cdot \Phi)\|^2 + \nu h^e(\Phi)\Big) \leq \sum_e \Big(R^e(\Phi) + \lambda \|\nabla_{w,w=1.0} R^e(w \cdot \Phi)\|^2 + \nu h(\Phi)\Big).$$

In the LHS above, the first term corresponds to the risks across environments, the second term approximates invariance constraint (follows the IRMv1 objective (Arjovsky et al., 2019)), and the third term is the entropy of the classifier in each environment.

In the RHS, $h(\Phi)$ is the entropy of $\Phi$ unconditional on the environment (the entropy on the left-hand side is entropy conditional on the environment assuming all the environments are equally likely). Optimizing over differential entropy is not easy, and thus we resort to minimizing an upper bound of it (Kirsch et al., 2020). We use the standard result that among all continuous random variables with the same variance, Gaussian has the maximum differential entropy. Since the entropy of Gaussian increases with its variance, we use the variance of $\Phi$ instead of the differential entropy (For further details, see the Appendix). Our final objective is given as

$$\sum_e \Big(R^e(\Phi) + \lambda \|\nabla_{w,w=1.0} R^e(w \cdot \Phi)\|^2 + \gamma \mathsf{Var}(\Phi)\Big). \quad (7)$$

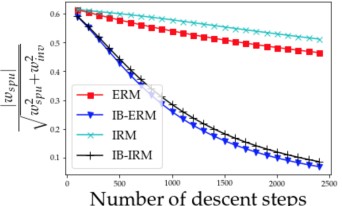

Figure 3: Comparing convergence of $\frac{|w_{spu}|}{\sqrt{w^2_{spu} + w^2_{inv}}}$ (metric from Nagarajan et al. (2021)) for average selection bias $p = 0.9$.

**On the behavior of gradient descent with and without information bottleneck.** In the entire discussion so far, we have focused on ensuring that the set of optimal solutions to the desired objective (IB-IRM, IB-ERM, etc.) correspond to the solutions of the OOD generalization problem (eq. (1)). In some simple cases, such as the 2D case (eq. (4)), it can be shown that gradient descent is biased towards selecting the ideal classifier (Soudry et al., 2018; Nagarajan et al., 2021). Even though gradient descent can eventually learn the ideal classifier that only relies on the invariant features, training is frustratingly slow as was shown by Nagarajan et al. (2021). In the next theorem, we characterize the impact of using IB penalty ($\mathsf{Var}(\Phi)$) in the 2D example (eq. (4)). We compare the methods in terms of $|\frac{w_{spu}(t)}{w_{inv}(t)}|$, which was the metric used in Nagarajan et al. (2021); $w_{spu}(t)$ and $w_{inv}(t)$ are the weights for the spurious feature and the invariant feature at time $t$ of training (assuming training happens with continuous time gradient descent).

**Theorem 5.** *Impact of IB on learning speed.* *Suppose each $e \in \mathcal{E}_{tr}$ follows the 2D case from equation* (4)*. Set $\lambda = 0$, $\gamma > 0$ in equation* (7) *to get the IB-ERM objective with $\ell$ as exponential loss. Continuous-time gradient descent on this IB-ERM objective achieves $|\frac{w_{\mathsf{spu}}(t)}{w_{\mathsf{inv}}(t)}| \leq \epsilon$ in time less than $\frac{W_0(\frac{1}{2\gamma})}{2(1-p)\epsilon}$ ($W_0(\cdot)$ denotes the principal branch of the Lambert W function), while in the same time the ratio for ERM $|\frac{w_{\mathsf{spu}}(t)}{w_{\mathsf{inv}}(t)}| \geq \ln(\frac{1+2p}{3-2p})/\ln\left(1 + \frac{W_0(\frac{1}{2\gamma})}{2(1-p)\epsilon}\right)$, where $p = \frac{1}{|\mathcal{E}_{tr}|} \sum_{e \in \mathcal{E}_{tr}} p^e$ .*

$|\frac{w_{\mathsf{spu}}(t)}{w_{\mathsf{inv}}(t)}|$ converges to zero for both methods, but it converges much faster for IB-ERM (for $p = 0.9, \epsilon = 0.001, \gamma = 0.58$, the ratio for IB-ERM is $|\frac{w_{\mathsf{spu}}(t)}{w_{\mathsf{inv}}(t)}| \leq 0.001$ and ratio for ERM is $|\frac{w_{\mathsf{spu}}(t)}{w_{\mathsf{inv}}(t)}| \geq 0.09$). In the above theorem, we analyzed the impact of information bottleneck only. The convergence analysis for both the penalties jointly comes with its own challenges, and we hope to explore this in future work. However, we carried out experiments with gradient descent on all the objectives for the 2D example (eq. (4)). See Figure 3 for the comparisons.

## 5 Experiments

**Methods, datasets & metrics.** We compare our approaches – information bottleneck based ERM (IB-ERM) and information bottleneck based IRM (IB-IRM) with ERM and IRM. We also compare with an Oracle model trained on data where spurious features are permuted to remove spurious correlations. We use all the datasets in Table 2, Terra Incognita dataset (Beery et al., 2018), and COCO (Ahmed et al., 2021). We follow the same protocol for tuning hyperparameters from Aubin et al. (2021); Arjovsky et al. (2019) for their respective datasets (see the Appendix for more details). As is reported in literature, for Example 2/2S, Example 3/3S we use classification error and for AC-CMNIST, CS-CMNIST, Terra Incognita, and COCO we use accuracy. For Example 1/1S, we use mean square error (MSE). The code for experiments can be found at `https://github.com/ahujak/IB-IRM`.

**Summary of results.** In Table 3, we provide a comparison of methods for different examples in linear unit tests (Aubin et al., 2021) for three and six training environments. In Table 4, we provide a comparison of the methods for different CMNIST datasets, Terra Incognita and COCO dataset. Based on our Theorem 4, we do not expect ERM and IB-ERM to do well on Example 1/1S, Example 3/3S and AC-CMNIST as these datasets fall in the PIIF category, i.e, the invariant features are partially informative. On these examples, we find that IRM and IB-IRM do better than ERM and IB-ERM (for Example 3/3S when there are three environments all methods perform poorly). Based on our Theorem 4, we do not expect IRM and ERM to do well on Example 2/2S, CS-CMNIST, Terra Incognita and COCO dataset,[9] as these datasets fall in the FIIF category, i.e., the invariant features are fully informative. On these FIIF examples, we find that IB-ERM always performs well (close to oracle), and in some cases IB-IRM also performs well. Our experiments confirm that IB penalty has a crucial role to play in FIIF settings and IRMv1 penalty has a crucial role to play in PIIF settings (to further this claim, we provide an ablation study in the Appendix). On Example 1/1S, AC-CMNIST, we find that IB-IRM is able to extract the benefit of IRMv1 penalty. On CS-CMNIST and Example 2/2S we find that IB-IRM is able to extract the benefit of IB penalty. In settings such as COCO dataset, where IB-IRM does not perform as well as IB-ERM, better hyperparameter tuning strategies should be able to help IB-IRM adapt and put a higher weight on IB penalty. Overall, we can conclude that IB-ERM improves over ERM (significantly in FIIF and marginally in PIIF settings), and IB-IRM improves over IRM (improves in FIIF settings and retains advantages in PIIF settings).

**Remark.** As we move from three to six environments, we observe that MSE in Example 1/1S exhibits a larger variance. This is because of the way data is generated, the new environments that are sampled have labels that have a higher noise level (we follow the same procedure as in Aubin et al. (2021)).

## 6 Extensions, limitations, and future work

**Extension to non-linear models and multi-class classification.** In this work our theoretical analysis focused on linear models. Consider the map $X \leftarrow S(Z_{\mathsf{inv}}, Z_{\mathsf{spu}})$ in Assumption 2. Suppose $S$ is non-linear and bijective. We can divide the learning task into two parts a) invert $S$ to obtain $Z_{\mathsf{inv}}, Z_{\mathsf{spu}}$ and b) learn a linear model that only relies on the invariant features $Z_{\mathsf{inv}}$ to predict the label $Y$. For

---

[9]We place Terra Incognita and COCO dataset in the FIIF assuming that the humans who labeled the images did not need to rely on unreliable/spurious features such as background to generate the labels.

|  | #Envs | ERM | IB-ERM | IRM | IB-IRM | Oracle |
|---|---|---|---|---|---|---|
| Example1 | 3 | $13.36 \pm 1.49$ | $12.96 \pm 1.30$ | $11.15 \pm 0.71$ | $11.68 \pm 0.90$ | $10.42 \pm 0.16$ |
| Example1s | 3 | $13.33 \pm 1.49$ | $12.92 \pm 1.30$ | $11.07 \pm 0.68$ | $11.74 \pm 1.03$ | $10.45 \pm 0.19$ |
| Example2 | 3 | $0.42 \pm 0.01$ | $0.00 \pm 0.00$ | $0.45 \pm 0.00$ | $0.00 \pm 0.00$ | $0.00 \pm 0.00$ |
| Example2s | 3 | $0.45 \pm 0.01$ | $0.00 \pm 0.01$ | $0.45 \pm 0.01$ | $0.06 \pm 0.12$ | $0.00 \pm 0.00$ |
| Example3 | 3 | $0.48 \pm 0.07$ | $0.49 \pm 0.06$ | $0.48 \pm 0.07$ | $0.48 \pm 0.07$ | $0.01 \pm 0.00$ |
| Example3s | 3 | $0.49 \pm 0.06$ | $0.49 \pm 0.06$ | $0.49 \pm 0.07$ | $0.49 \pm 0.07$ | $0.01 \pm 0.00$ |
| Example1 | 6 | $33.74 \pm 60.18$ | $32.03 \pm 57.05$ | $23.04 \pm 40.64$ | $25.66 \pm 45.96$ | $22.21 \pm 39.25$ |
| Example1s | 6 | $33.62 \pm 59.80$ | $31.92 \pm 56.70$ | $22.92 \pm 40.60$ | $25.60 \pm 45.62$ | $22.13 \pm 38.93$ |
| Example2 | 6 | $0.37 \pm 0.06$ | $0.02 \pm 0.05$ | $0.46 \pm 0.01$ | $0.43 \pm 0.11$ | $0.00 \pm 0.00$ |
| Example2s | 6 | $0.46 \pm 0.01$ | $0.02 \pm 0.06$ | $0.46 \pm 0.01$ | $0.45 \pm 0.10$ | $0.00 \pm 0.00$ |
| Example3 | 6 | $0.33 \pm 0.18$ | $0.26 \pm 0.20$ | $0.14 \pm 0.18$ | $0.19 \pm 0.19$ | $0.01 \pm 0.00$ |
| Example3s | 6 | $0.36 \pm 0.19$ | $0.27 \pm 0.20$ | $0.14 \pm 0.18$ | $0.19 \pm 0.19$ | $0.01 \pm 0.00$ |

Table 3: Comparisons on linear unit tests in terms of mean square error (regression) and classification error (classification). "#Envs" means the number of training environments.

|  | ERM | IB-ERM | IRM | IB-IRM |
|---|---|---|---|---|
| CS-CMNIST | $60.27 \pm 1.21$ | $71.80 \pm 0.69$ | $61.49 \pm 1.45$ | $71.79 \pm 0.70$ |
| AC-CMNIST | $16.84 \pm 0.82$ | $50.24 \pm 0.47$ | $66.98 \pm 1.65$ | $67.67 \pm 1.78$ |
| Terra Incognita | $49.80 \pm 4.40$ | $56.40 \pm 2.10$ | $54.60 \pm 1.30$ | $54.10 \pm 2.00$ |
| COCO | $22.70 \pm 1.04$ | $31.66 \pm 2.39$ | $18.47 \pm 10.20$ | $25.10 \pm 1.03$ |

Table 4: Classification accuracy percentage on colored MNISTs, Terra Incognita and COCO dataset.

part b), we can rely on the approaches proposed in this work. For part a), we need to leverage advancements in the field of non-linear ICA (Khemakhem et al., 2020). The current state-of-the-art to solve part a) requires strong structural assumptions on the dependence between all the components of $Z_{\mathsf{inv}}, Z_{\mathsf{spu}}$ (Lu et al., 2021). Therefore, solving part a) and part b) in conjunction with minimal assumptions forms an exciting future work. In the entire work, the discussion was focused on binary classification tasks and regression tasks. For multi-class classification settings, we consider natural extension of the SEM in Assumption 2 (See the Appendix) and our main results continue to hold.

**On the choice for IB penalty and IRMv1 penalty.** We use the approximation for entropy (in equation (7)) described in Kirsch et al. (2020). The approximation (even though an upper bound) serves as an effective proxy for the true information bottleneck as shown in the experiments in Kirsch et al. (2020) (e.g., see their experiment on Imagenette dataset). Also, our experiments validate this approximation even in moderately high dimensions, as an example in CS-CMNIST, the dimension of the layer at which bottleneck constraints are applied is 256. Developing tighter approximations for information bottleneck in high dimensions and analyzing their impact on OOD generalization is an important future work. In recent works (Rosenfeld et al., 2021; Kamath et al., 2021; Gulrajani and Lopez-Paz, 2021), there has been criticism of different aspects of IRM, e.g., failure of IRMv1 penalty in non-linear models, the tuning of IRMv1 penalty, etc. Since we use IRMv1 penalty in our proposed loss, these criticisms apply to our objective as well. Other approximations of invariance have been proposed in the literature (Koyama and Yamaguchi, 2020; Ahuja et al., 2020; Chang et al., 2020). Exploring their benefits together with information bottleneck is a fruitful future work. Before concluding, we want to remark that we have already discussed the closest related works. However, we also provide a detailed discussion of the broader related literature in the Appendix.

## 7 Conclusion

In this work, we revisited the fundamental assumptions for OOD generalization for settings when invariant features capture all the information about the label. We showed how linear classification tasks are different and need much stronger assumptions than linear regression tasks. We provide a sharp characterization of performance of ERM and IRM under different assumptions on support overlap of invariant and spurious features. We showed that support overlap of invariant features is necessary or otherwise OOD generalization is impossible. However, ERM and IRM seem to fail even in the absence of support overlap of spurious features. We prove that a form of the information bottleneck constraint along with invariance goes a long way in overcoming the failures while retaining the existing provable guarantees.

## Acknowledgements

We thank Reyhane Askari Hemmat, Adam Ibrahim, Alexia Jolicoeur-Martineau, Divyat Mahajan, Ryan D'Orazio, Nicolas Loizou, Manuela Girotti, and Charles Guille-Escuret for the feedback. Kartik Ahuja would also like to thank Karthikeyan Shanmugam for discussions pertaining to the related works.

## Funding disclosure

We would like to thank Samsung Electronics Co., Ldt. for funding this research. Kartik Ahuja acknowledges the support provided by IVADO postdoctoral fellowship funding program. Yoshua Bengio acknowledges the support from CIFAR and IBM. Ioannis Mitliagkas acknowledges support from an NSERC Discovery grant (RGPIN-2019-06512), a Samsung grant, Canada CIFAR AI chair and MSR collaborative research grant. Irina Rish acknowledges the support from Canada CIFAR AI Chair Program and from the Canada Excellence Research Chairs Program. We thank Compute Canada for providing computational resources.

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
