# A   Appendix

**Organization.** In Section A.1, we discuss the societal impact of this work. In Section A.2, we provide further details on the experiments. In Section A.3, we provide a detailed discussion on structural equation models and the linear general position assumption used to prove Theorem 1. In Section A.4, we first cover the notations used in the proofs, followed by some technical remarks to be kept in mind for all the proofs, and then we provide the proof of the impossibility result in Theorem 2. In Section A.5, we provide the proof for sufficiency and insufficiency characterization of ERM and IRM discussed in Theorem 3. In Section A.6, we provide the proof for Theorem 4, which compares IB-IRM, IB-ERM with IRM and ERM. In Section A.7, we discuss the step-by-step derivation of the final objective in equation (7). In Section A.8, we provide the proof for Theorem 5, which compares the impact of information bottleneck penalty on the learning speed. In Section A.9, we provide an analysis of settings when both IRM and IB penalty work together in conjunction. Also, at the end of each section describing a proof, we provide remarks on various aspects, including some simple extensions that our results already cover. Although in the main manuscript we covered the relevant related works, in Section A.10, we provide a more detailed discussion on other related works.

## A.1   Societal impact

When machine learning models are deployed to assist in making decisions in safety-critical applications (e.g., self-driving cars, healthcare, etc.), we want to ensure that they make decisions that can be trusted well beyond the regime of the training data that they are exposed to. The models used in current practice are prone to exploiting spurious correlations/shortcuts in arriving at decisions and are thus not always reliable. In this work, we took some steps towards building a well-founded theory and proposing methods based on the same that can eventually help us build machines that work well beyond the training data regime. At this point, we do not anticipate a negative impact specifically of this work.

## A.2   Experiments details

In this section, we provide further details on the experiments. The codes to reproduce the experiments is provided at `https://github.com/ahujak/IB-IRM`. We have also added the codes to DomainBed (`https://github.com/facebookresearch/DomainBed`).

### A.2.1   Datasets

We first describe the datasets (Example 1/1S, Example 2/2S, Example 3/3S) introduced in Aubin et al. (2021); these datasets are referred to as the linear unit tests. The results for linear unit tests are presented in Table 3.

**Example 1/1S (PIIF).** This example follows the linear regression SEM from Assumption 1. The dataset in environment $e \in \mathcal{E}_{all}$ is sampled from the following

$$
\begin{aligned}
Z_{\mathsf{inv}}^e &\sim \mathcal{N}_m(0, (\sigma^e)^2), &\quad \tilde{Y}^e &\sim \mathcal{N}_m(W_{yz} Z_{\mathsf{inv}}^e, (\sigma^e)^2), \\
Z_{\mathsf{spu}}^e &\sim \mathcal{N}_o(W_{zy} \tilde{Y}^e, 1), &\quad Z^e &\leftarrow (Z_{\mathsf{inv}}^e, Z_{\mathsf{spu}}^e), \\
Y^e &\leftarrow \frac{2}{(m+o)} \mathbf{1}_m^\mathsf{T} \tilde{Y}^e, &\quad X^e &\leftarrow S(Z^e),
\end{aligned}
$$

where $W_{yz} \in \mathbb{R}^{m \times m}$, $W_{zy} \in \mathbb{R}^{o \times m}$ are matrices drawn i.i.d. from the standard normal distribution, $\mathbf{1}_m \in \mathbb{R}^m$ is a vector of ones, $\mathcal{N}_k$ is a $k$ dimensional vector from the normal distribution. For the first three environments ($e_0, e_1, e_2$), the variances are fixed as $(\sigma^{e_0})^2 = 0.1$, $(\sigma^{e_1})^2 = 1.5$, and $(\sigma^{e_2})^2 = 2.0$. When the number of environments is greater than three, then $(\sigma^{e_j})^2 \sim \mathsf{Uniform}(10^{-2}, 10)$. The scrambling matrix $S$ is set to identity in Example 1 and a random unitary matrix is selected to rotate the latents in Example 1S. In the above dataset, the invariant features are causal and partially informative about the label. The spurious features are anti-causally related to the label and carry extra information about the label not contained in the invariant features.

**Example 2/2S (FIIF).** This example follows the linear classification SEM from Assumption 2 with zero noise. The dataset generalizes the 2D cow versus camel classification task in equation (4). Let

$$\theta_{\text{cow}} = \mathbf{1}_{\text{m}}, \qquad \theta_{\text{camel}} = -\theta_{\text{cow}}, \qquad \nu_{\text{animal}} = 10^{-2},$$
$$\theta_{\text{grass}} = \mathbf{1}_{\text{o}}, \qquad \theta_{\text{sand}} = -\theta_{\text{grass}}, \qquad \nu_{\text{background}} = 1.$$

The dataset in environment $e \in \mathcal{E}_{all}$ is sampled from the following distribution

$$U^e \sim \text{Categorical}\big(p^e s^e, (1-p^e)s^e, p^e(1-s^e), (1-p^e)(1-s^e)\big),$$

$$Z_{\text{inv}}^e \sim \begin{cases} (\mathcal{N}_m(0,0.1) + \theta_{\text{cow}})\nu_{\text{animal}} & \text{if } U^e \in \{1,2\}, \\ (\mathcal{N}_m(0,0.1) + \theta_{\text{camel}})\nu_{\text{animal}} & \text{if } U^e \in \{3,4\}, \end{cases}$$

$$Z_{\text{spu}}^e \sim \begin{cases} (\mathcal{N}_o(0,0.1) + \theta_{\text{grass}})\nu_{\text{background}} & \text{if } U^e \in \{1,4\}, \\ (\mathcal{N}_o(0,0.1) + \theta_{\text{sand}})\nu_{\text{background}} & \text{if } U^e \in \{2,3\}, \end{cases}$$

$$Z^e \leftarrow (Z_{\text{inv}}^e, Z_{\text{spu}}^e), \quad X^e \leftarrow S(Z^e),$$

$$Y^e \leftarrow \mathsf{I}(\mathbf{1}_m^\mathsf{T} Z_{\text{inv}}^e),$$

where for the first three environments the background parameters are $p^{e_0} = 0.95$, $p^{e_1} = 0.97$, $p^{e_2} = 0.99$ and the animal parameters are $s^{e_0} = 0.3$, $s^{e_1} = 0.5$, $s^{e_2} = 0.7$. When the number of environments are greater than three, then $p^{e_j} \sim \text{Uniform}(0.9, 1)$, and $s^{e_j} \sim \text{Uniform}(0.3, 0.7)$. The scrambling matrix $S$ is set to identity in Example 2 and a random unitary matrix is selected to rotate the latents in Example 2S. In the above dataset, the invariant features are causal and carry full information about the label. The spurious features are correlated with the invariant features through a confounding selection bias $U^e$.

**Example 3/3S (PIIF).** This example is a classification problem following the SEM assumed in (Rosenfeld et al., 2021b). The example is meant to present a linear version of the spiral classification problem in (Parascandolo et al., 2021). Let $\theta_{\text{inv}} = 0.1 \cdot \mathbf{1}_m$, and $\theta_{\text{spu}}^e \sim \mathcal{N}_o(0,1)$ for all the environments. The dataset in environment $e \in \mathcal{E}_{all}$ is sampled from the following distribution

$$Y^e \sim \text{Bernoulli}\left(\frac{1}{2}\right),$$

$$Z_{\text{inv}}^e \sim \begin{cases} \mathcal{N}_m(+\theta_{\text{inv}}, 0.1) \text{ if } Y^e = 0, \\ \mathcal{N}_m(-\theta_{\text{inv}}, 0.1) \text{ if } Y^e = 1, \end{cases} \tag{8}$$

$$Z_{\text{spu}}^e \sim \begin{cases} \mathcal{N}_o(+\theta_{\text{spu}}^e, 0.1) \text{ if } Y^e = 0, \\ \mathcal{N}_o(-\theta_{\text{spu}}^e, 0.1) \text{ if } Y^e = 1, \end{cases}$$

$$Z^e \leftarrow (Z_{\text{inv}}^e, Z_{\text{spu}}^e), \quad X^e \leftarrow S(Z^e).$$

The scrambling matrix $S$ is set to identity in Example 3 and a random unitary matrix is selected to rotate the latents in Example 3S. In the above dataset, the invariant features are anti-causally related to the label $Y^e$. The spurious features carry extra information about the label not contained in the invariant features.

**AC-CMNIST dataset (PIIF).** We follow the same construction as was proposed in Arjovsky et al. (2019). We set up a binary classification task– identify whether the digit is less than 5 (not including 5) or more than 5. There are three environments – two training environments containing 25,000 data points each, one test environment containing 10,000 points. Define a preliminary label $\tilde{Y} = 0$ if the digit is between 0-4 and $\tilde{Y} = 1$ if the digit is between 5-9. We add noise to this preliminary label by flipping it with a 25 percent probability to construct the final label. We flip the final labels to obtain the color id $Z_{\text{spu}}^e$, where the flipping probabilities are environment-dependent. The flipping probabilities are 0.2, 0.1, and 0.9, in the first, second, and third environment respectively. The third environment is the testing environment. If $Z_{\text{spu}}^e = 1$, we color the digit red, otherwise we color it to be green. In this dataset, the color (spurious feature) carries extra information about the label not contained in the uncolored image.

**CS-CMNIST dataset (FIIF).** We follow the same construction based on Ahuja et al. (2021b), except instead of a binary classification task, we set up a ten-class classification task, where the ten classes are

the ten digits. For each digit class, we have an associated color.[10] There are also three environments – two training environments containing 20,000 data points each, one test containing 20,000 points. In the two training environments, the $p^e$ is set to 1.0 and 0.9, i.e., given the digit label the image is colored with the associated color with probability $p^e$ and with a random color with probability $1 - p^e$. In the testing environment, the $p^e$ is set to 0, i.e., all the images are colored completely at random. In this dataset, the color (spurious feature) does not carry any extra information about the label that is not already contained in the uncolored image.

**Terra Incognita dataset (FIIF).** This dataset is a subset of the Caltech Camera Traps dataset (Beery et al., 2018) as formulated in Gulrajani and Lopez-Paz (2021). We set up a ten-class classification task for $3 \times 224 \times 224$ images - identifying between 9 different species of wild animal and no animal ({ bird, bobcat, cat, coyote, dog, empty, opossum, rabbit, raccoon, squirrel}). There are four domains - {L100, L38, L43, L46} - which represents different locations of the cameras in the American Southwest. For a given location the background never change, except for illumination difference across the time of day and vegetation changes across seasons. The data is unbalanced in the number of images per location, distribution of species per location, and distribution of species overall.

**COCO dataset (FIIF).** We use COCO on colours dataset described in Ahmed et al. (2021) (See the details in Appendix A.2 of Ahmed et al. (2021)). There are ten object classes and for each object class there is a majority color associated with it, i.e., an object class assumes the background color assigned to it with 0.8 probability. At test time, the object backgrounds are colored randomly with colors different from the ones seen in training.

### A.2.2 Training and evaluation procedure

**Example 1/1S, 2/2S, 3/3S.** We follow the same protocol as was prescribed in Aubin et al. (2021) for the model selection, hyperparameter selection, training, and evaluation. For all three examples, the models used are linear. The training loss is the square error for the regression setting (Example 1/1S), and binary cross-entropy for the classification setting (Example 2/2S, 3/3S). For the two new approaches, IB-IRM, and IB-ERM, there is a new hyperparameter $\gamma$ associated with the $\mathrm{Var}(\Phi)$ term in the final objective in equation (7). We use random hyperparameter search and use 20 hyperparameter queries and average over 50 data seeds; these numbers are the same as what was used in Aubin et al. (2021). We sample the $\gamma$ from $1 - 10^{\mathrm{Uniform}(-2,0)}$ following the practice in unit test experiments (Aubin et al., 2021). Note that the hyperparameters are trained using training environment distribution data, which is called the train-domain validation set evaluation procedure in Gulrajani and Lopez-Paz (2021). For the evaluation of performance on Example 1/1s, we reported mean square errors and standard deviations. For the evaluation of performance on Example 2/2S, Example 3/3s, we reported classification errors and standard deviations.

**AC-CMNIST dataset.** We use the default MLP architecture from `https://github.com/facebookresearch/InvariantRiskMinimization`. There are two fully connected layers each with output size 256, ReLU activation, and $\ell_2$-regularizer coefficient of $1e-3$. These layers are followed by the output layer of size two. We use Adam optimizer for training with a learning rate set to $1e-3$. We optimize the cross-entropy loss function. We set the batch size to 256. The total number of steps is set to 500. We use grid search to search the following hyperparameters, $\lambda$ for IRMv1 penalty, and $\gamma$ for the IB penalty. For IRM, we need to select the IRMv1 penalty $\lambda$, we set a grid of 25 values uniformly spaced in the interval $[1e - 1, 1.8e4]$. For IB-ERM, we need to select the IB penalty $\gamma$, we set a grid of 25 values uniformly spaced in the interval $[1e - 1, 1.8e4]$. For IB-IRM, we need to select both $\lambda$ and $\gamma$, we set a $5 \times 5$ uniform grid that searches over $[1e - 1, 1.8e4] \times [1e - 1, 1.8e4]$. Thus for IB-IRM, IB-ERM, and IRM, we search over 25 hyperparameter values. There are two procedures we tried to tune the hyperparameters – a) train-domain validation set tuning procedure (Gulrajani and Lopez-Paz, 2021) which takes samples from the same distribution as train domain and does limited model queries (we set 25 queries), b) oracle test-domain validation set hyperparameter tuning procedure (Gulrajani and Lopez-Paz, 2021), which takes samples from the same distribution as test domain and does limited model queries (we set 25 queries). In Arjovsky et al. (2019), the authors had used oracle test-domain validation set-based tuning, which is not ideal and is a limitation of all current approaches on AC-CMNIST. We used the same procedure in Table 4 (5 percent of the total data 50000 follows the test environment distribution). In Section A.2.3, we show the results for

---

[10]The list of the RGB values for the ten colors are: [0, 100, 0], [188, 143, 143], [255, 0, 0], [255, 215, 0], [0, 255, 0], [65, 105, 225], [0, 225, 225], [0, 0, 255], [255, 20, 147], [160, 160, 160].

all the methods when we use train-domain validation set tuning. For the evaluation, we reported the accuracy and standard deviations (averaged over thirty trials).

**CS-CMNIST dataset.** We use a ConvNet architecture with three convolutional layers with feature map dimensions of 64,128 and 256. Each convoluional layer is followed by a ReLU activation and batch normalization layer. The final output layer is a linear layer with output dimension equal to the number of classes. We use SGD optimizer for training with a learning rate set to $1e - 1$ and decay every 600 steps. We optimize the cross-entropy loss function without weight decay. We set the batch size to 128. The total number of steps is set to 2000. We use grid search to search the following hyperparameters, $\lambda$ for IRMv1 penalty, and $\gamma$ for the IB penalty. For IRM, we need to select the IRMv1 penalty $\lambda$, we set a grid of 25 values uniformly spaced in the interval $[1e - 1, 1.8e4]$. For IB-ERM, we need to select the IB penalty $\gamma$, we set a grid of 25 values uniformly spaced in the interval $[1e - 1, 1.8e4]$. For IB-IRM, we need to select both $\lambda$ and $\gamma$, we set a $5 \times 5$ uniform grid that searches over $[1e - 1, 1.8e4] \times [1e - 1, 1.8e4]$. Thus for IB-IRM, IB-ERM, and IRM, we search over 25 hyperparameter values. In the paragraph above, we described that for AC-CMNIST all the procedures only work when using the oracle test-domain validation procedure. In the results of the CS-CMNIST experiment in the main manuscript, we showed results for the train domain validation procedure and found that IB-IRM and IB-ERM yield better performance. For completeness, we also carried oracle test-domain validation procedure-based hyperparameter tuning for CS-CMNIST and the results are discussed in Section A.2.3. For the evaluation, we reported accuracy and standard deviations (averaged over five trials). In both CMNIST datasets, we had experimented with placing the IB penalty at the output layer (logits) and the penultimate layer (layer just before the logits), and found that it is much more effective to place the IB penalty on the penultimate layer. Thus in both the CMNIST datasets, the results presented use IB penalty on the penultimate layer.

**Terra Incognita dataset.** We use the pretrained ResNet-50 model as a featurizer that outputs feature maps of size 2048 for a given image on top of which we add a 1 layer MLP which makes the classification $(2048 \rightarrow 9)$. We use a random hyper parameter sweep over 20 random hyperparameter configurations on which we look at the train-domain validation set to perform model selection, as described in Gulrajani and Lopez-Paz (2021). The distribution of the hyper parameters are shown in Table 5. Results shown in Table 4 are for the environment L100 as test environment, the reported accuracies are averaged over 3 random trial seed. For both the information bottleneck penalized algorithms (IB-ERM and IB-IRM), we apply the penalty on the feature map given by the featurizer, conditional on the environment.

Table 5: Hyperparameters distributions for random search given included penalty of the algorithm.

| Penalty | Parameter | Random distribution |
|---|---|---|
| All | dropout | $\mathrm{RandomChoice}([0, 0.1, 0.5])$ |
|  | learning rate | $10^{\mathrm{Uniform}(-5, -3.5)}$ |
|  | batch size | $2^{\mathrm{Uniform}(3, 5.5)}$ |
|  | weight decay | $10^{\mathrm{Uniform}(-6, -2)}$ |
| IRMv1 | penalty weight | $10^{\mathrm{Uniform}(-1, 5)}$ |
|  | annealing steps | $10^{\mathrm{Uniform}(0, 4)}$ |
| IB | penalty weight | $10^{\mathrm{Uniform}(-1, 5)}$ |
|  | annealing steps | $10^{\mathrm{Uniform}(0, 4)}$ |

**COCO dataset.** Other than the IB penalty, we use the exact same hyperparameters (default values) and setup as describe in Appendix B.2 of Ahmed et al. (2021) paper and the codebase that Ahmed et al. (2021) paper provides. For all experiments that involve an IB loss term component, IB penalty weighting of 1.0 is used and IB penalty weighting is linearly ramped up to 1.0 from epoch 1 to 200. For all experiments that involve an IRM loss term component, IRM penalty weighting of 1.0 is used, and IRM penalty weighting is linearly ramped up to 1.0 from epoch 1 to 200. Batch size of 64 is used for all experiments. We do not tune the hyperparameters in this experiment. Mean and standard deviation of classification accuracy are obtained via 4 seeds for each method.

### A.2.3 Supplementary experiments

**AC-CMNIST.** In the AC-CMNIST dataset, for completeness, we report the accuracy of the Oracle model, where the Oracle model at train time is fed images where the background colors do not have any correlation with the label. Oracle model achieved a test accuracy $70.39 \pm 0.47$ percent. In Table 5, we provide the supplementary experiments for AC-CMNIST carried out with train-domain validation set tuning procedure (Gulrajani and Lopez-Paz, 2021). It can be seen that none of the methods work in this case. In Table 6, we provide the supplementary experiments for AC-CMNIST carried out with test-domain validation set tuning procedure (Gulrajani and Lopez-Paz, 2021). In this case, both IB-IRM and IRM perform well.

| Method | 5% | 10% | 15% | 20% |
|---|---|---|---|---|
| ERM | $17.17 \pm 0.62$ | $18.06 \pm 1.72$ | $18.74 \pm 1.23$ | $19.11 \pm 1.18$ |
| IB-ERM | $17.69 \pm 0.54$ | $17.80 \pm 1.81$ | $16.27 \pm 1.20$ | $18.18 \pm 1.46$ |
| IRM | $16.48 \pm 2.50$ | $17.85 \pm 1.67$ | $17.32 \pm 2.12$ | $18.09 \pm 2.78$ |
| IB-IRM | $18.37 \pm 1.44$ | $17.83 \pm 0.65$ | $18.54 \pm 1.42$ | $19.24 \pm 1.49$ |

Table 6: AC-CMNIST. Comparisons of the methods using the train-domain validation set tuning procedure (Gulrajani and Lopez-Paz, 2021). The percentages in the columns indicate what fraction of the total data (50000 points) is used for validation.

| Method | 5% | 10% | 15% | 20% |
|---|---|---|---|---|
| ERM | $16.84 \pm 0.82$ | $17.01 \pm 0.83$ | $16.79 \pm 0.89$ | $16.27 \pm 0.93$ |
| IB-ERM | $50.24 \pm 0.47$ | $50.25 \pm 0.46$ | $50.52 \pm 0.45$ | $50.34 \pm 0.56$ |
| IRM | $66.98 \pm 1.65$ | $67.57 \pm 1.39$ | $67.01 \pm 1.86$ | $67.29 \pm 1.62$ |
| IB-IRM | $67.67 \pm 1.78$ | $68.22 \pm 1.62$ | $67.56 \pm 1.71$ | $67.24 \pm 1.36$ |

Table 7: CS-CMNIST. Comparisons of the methods using the oracle test-domain validation set tuning procedure (Gulrajani and Lopez-Paz, 2021). The percentages in the columns indicate what fraction of the total data (50000 points) is used for validation.

**AC-CMNIST.** In the CS-CMNIST dataset, for completeness, we report the accuracy of the Oracle model, which achieved a test accuracy of $99.03 \pm 0.08$ percent. In Table 7, we provide the supplementary experiments for CS-CMNIST carried out with train-domain validation set tuning procedure (Gulrajani and Lopez-Paz, 2021). In Table 8, we provide the supplementary experiments for CS-CMNIST carried out with test-domain validation set tuning procedure (Gulrajani and Lopez-Paz, 2021). In both cases, both IB-IRM and IB-ERM RM perform well. Unlike AC-CMNIST, in the CS-CMNIST dataset both the validation procedures lead to a similar performance.

| Method | 5% | 10% | 15% | 20% |
|---|---|---|---|---|
| ERM | $60.27 \pm 1.21$ | $61.02 \pm 0.59$ | $60.35 \pm 1.01$ | $58.59 \pm 1.67$ |
| IB-ERM | $71.80 \pm 0.69$ | $71.51 \pm 1.01$ | $71.27 \pm 1.04$ | $70.68 \pm 1.02$ |
| IRM | $61.49 \pm 1.45$ | $61.74 \pm 1.28$ | $60.01 \pm 0.59$ | $59.96 \pm 0.96$ |
| IB-IRM | $71.79 \pm 0.70$ | $71.57 \pm 1.01$ | $71.37 \pm 0.62$ | $70.65 \pm 0.90$ |

Table 8: CS-CMNIST. Comparisons of the methods using the train-domain validation set tuning procedure (Gulrajani and Lopez-Paz, 2021). The percentages in the columns indicate what fraction of the total data (50000 points) is used for validation.

| Method | 5% | 10% | 15% | 20% |
|---|---|---|---|---|
| ERM | $61.27 \pm 1.40$ | $61.02 \pm 1.59$ | $60.35 \pm 1.01$ | $58.59 \pm 1.67$ |
| IB-ERM | $71.65 \pm 0.76$ | $71.68 \pm 1.23$ | $71.27 \pm 0.89$ | $70.07 \pm 1.18$ |
| IRM | $62.00 \pm 1.60$ | $62.01 \pm 1.33$ | $60.26 \pm 0.51$ | $59.96 \pm 0.96$ |
| IB-IRM | $71.90 \pm 0.78$ | $71.07 \pm 0.95$ | $71.18 \pm 0.80$ | $70.75 \pm 1.00$ |

Table 9: CS-CMNIST. Comparisons of the methods using the oracle test-domain validation set tuning procedure (Gulrajani and Lopez-Paz, 2021). The percentages in the columns indicate what fraction of the total data (50000 points) is used for validation

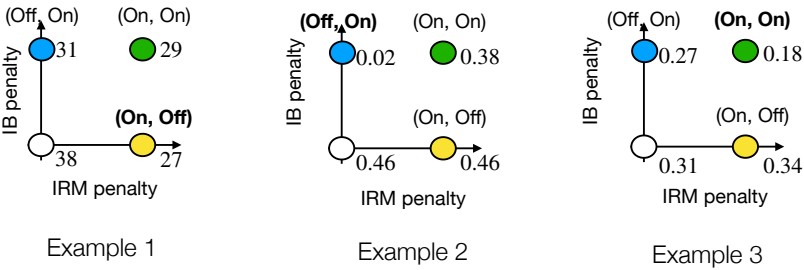

Figure 4: Illustrating the impact of the IB and IRM penalty on linear unit tests (Aubin et al., 2021)

**Ablation to understand the role of invariance penalty and information bottleneck.** In the main body, we compared IB-IRM, IB-ERM, IRM, and ERM with the penalty of the respective methods tuned using the validation procedures from Gulrajani and Lopez-Paz (2021). In this section, we carry out an ablation analysis on linear unit tests (Aubin et al., 2021) to understand the role of the different penalties. In Figure 4, for each example we consider the setting with six environments and show four points on a square with corresponding performance values. The bottom corner corresponds to ERM when both penalties are turned off, top corner is when both penalties are turned on, and the other two corners are when one of the penalties are on. In Example 1, which corresponds to PIIF setting, we find that IRM penalty alone helps the most. In Example 2, which corresponds to FIIF setting, we find that IB penalty helps the most. In Example 3, which again corresponds to PIIF, we find that both penalties help.

### A.2.4 Compute description

Our computing resource is one Tesla V100-SXM2-16GB with 18 CPU cores.

### A.2.5 Assets used and the license details

In this work, we mainly relied on the following github repositories – Domainbed[11], IRM [12], linear unit tests[13]. All the repositories mentioned above use the MIT license. We used the standard MNIST dataset [14] to generate the colored MNIST datasets. Other datasets we used are synthetic.

---

[11]`https://github.com/facebookresearch/DomainBed` based on Gulrajani and Lopez-Paz (2021)

[12]`https://github.com/facebookresearch/InvariantRiskMinimization` based on Arjovsky et al. (2019)

[13]`https://github.com/facebookresearch/InvarianceUnitTests` based on Aubin et al. (2021)

[14]`http://yann.lecun.com/exdb/mnist/`

### A.3 Background on structural equation models

For completeness, we provide a more detailed background on structural equation models (SEMs), which is borrowed from Arjovsky et al. (2019).

#### A.3.1 Structural equation models and assumptions on $\mathcal{E}_{all}$

**Definition 1.** *A structural equation model $\mathcal{C} = (\mathcal{S}, N)$ that describes the random vector $X = (X_1, \ldots, X_d)$ is given as follows*

$$\mathcal{S}_i : X_i \leftarrow f_i(\mathsf{Pa}(X_i), N_i), \tag{9}$$

*where $\mathsf{Pa}(X_i)$ are the parents of $X_i$, $N_i$ is independent noise, and $N = (N_1, \ldots, N_d)$ is the noise vector. $X_j$ is said to cause $X_i$ if $X_j \in \mathsf{Pa}(X_i)$. We draw the causal graph by placing one node for each $X_i$ and drawing a directed edge from each parent to the child. The causal graphs are assumed to be acyclic.*

**Definition 2.** *An intervention $e$ on $\mathcal{C}$ is the process of replacing one or several of its structural equations to obtain a new intervened SEM $\mathcal{C}^e = (\mathcal{S}^e, N^e)$, with structural equations given as*

$$\mathcal{S}_i^e : X_i^e \leftarrow f_i^e(\mathsf{Pa}(X_i^e), N_i^e), \tag{10}$$

*where the variable $X_i^e$ is said to be intervened if $\mathcal{S}_i \neq \mathcal{S}_i^e$ or $N_i \neq N_i^e$*

The above family of interventions are used to model the environments.

**Definition 3.** *Consider a SEM $\mathcal{C}$ that describes the random vector $(X, Y)$, where $X = (X_1, \ldots, X_d)$, and the learning goal is to predict $Y$ from $X$. The set of all environments obtained using interventions $\mathcal{E}_{all}(\mathcal{C})$ indexes all the interventional distributions $\mathbb{P}^e$, where $(X^e, Y^e) \sim \mathbb{P}^e$. An intervention $e$ is valid if the following conditions are met: i) the causal graph remains acyclic, ii) $\mathbb{E}[Y^e|\mathsf{Pa}(Y)] = \mathbb{E}[Y|\mathsf{Pa}(Y)]$, i.e. expectation conditional on parents is invariant, and the variance $\mathsf{Var}[Y^e|\mathsf{Pa}(Y)]$ remains within a finite range.*

Following the above definitions it is possible to show that a predictor that relies on causal parents only $v : \mathbb{R}^d \to \mathcal{Y}$ and is given as $v(x) = \mathbb{E}[f_Y(\mathsf{Pa}(Y), N_Y)]$ solves the OOD generalization problem in equation (1) over the environments $\mathcal{E}_{all}(\mathcal{C})$ that form valid interventions as stated in Definition 3. Next, we provide an example to show why $v$ is OOD optimal.

**Example to illustrate why predictors that rely on causes are robust.** We reuse the toy example from Arjovsky et al. (2019) to explain why models that rely on causes are more robust to valid interventions $\mathcal{E}_{all}$ discussed in the previous section.

$$
\begin{aligned}
Y^e &\leftarrow X_{\mathsf{inv}}^e + \epsilon^e \\
X_{\mathsf{spu}}^e &\leftarrow Y^e + \zeta^e
\end{aligned} \tag{11}
$$

where $X_{\mathsf{inv}}^e \in \mathcal{N}(0, (\sigma^e)^2)$ is the cause of $Y^e$, $N^e \in \mathcal{N}(0, (\sigma^e)^2)$ is noise, $X_{\mathsf{spu}}^e$ is the effect of $Y^e$ and $\zeta^e \in \mathcal{N}(0, 1)$ is also noise. Suppose there are two training environments $\mathcal{E}_{tr} = \{e_1, e_2\}$, in the first $(\sigma^{e_1})^2 = 1$ and in the second $(\sigma^{e_2})^2 = 2$. The three possible models $w_{\mathsf{inv}} X_{\mathsf{inv}}^e + w_{\mathsf{spu}} X_{\mathsf{spu}}^e$ we could build are as follows: a) regress only on $X_{\mathsf{inv}}^e$, then in the optimal model $w_{\mathsf{inv}} = 1, w_{\mathsf{spu}} = 0$, b) regress only on $X_{\mathsf{spu}}^e$ and get $w_{\mathsf{inv}} = 0, w_{\mathsf{spu}} = \frac{\sigma^2}{(\sigma^e)^2 + \frac{1}{2}}$, c) regress on $(X_{\mathsf{inv}}^e, X_{\mathsf{spu}}^e)$ to get $w_{\mathsf{inv}} = \frac{1}{(\sigma^e)^2 + 1}$ and $w_{\mathsf{inv}} = \frac{(\sigma^e)^2}{(\sigma^e)^2 + 1}$. Observe that the predictor that focuses on the cause only does not depend on $\sigma^2$ and is thus invariant to distribution shifts induced by change in $(\sigma^e)^2$, which is not the case with the other models. For environment in $\mathcal{E}_{all} \setminus \mathcal{E}_{tr}$ we can change the distribution of $X_{\mathsf{inv}}^e$ and $X_{\mathsf{spu}}^e$ arbitrarily. Consider an environment $e \in \mathcal{E}_{all}$ where $X_{\mathsf{spu}}^e$ is set to a very large constant $c$, the square error of the model that relies on spurious features grows with the magnitude of $c$ but the error of the model that relies on $X_{\mathsf{inv}}^e$ does not change. Another remark we would like to make here is that in the main manuscript, we defined the notions of invariant feature map $\Phi^*$, and spurious feature map $\Psi^*$. Observe that in this example $\Phi^*(X^e) = X_{\mathsf{inv}}^e$, and $\Psi^*(X^e) = X_{\mathsf{spu}}^e$.

#### A.3.2 Remark on the linear general position assumption and its implications on support overlap

In Theorem 1 that we informally stated from Arjovsky et al. (2019), there is one more technical condition on that we explain below. We also explain how this assumption does not restrict the support of the latents $Z^e$ from changing arbitrarily.

**Assumption 8.** *Linear general position. A set of training environments $\mathcal{E}_{tr}$ lie in a linear general position of degree $r$ if $|\mathcal{E}_{tr}| > d - r + \frac{d}{r}$ for some $r \in \mathbb{N}$ and for all non-zero $x \in \mathbb{R}^d$*

$$\dim\left(\mathsf{span}\left(\left\{\mathbb{E}_{X^e}[X^e X^{e\mathsf{T}}]x - \mathbb{E}_{X^e \epsilon^e}[X^e \epsilon^e]\right\}_{e \in \mathcal{E}_{tr}}\right)\right) > d - r. \tag{12}$$

The above assumption merely requires non-co-linearity of the training environments only. The set of matrices $\mathbb{E}_{X^e}[X^e X^{e\mathsf{T}}]$ not satisfying this assumption have a zero measure (Theorem 10 Arjovsky et al. (2019)). Consider the case when $S$ is identity and observe that the above assumption translates to only a restriction on co-linearity of $\mathbb{E}_{Z^e}[Z^e Z^{e\mathsf{T}}]$, where $Z^e = (Z^e_{\mathsf{inv}}, Z^e_{\mathsf{spu}})$. Assume that $\mathbb{E}_{Z^e}[Z^e Z^{e\mathsf{T}}]$ is positive definite. We explain how this Assumption 8 does not constraint the support of the latent random variables $Z^e$. From the set of matrices $\mathbb{E}_{Z^e}[Z^e Z^{e\mathsf{T}}]$ and $\mathbb{E}_{Z^e}[Z^e \epsilon^e]$ that satisfy the Assumption 8, we can construct another set of matrices with norm one that satisfy the above Assumption 8. Define a random variable $\tilde{Z}^e = \frac{Z^e}{c}$ and the matrices corresponding to it also satisfy the Assumption 8, where $c = \sqrt{\|\mathbb{E}_{Z^e}[Z^e Z^{e\mathsf{T}}]\|}$.

For all non-zero $z \in \mathbb{R}$,

$$\dim\left(\mathsf{span}\left(\left\{\mathbb{E}_{Z^e}[Z^e Z^{e\mathsf{T}}]z - \mathbb{E}_{Z^e \epsilon^e}[Z^e \epsilon^e]\right\}_{e \in \mathcal{E}_{tr}}\right)\right) > d - r \implies$$
$$\dim\left(\mathsf{span}\left(\left\{\mathbb{E}_{\tilde{Z}^e}[\tilde{Z}^e \tilde{Z}^{e\mathsf{T}}]\tilde{z} - \mathbb{E}_{\tilde{Z}^e \epsilon^e}[\tilde{Z}^e \epsilon^e]\right\}_{e \in \mathcal{E}_{tr}}\right)\right) > d - r, \tag{13}$$

where $\tilde{z} = zc$. Define $\Sigma^e = \mathbb{E}[Z^e Z^{e\mathsf{T}}]$ ($\tilde{\Sigma}^e = \mathbb{E}[\tilde{Z}^e \tilde{Z}^{e\mathsf{T}}]$) and $\rho^e = \mathbb{E}[Z^e \epsilon^e]$ ($\tilde{\rho}^e = \mathbb{E}[\tilde{Z}^e \epsilon^e]$). Observe that $\|\tilde{\Sigma}^e\| = 1$. So far we established that if there exist a set of matrices $\{\Sigma^e, \rho^e\}_{e \in \mathcal{E}_{tr}}$ satisfying the linear general position assumption (Assumption 8), then it also implies that there exist a set of matrices $\{\tilde{\Sigma}^e, \tilde{\rho}^e\}_{e \in \mathcal{E}_{tr}}$, where $\|\tilde{\Sigma}^e\| = 1$, that satisfy the linear general position assumption (Assumption 8). Next, we will show that the set of matrices $\{\tilde{\Sigma}^e\}_{e \in \mathcal{E}_{tr}}, \{\tilde{\rho}^e\}_{e \in \mathcal{E}_{tr}}$ can be constructed from random variables with bounded support. We will show that $\tilde{\Sigma}^e$ can be constructed by transforming a uniform random vector. Define a uniform random vector $K^e$, where each component $K^e_i \sim \mathsf{Uniform}[-\sqrt{3}, \sqrt{3}]$. Define $\bar{Z}^e = BK^e$. Observe that

$$\mathbb{E}[\bar{Z}^e \bar{Z}^{e,\mathsf{T}}] = BB^t. \tag{14}$$

Since every positive definite matrix can be decomposed as $BB^t$, we can use matrix $B$ to construct the required $\tilde{\Sigma}^e$. Since $\|\tilde{\Sigma}^e\| = 1$, we get $\|BB^t\| = 1 \implies \|B\| = 1$. Also, $\|\bar{Z}^e\| \leq \|B\|\|K^e\| = \|K^e\|$. Having fixed the matrix $B$ above, we use it to set the correlation $\mathbb{E}[K^e \epsilon^e]$

$$B\mathbb{E}[K^e \epsilon^e] = \tilde{\rho}^e \implies \mathbb{E}[K^e \epsilon^e] = B^{-1}\tilde{\rho}^e \tag{15}$$

Thus we can conclude without loss of generality that from any set of matrices $\{\Sigma^e, \rho^e\}_{e \in \mathcal{E}_{tr}}$ satisfying the linear general position assumption, we can construct random variables with bounded support that satisfy the linear general position assumption. By solving IRM (equation (3)) over such training environments with bounded support, we can still recover the ideal invariant predictor that solves the OOD generalization problem in equation (1) (i.e., $\nexists e \in \mathcal{E}_{all}$ for which risk $> \sigma^2_{\mathsf{sup}}$). The above conditions show that we can have the data in $\mathcal{E}_{tr}$ come from a region with bounded support, and the environments in $\mathcal{E}_{all} \setminus \mathcal{E}_{tr}$ are not required to satisfy support overlap with data from $\mathcal{E}_{tr}$, which is in stark contrast to the linear classification results that we showed.

### A.4 Notations and proof of Theorem 2 (impossibility of guaranteed OOD generalization for linear classification)

**Notations for the proofs.** We describe the common notations used in the proofs that follow. We also remind the reader of the notation from the main manuscript for convenience. $\circ$ is used to denote the composition of functions, $\cdot$ is used for matrix multiplication. $\mathbb{P}^e$ denotes the probability distribution over the input feature values $X^e$, and the labels $Y^e$ in environment $e$. $Z^e$ describes the latent variables decomposed into $(Z^e_{\text{inv}}, Z^e_{\text{spu}})$. $S$ is the matrix relating $X^e$ and $Z^e$ and $X^e = S(Z^e)$. $w$ denotes a linear classifier, $\Phi$ denotes the representation map that transforms input data into a representation, which is then fed to the classifier. $\mathsf{I}$ is the indicator function, which takes a value $1$ when the input is greater than or equal to zero, and $0$ otherwise. $\mathsf{sgn}$ is the sign function, which takes a value $1$ when the input is greater than or equal to zero, and $-1$ otherwise. In all the results, except for Theorem 5, we use $\ell$ as 0-1 loss for classification, and square loss for regression. For a discrete random variable $X \in \mathbb{R}^d$, the support is defined as $\mathcal{X} = \{x \in \mathbb{R}^d \mid \mathbb{P}_X(x) > 0\}$, where $\mathbb{P}_X(x)$ is the probability of $X = x$. For a continuous random variable $X \in \mathbb{R}^d$, the support is defined as $\mathcal{X} = \{x \in \mathbb{R}^d \mid d\mathbb{P}_X(x) > 0\}$, where $d\mathbb{P}_X(x)$ is the Radon-Nikodym derivative of $\mathbb{P}_X$ w.r.t the Lebesgue measure over the completion of the Borel sets in $\mathbb{R}^d$ (Ash and Doléans-Dade, 2000). $\mathcal{Z}^e$, $\mathcal{Z}^e_{\text{inv}}$, $\mathcal{Z}^e_{\text{spu}}$, and $\mathcal{X}^e$ are the support of $Z^e$, $Z^e_{\text{inv}}$, $Z^e_{\text{spu}}$, and $X^e$ respectively in environment $e$.

**Remark on Assumption 2.** In all the proofs that follow, we assume that the dimension of invariant feature $m$ is greater than or equal to 2. Also, all the components $w^*_{\text{inv}}$ are non-zero without loss of generality (if some component was zero, then such a latent can be a part of $Z^e_{\text{spu}}$. $\mathcal{X} = \mathbb{R}^d$ and $\mathcal{Y} = \{0, 1\}$ for classification and $\mathcal{Y} = \mathbb{R}$ for regression. Before we can prove Theorem 2, we need to prove intermediate lemmas needed as preliminary results for it.

Define

$$\mathcal{W}_{\text{inv}} = \left\{ (w_{\text{inv}}, 0) \in \mathbb{R}^{m+o} \mid \|w_{\text{inv}}\| = 1, \ \forall z_{\text{inv}} \in \cup_{e \in \mathcal{E}_{tr}} \mathcal{Z}^e_{\text{inv}}, \ \mathsf{I}(w^*_{\text{inv}} \cdot z_{\text{inv}}) = \mathsf{I}(w_{\text{inv}} \cdot z_{\text{inv}}) \right\} \tag{16}$$

This set $\mathcal{W}_{\text{inv}}$ defines a family of hyperplanes equivalent to the labelling hyperplane $w^*_{\text{inv}}$ on the training environments. Define a classifier $g^* : \mathcal{X} \to \mathcal{Y}$ as

$$g^* = \mathsf{I} \circ \left( (w^*_{\text{inv}}, 0) \circ S^{-1} \right) \tag{17}$$

The classifier $g^*$ takes $X^e$ as input and outputs $\mathsf{I}(w^*_{\text{inv}} \cdot Z^e_{\text{inv}})$.

**Lemma 1.** *If we consider the set of all the environments that follow Assumption 2, then the classifier based on the labelling hyperplane $g^*$ solves equation (1) and achieves a risk of $q$ in each environment.*

**Proof of Lemma 1.** Observe that $g^*$ is the classifier one would get by solving for the Bayes optimal classifier on each environment. The justification goes as follows. If $w^*_{\text{inv}} \cdot Z^e_{\text{inv}} \geq 0$, then $\mathbb{P}(Y^e = 0|X^e) < \mathbb{P}(Y^e = 1|X^e)$ (since $q < \frac{1}{2}$), which implies the prediction is 1. If $w^*_{\text{inv}} \cdot Z^e_{\text{inv}} < 0$, then $\mathbb{P}(Y^e = 1|X^e) < \mathbb{P}(Y^e = 0|X^e)$, which implies the prediction is 0. We show that $g^*$ achieves an error of $q$ in each environment,

$$\begin{aligned} R^e(g^*) &= \mathbb{E}\left[ Y^e \oplus \mathsf{I}(w^*_{\text{inv}} \cdot Z^e_{\text{inv}}) \right] \\ &= \mathbb{E}\left[ \left( \mathsf{I}(w^*_{\text{inv}} \cdot Z^e_{\text{inv}}) \oplus N^e \right) \oplus \mathsf{I}(w^*_{\text{inv}} \cdot Z^e_{\text{inv}}) \right] = q. \end{aligned} \tag{18}$$

Define $\mathcal{F}$ to be the set of all the maps $\mathbb{R}^d \to \mathcal{Y}$. From the equation (18) we get,

$$\begin{aligned} &\forall e \in \mathcal{E}_{all}, \forall f \in \mathcal{F}, \ R^e(f) \geq q, \\ \implies &\forall f \in \mathcal{F}, \ \max_{e \in \mathcal{E}_{all}} R^e(f) \geq q, \\ \implies &\min_{f \in \mathcal{F}} \max_{e \in \mathcal{E}_{all}} R^e(f) \geq q. \end{aligned} \tag{19}$$

$g^*$ achieves the lower bound above as it achieves an error of $q$ in each environment. This completes the proof. $\qquad\square$

We relax the Assumption 2 to the case where we allow for spurious features to carry extra information about the label.

**Assumption 9.** *Linear classification structural equation model. (PIIF) In each $e \in \mathcal{E}_{all}$,*

$$Y^e \leftarrow \mathsf{I}\big(w_{\text{inv}}^* \cdot Z_{\text{inv}}^e\big) \oplus N^e, \qquad N^e \sim \text{Bernoulli}(q), q < \frac{1}{2}, \qquad N^e \perp Z_{\text{inv}}^e, \tag{20}$$

$$X^e \leftarrow S\big(Z_{\text{inv}}^e, Z_{\text{spu}}^e\big).$$

Observe that the SEM above in Assumption 9 is analogous the the SEM in Assumption 1. Also, observe that in the above SEM $\exists\, e$ such that $N^e \not\perp Z_{\text{spu}}^e$, which makes the invariant features partially informative about the label. We show that the Lemma 1 extends to the above SEMs (Assumption 9) as well.

**Lemma 2.** *If we consider the set of all the environments that follow Assumption 9, then $g^*$ solves equation* (1) *and achieves a risk of $q$ in each environment.*

**Proof of Lemma 2.** Consider the environment $e' \in \mathcal{E}_{all}$, where $N^{e'} \perp (Z_{\text{inv}}^{e'}, Z_{\text{spu}}^{e'})$. Observe that in this environment $g^*$ is a Bayes optimal classifier and achieves a risk value of $q$.

$$\forall f \in \mathcal{F}, R^{e'}(f) \geq q \implies \forall f \in \mathcal{F}, \max_{e \in \mathcal{E}_{all}} R^e(f) \geq q,$$
$$\implies \min_{f \in \mathcal{F}} \max_{e \in \mathcal{E}_{all}} R^e(f) \geq q \tag{21}$$

$g^*$ achieves the lower bound above as it achieves an error of $q$ in each environment. This completes the proof. $\qquad\qquad\square$

**Lemma 3.** *If Assumption 2, 3, and 7 hold, and $m \geq 2$, then the set $\mathcal{W}_{\text{inv}}$ (eq.* (16)*) consists of infinitely many hyperplanes that are not aligned with $w_{\text{inv}}^*$.*

**Proof of Lemma 3.** For each $z_{\text{inv}} \in \cup_{e \in \mathcal{E}_{tr}} \mathcal{Z}_{\text{inv}}^e$ define $y^* = \text{sgn}(w_{\text{inv}}^* \cdot z_{\text{inv}})$.

From the definition of Inv-Margin in Assumption 7, it follows that $\exists\, c > 0$ such that $\forall z_{\text{inv}} \in \cup_{e \in \mathcal{E}_{tr}} \mathcal{Z}_{\text{inv}}^e$

$$y^*\big(w_{\text{inv}}^* \cdot z_{\text{inv}}\big) \geq c. \tag{22}$$

Next, we choose a $\gamma \in \mathbb{R}^m$ that is not in the same direction as $w_{\text{inv}}^*$, i.e., $\nexists\, a \in \mathbb{R}$ such that $\gamma = aw_{\text{inv}}^*$ (such a direction always exists since $m \geq 2$). Define the margin of $w_{\text{inv}}^* + \gamma$ w.r.t labels $y^*$ from $w_{\text{inv}}^*$

$$y^*\big(w_{\text{inv}}^* \cdot z_{\text{inv}} + \gamma \cdot z_{\text{inv}}\big). \tag{23}$$

Using Cauchy-Schwarz inequality we get

$$|y^*(\gamma \cdot z_{\text{inv}})| = |\gamma \cdot z_{\text{inv}}| \leq \|\gamma\| \|z_{\text{inv}}\|. \tag{24}$$

Since the support of the invariant features in training set $\cup_{e \in \mathcal{E}_{tr}} \mathcal{Z}_{\text{inv}}^e$ is bounded, we set the magnitude of $\gamma$ sufficiently small to control $y^*\big(\gamma \cdot z_{\text{inv}}\big)$. Since $\cup_{e \in \mathcal{E}_{tr}} \mathcal{Z}_{\text{inv}}^e$, is bounded $\exists\, z^{\text{sup}} > 0$ such that $\forall z_{\text{inv}} \in \cup_{e \in \mathcal{E}_{tr}} \mathcal{Z}_{\text{inv}}^e, \|z_{\text{inv}}\| < z_{\text{sup}}$. If $\|\gamma\| \leq \frac{c}{2z^{\text{sup}}}$, then from equation (24), we get that for each $z_{\text{inv}} \in \cup_{e \in \mathcal{E}_{tr}} \mathcal{Z}_{\text{inv}}^e, |y(\gamma \cdot z_{\text{inv}})| \leq \frac{c}{2}$. Using this we get for each $z_{\text{inv}} \in \cup_{e \in \mathcal{E}_{tr}} \mathcal{Z}_{\text{inv}}^e$

$$y^*\big((w_{\text{inv}}^* + \gamma) \cdot z_{\text{inv}}\big) = y^*\big(w_{\text{inv}}^* \cdot z_{\text{inv}}\big) + y^*\big(\gamma \cdot z_{\text{inv}}\big) \geq y^* w_{\text{inv}} \cdot z_{\text{inv}} - |y^* \gamma \cdot z_{\text{inv}}| \geq \frac{c}{2}. \tag{25}$$

From equation (22) and (25), we have that

$$\text{sgn}\big((w_{\text{inv}}^* + \gamma) \cdot z_{\text{inv}}\big) = \text{sgn}\big(w_{\text{inv}}^* \cdot z_{\text{inv}}\big) \implies \mathsf{I}\big((w_{\text{inv}}^* + \gamma) \cdot z_{\text{inv}}\big) = \mathsf{I}\big(w_{\text{inv}}^* \cdot z_{\text{inv}}\big).$$

The same condition would also hold if we normalized the classifier. As a result,

$$\Big(\frac{1}{\|w_{\text{inv}}^* + \gamma\|}(w_{\text{inv}}^* + \gamma), 0\Big) \in \mathcal{W}_{\text{inv}}.$$

Also, observe that we can construct infinite such vectors that belong to $\mathcal{W}_{\text{inv}}$. A simple way to check this this is consider $\gamma' = \theta\gamma$, where $\theta \in (0, 1)$. The same condition in equation (25) also holds with $\gamma$ replaced with $\gamma'$. We define this set as follows

$$\mathcal{W}_{\text{inv}}(\gamma) = \Big\{\Big(\frac{1}{\|w_{\text{inv}}^* + \theta\gamma\|}(w_{\text{inv}}^* + \theta\gamma), 0\Big) \in \mathbb{R}^{m+o} \mid \theta \in [0, 1]\Big\}, \tag{26}$$

and from the reasoning presented above it follows that $\mathcal{W}_{\text{inv}}(\gamma) \subseteq \mathcal{W}_{\text{inv}}$. This completes the proof.

$\square$

We restate Theorem 2 for convenience.

**Theorem 6.** *Impossibility of guaranteed OOD generalization for linear classification. Suppose each $e \in \mathcal{E}_{all}$ follows Assumption 2. If for all the training environments $\mathcal{E}_{tr}$, the latent invariant features are bounded and strictly separable, i.e., Assumption 3 and 7 hold, then every deterministic algorithm fails to solve the OOD generalization (eq. (1)), i.e., for the output of every algorithm $\exists \ e \in \mathcal{E}_{all}$ in which the error exceeds the minimum required value $q$ (noise level).*

**Proof of Theorem 6.** Consider any algorithm, it takes the data from all the training environments as inputs and outputs a classifier. We write the algorithm as a map $F : \cup_{i=1}^{\infty} \left( \mathcal{X} \times \mathcal{Y} \right)^i \dots |\mathcal{E}_{tr}|$ times $\cup_{i=1}^{\infty}$ $\left( \mathcal{X} \times \mathcal{Y} \right)^i \to \mathcal{Y}^{\mathcal{X}}$, where $F$ takes as input data from each of the training environments and outputs a classifier, which takes as input a data point from $\mathcal{X}$ and outputs the label in $\mathcal{Y}$. For datasets $\{D^e\}_{e \in \mathcal{E}_{tr}}$ from the different training environments the output of the learner is $F\left( \{D^e\}_{e \in \mathcal{E}_{tr}} \right)$. For simplicity of notation, let us denote $F\left( \{D^e\}_{e \in \mathcal{E}_{tr}} \right)$ as $f$. We first show that if $f \neq g^*$, where $g^*$ is defined in equation (17), then the learner cannot be OOD optimal. Take the point $x$ where the $f \neq g^*$. Let $z = S^{-1}(x)$. Define a test environment where $Z^e = z$ occurs with probability 1. In such an environment, the error achieved by $f$ would be $1 - q$ ($\mathbb{E}[f \oplus g^* \oplus N^e] = \mathbb{E}[1 \oplus N^e] = 1 - q$). As a result, $f$ cannot solve equation (1). This observation combined with Lemma 1 leads us to the conclusion that $f = g^*$ is necessary and sufficient to solve equation (1) when $\mathcal{E}_{all}$ follow Assumption 2.

We define a family of classifiers using $\mathcal{W}_{\text{inv}}$ (from eq. (16)) as follows

$$\mathcal{W}_{\text{inv}}^{\dagger} = \left\{ \mathsf{I} \circ \left( (w, 0) \circ S^{-1} \right) \ \big| \ (w, 0) \in \mathcal{W}_{\text{inv}} \right\}. \tag{27}$$

Next, we would like to show that the set $\mathcal{W}_{\text{inv}}^{\dagger}$ consists of infinitely many distinct functions.

Choose any $w_{\text{inv}}'$ such that $(w_{\text{inv}}', 0) \in \mathcal{W}_{\text{inv}}$ and $w_{\text{inv}}' \neq w_{\text{inv}}^*$. Define $g' = \mathsf{I} \circ \left( (w_{\text{inv}}', 0) \circ S^{-1} \right)$. We will next show that $g^* \neq g'$, where $g^*$ was defined in equation (17).

Define

$$\begin{bmatrix} w_{\text{inv}}^* \\ w_{\text{inv}}' \end{bmatrix} z_{\text{inv}} = \begin{bmatrix} 1 \\ -1 \end{bmatrix}. \tag{28}$$

There are two possibilities a) $w_{\text{inv}}'$ is not aligned with $w_{\text{inv}}^*$ in which case the rank of the matrix in the above equation (28) is two and as a result the range space of the matrix spans all two-dimensional vectors, b) $w_{\text{inv}}'$ is aligned with $w_{\text{inv}}^*$ but since $\|w_{\text{inv}}'\| = 1$, $w_{\text{inv}}' = -w_{\text{inv}}^*$ in which case $z_{\text{inv}} = w_{\text{inv}}^*$ solves the above equation (28). In both the cases the equation (28) has a solution. Let the solution of the above equation (28) be $\tilde{z}_{\text{inv}}$. Define $\tilde{x} = S \cdot (\tilde{z}_{\text{inv}}, 0)$. Therefore, from equation (28) it follows that $g^*(\tilde{x}) \neq g'(\tilde{x})$. See the simplification below for the justification.

$$\begin{aligned} g^*(\tilde{x}) &= \mathsf{I}\left( (w_{\text{inv}}^*, 0) \cdot S^{-1}(\tilde{x}) \right) = \mathsf{I}(w_{\text{inv}}^* \cdot \tilde{z}_{\text{inv}}) = 1 \\ g'(\tilde{x}) &= \mathsf{I}\left( (w_{\text{inv}}', 0) \cdot S^{-1}(\tilde{x}) \right) = \mathsf{I}(w_{\text{inv}}' \cdot \tilde{z}_{\text{inv}}) = 0 \end{aligned} \tag{29}$$

We showed above that $g^* \in \mathcal{W}_{\text{inv}}^{\dagger}$ and $g' \in \mathcal{W}_{\text{inv}}^{\dagger}$ are two distinct functions. Recall in Lemma 4, we showed $\mathcal{W}_{\text{inv}}$ has infinitely many distinct hyperplanes. We can select any pair of hyperplanes $\mathcal{W}_{\text{inv}}$, for the corresponding functions in the set $\mathcal{W}_{\text{inv}}^{\dagger}$ the condition in equation (28) continues to hold. Thus we can conclude that there are infinitely many distinct functions in $\mathcal{W}_{\text{inv}}^{\dagger}$.

Recall we described above that an algorithm can successfully solve equation (1), if and only if the output $f = g^*$. Observe that the same exact training data $\{D^e\}_{e \in \mathcal{E}_{tr}}$ can be generated by any other labelling hyperplane $w_{\text{inv}}' \neq w_{\text{inv}}^*$, where $(w_{\text{inv}}', 0) \in \mathcal{W}_{\text{inv}}$ (this follows from the definition of $\mathcal{W}_{\text{inv}}$ in equation (16)). Define $g' = \mathsf{I} \circ \left( (w', 0) \circ S^{-1} \right)$, where $g' \in \mathcal{W}_{\text{inv}}^{\dagger}$. From the justification above, we

know that $g' \neq g$. Since $g' \neq g^*$ the algorithm can only be successful on one of the two labelling hyperplanes $w_{\text{inv}}'$ or $w_{\text{inv}}^*$. In fact, since we showed that there are infinitely many possible distinct hyperplanes in $\mathcal{W}_{\text{inv}}$, the algorithm can only succeed on one of them. To summarize, the algorithm fails almost everywhere on the entire set, $\mathcal{W}_{\text{inv}}$, of equivalent generating models. This completes the proof. $\qquad\square$

**Remark on extension under partially informative invariant features, i.e., Assumption 9.** The impossibility result extends to the case when the environments follow Assumption 9. The first thing to note is that from Lemma 2, $g^*$ continues to be the OOD optimal solution hyperplane. In the above proof, we had shown the construction of how there are infinitely many possible equally good hyperplanes that could have generated the data. To arrive at those hyperplanes, we relied on Lemma 3, where we showed that there are multiple candidate hyperplanes that could have generated the same training data. In the lemma, we only exploited the separability of latent invariant features and boundedness. If we continue to assume separability and boundedness for invariant features, then the result from Lemma 3 can be used in this case as well. As a result, we can continue to use the claim that there are multiple equally good candidate hyperplanes that the algorithm cannot distinguish. Thus the impossibility result extends to this setup too.

**Remark on inveribility of $S$.** The entire proof only requires us to assume to be able to have invertibility on the latent invariant features, i.e., we should be able to recover $Z_{\text{inv}}^e$ from $X^e$. Therefore, Theorem 2 extends to matrices $S$ that are only invertible upto the $Z_{\text{inv}}^e$.

**Remark on impossibility under continuous random variable assumption.** In the proof, we showed that if the test environment $e$ places all the mass on the solution of equation (28), then the algorithm fails. In the setting, where we are only allowed to work with continuous random variables, can we continue to claim impossibility? The answer is yes. The reason is quite simple, we can instead of using the solution to equation (28) construct a small ball around that region. Since the solution to equation (28) that we constructed is in the interior of the half-spaces such an argument works.

**Remark on multi-class classification.** We describe a natural extension of the model in Assumption 2 to $k$-class classification.

**Assumption 10.** *Linear classification structural equation model (FIIF) for multi-class classification. In each $e \in \mathcal{E}_{all}$*

$$
\begin{aligned}
Y^e &\leftarrow \arg\max(W_{\text{inv}}^* \cdot Z_{\text{inv}}^e) \\
X^e &\leftarrow S(Z_{\text{inv}}^e, Z_{\text{spu}}^e),
\end{aligned}
\tag{30}
$$

*where $W_{\text{inv}}^* \in \mathbb{R}^{k \times m}$, $\arg\max$ is taken over the $k$ rows to generate the label $Y^e$, $S \in \mathbb{R}^{d \times d}$.*

We can add noise as well in the above SEM, which uniformly at random switches the class. The key geometric intuition for the impossibility result that we proved above, which was illustrated in Figure 1, carries over to this case provided the label generating hyperplane separates the supports of adjacent classes with a finite margin. Following the same geometric intuition, we can generalize the formal impossibility proof to this case as well for the SEM in Assumption 10.

### A.5 Proof of Theorem 3: sufficiency and insufficiency of ERM and IRM

**Lemma 4.** *If Assumptions 2, 4, 7 hold, then there exists a classifier which puts a non-zero weight on the spurious feature and continues to be Bayes optimal in all the training environments.*

**Proof of Lemma 4.** We will follow the construction based on Lemma 3's proof.

Choose an arbitrary non-zero vector $\gamma \in \mathbb{R}^o$. We will derive a bound on the margin of $(w_{\mathsf{inv}}^*, \gamma)$. Consider a $z_{\mathsf{inv}} \in \cup_{e \in \mathcal{E}_{tr}} \mathcal{Z}_{\mathsf{inv}}^e$ and a $z_{\mathsf{spu}} \in \cup_{e \in \mathcal{E}_{tr}} \mathcal{Z}_{\mathsf{spu}}^e$. Define $y^* = \mathsf{sgn}(w_{\mathsf{inv}}^* \cdot z_{\mathsf{inv}})$. The margin $(w_{\mathsf{inv}}^*, \gamma)$ at this point $(z_{\mathsf{inv}}, z_{\mathsf{spu}})$ with respect to $y^*$ is defined as

$$y^* \big( w_{\mathsf{inv}}^* \cdot z_{\mathsf{inv}} \big) + y^* \big( \gamma \cdot z_{\mathsf{spu}} \big). \tag{31}$$

Using Cauchy-Schwarz inequality, we get

$$|y^* \big( \gamma \cdot z_{\mathsf{spu}} \big)| = |\gamma \cdot z_{\mathsf{spu}}| \leq \|\gamma\| \|z_{\mathsf{spu}}\|. \tag{32}$$

Since the train support of spurious feature is bounded we can set the magnitude of $\gamma$ sufficiently small to control $y^* \big( \gamma \cdot z_{\mathsf{spu}} \big)$. If $\|\gamma\| \leq \frac{c}{2z^{\mathsf{sup}}}$, then $|\gamma \cdot z_{\mathsf{spu}}| \leq \frac{c}{2}$, where $z^{\mathsf{sup}}$ satisfies the following condition – for each $z \in \cup_{e \in \mathcal{E}_{tr}} \mathcal{Z}_{\mathsf{spu}}^e$ and $\|z\| \leq z^{\mathsf{sup}}$. We can use this to find a bound on the margin as follows. Recall from equation (22) we have

$$y^* \big( w_{\mathsf{inv}}^* \cdot z_{\mathsf{inv}} \big) \geq c. \tag{33}$$

We use the condition $|\gamma \cdot z_{\mathsf{spu}}| \leq \frac{c}{2}$ in the simplification below

$$y^* \big( w_{\mathsf{inv}}^* \cdot z_{\mathsf{inv}} \big) + y^* \big( \gamma \cdot z_{\mathsf{spu}} \big) \geq c - |\gamma \cdot z_{\mathsf{spu}}| \geq \frac{c}{2}. \tag{34}$$

From the above equation it follows that $\mathsf{sgn}\big((w_{\mathsf{inv}}^*, \gamma) \cdot (z_{\mathsf{inv}}, z_{\mathsf{spu}})\big) = \mathsf{sgn}\big((w_{\mathsf{inv}}^*, 0) \cdot (z_{\mathsf{inv}}, z_{\mathsf{spu}})\big) \implies \mathsf{I}\big((w_{\mathsf{inv}}^*, \gamma) \cdot (z_{\mathsf{inv}}, z_{\mathsf{spu}})\big) = \mathsf{I}\big((w_{\mathsf{inv}}^*, 0) \cdot (z_{\mathsf{inv}}, z_{\mathsf{spu}})\big)$. This condition holds for each $z_{\mathsf{inv}} \in \cup_{e \in \mathcal{E}_{tr}} \mathcal{Z}_{\mathsf{inv}}^e$ and a $z_{\mathsf{spu}} \in \cup_{e \in \mathcal{E}_{tr}} \mathcal{Z}_{\mathsf{spu}}^e$. We use this condition to compute the error of a classifier based on $(w_{\mathsf{inv}}^*, \gamma)$ below. Define $g_{\mathsf{spu}}^* = \mathsf{I} \circ (w_{\mathsf{inv}}^*, \gamma) \circ S^{-1}$. The error achieved by $g_{\mathsf{spu}}^*$ is

$$
\begin{aligned}
R^e(g_{\mathsf{spu}}^*) &= \mathbb{E}\Big[ Y^e \oplus \mathsf{I}\big((w_{\mathsf{inv}}^*, \gamma) \cdot (z_{\mathsf{inv}}, z_{\mathsf{spu}})\big) \Big] \\
&= \mathbb{E}\Big[ \mathsf{I}\big((w_{\mathsf{inv}}^*, 0) \cdot (z_{\mathsf{inv}}, z_{\mathsf{spu}})\big) \oplus N^e \oplus \mathsf{I}\big((w_{\mathsf{inv}}^*, \gamma) \cdot (z_{\mathsf{inv}}, z_{\mathsf{spu}})\big) \Big] = \mathbb{E}\big[N^e\big] = q.
\end{aligned}
\tag{35}
$$

The same calculation as above equation (35) holds in all the training environments. Thus $g_{\mathsf{spu}}^*$ achieves the minimum error possible $q$ for all the training environments $e \in \mathcal{E}_{tr}$. $\qquad \square$

We restate Theorem 3 for convenience.

**Theorem 7.** *Sufficiency and Insufficiency of ERM and IRM. Suppose each $e \in \mathcal{E}_{all}$ follows Assumption 2. Assume that a) the invariant features are strictly separable, bounded, and satisfy support overlap, b) the spurious features are bounded (Assumptions 3-5, 7 hold).*

*• Sufficiency: If the spurious features satisfy support overlap (Assumption 6 holds), then both ERM and IRM solve the OOD generalization problem (eq. (1)). Also, there exist ERM and IRM solutions that rely on the spurious features and still achieve OOD generalization.*

*• Insufficiency: If spurious features do not satisfy support overlap, then both ERM and IRM fail at solving the OOD generalization problem (eq. (1)). Also, there exist no such classifiers that rely on the spurious features and still achieve OOD generalization.*

**Proof of Theorem 7.** Let us begin with the first part of the Theorem. We first show that there exist solutions to ERM and IRM that rely on spurious features that also achieve OOD generalization (that is solve (1)). Since Assumptions 2, 4, 7, hold we can use Lemma 4. From Lemma 4, it follows that for each $z_{\mathsf{inv}} \in \cup_{e \in \mathcal{E}_{tr}} \mathcal{Z}_{\mathsf{inv}}^e$ and for each $z_{\mathsf{spu}} \in \cup_{e \in \mathcal{E}_{tr}} \mathcal{Z}_{\mathsf{inv}}^e$:

$$\mathsf{I}\big((w_{\mathsf{inv}}^*, \gamma) \cdot (z_{\mathsf{inv}}, z_{\mathsf{spu}})\big) = \mathsf{I}\big((w_{\mathsf{inv}}^*, 0) \cdot (z_{\mathsf{inv}}, z_{\mathsf{spu}})\big). \tag{36}$$

From Assumption 5 and 6 it follows that for each $z_{\mathsf{inv}} \in \cup_{e\in\mathcal{E}_{all}} \mathcal{Z}_{\mathsf{inv}}^e$ and for each $z_{\mathsf{spu}} \in \cup_{e\in\mathcal{E}_{all}} \mathcal{Z}_{\mathsf{inv}}^e$.

$$\mathsf{I}\big((w_{\mathsf{inv}}^*, \gamma) \cdot (z_{\mathsf{inv}}, z_{\mathsf{spu}})\big) = \mathsf{I}\big((w_{\mathsf{inv}}^*, 0) \cdot (z_{\mathsf{inv}}, z_{\mathsf{spu}})\big) \tag{37}$$

Therefore, the error of the classifier $g_{\mathsf{spu}}^* = \mathsf{I} \circ (w_{\mathsf{inv}}^*, \gamma) \circ S^{-1}$ in each environment $e \in \mathcal{E}_{all}$ is

$$
\begin{aligned}
R^e(g_{\mathsf{spu}}^*) &= \mathbb{E}\Big[Y^e \oplus \mathsf{I}\big((w_{\mathsf{inv}}^*, \gamma) \cdot (z_{\mathsf{inv}}, z_{\mathsf{spu}})\big)\Big] \\
&= \mathbb{E}\Big[\mathsf{I}\big((w_{\mathsf{inv}}^*, 0) \cdot (z_{\mathsf{inv}}, z_{\mathsf{spu}})\big) \oplus N^e \oplus \mathsf{I}\big((w_{\mathsf{inv}}^*, \gamma) \cdot (z_{\mathsf{inv}}, z_{\mathsf{spu}})\big)\Big] = \mathbb{E}\big[N^e\big] = q.
\end{aligned}
\tag{38}
$$

$g_{\mathsf{spu}}^*$ is Bayes optimal on each environment $e \in \mathcal{E}_{all}$. Therefore, $g_{\mathsf{spu}}^*$ also solves equation (1). Since $g_{\mathsf{spu}}^*$ is optimal in all the environments, it also solves ERM as it also minimizes the sum of risks across training environments. $g_{\mathsf{spu}}^*$ is also a valid invariant predictor since it is simultaneously optimal across all the environments. Since $g_{\mathsf{spu}}^*$ achieves an average error of $q$ across training environments, each solution to ERM and IRM has to achieve an error of $q$ in all the training environments as well. Since the solution to ERM and IRM achieves an error of $q$ it cannot differ from $g^*$ at any point in the training support. This argument holds in a pointwise sense when $Z_{\mathsf{inv}}^e$ is a discrete random variable, otherwise, say when $Z_{\mathsf{inv}}^e$ is a continuous random variable this argument can only be violated over a set of measure zero.[15] Owing to the support overlap between $\mathcal{E}_{tr}$ and $\mathcal{E}_{all}$, each solution to ERM and IRM continues to succeed in $\mathcal{E}_{all}$. This completes the first part of the proof.

We now move to the next part of the theorem, where the spurious features do not satisfy support overlap assumption (Assumption 6). Consider a linear classifier that the method learns $\mathsf{I} \circ w$, where $\mathsf{I}$ is composed with a linear function. The classifier operates on $x$, and we get $\mathsf{I}(w \cdot x)$ and since $x = Sz$ (from Assumption 2) we can write this as $\mathsf{I}(w \cdot S(z))$. To simplify notation, we call $\mathsf{I} \circ w \circ S = \mathsf{I} \circ \tilde{w}$. Our goal is to show that if $\tilde{w}$ assigns a non-zero weight to the spurious features, then $\mathsf{I} \circ w \circ S$ cannot solve the OOD generalization problem (eq. (1)). We write $\tilde{w} = (\tilde{w}_{\mathsf{inv}}, \tilde{w}_{\mathsf{spu}})$. Suppose $\tilde{w}_{\mathsf{spu}} \neq 0$ and yet the classifier solves the problem in equation (1). Consider the classifier that generates the data $(w_{\mathsf{inv}}^*, 0)$. Pick any point $z_{\mathsf{inv}} \in \cup_{e\in\mathcal{E}_{all}} \mathcal{Z}_{\mathsf{inv}}^e$ and pick any non-zero $z_{\mathsf{spu}}^e \in \mathbb{R}^o$. Call $z = (z_{\mathsf{inv}}, z_{\mathsf{spu}})$ We divide the analysis into two cases.

Case 1: $\mathsf{I}\big((\tilde{w}_{\mathsf{inv}}, \tilde{w}_{\mathsf{spu}}) \cdot z\big) \neq \mathsf{I}\big((w_{\mathsf{inv}}^*, 0) \cdot z\big)$. In this case, $(\tilde{w}_{\mathsf{inv}}, \tilde{w}_{\mathsf{spu}})$ cannot solve equation (1) as there exists a test environment where we have all the mass on $z$.

Case 2: $\mathsf{I}\big((\tilde{w}_{\mathsf{inv}}, \tilde{w}_{\mathsf{spu}}) \cdot z\big) = \mathsf{I}\big((w_{\mathsf{inv}}^*, 0) \cdot z\big)$. Observe that since $\tilde{w}_{\mathsf{spu}} \neq 0$, we can increase or decrease one of the components of $z_{\mathsf{spu}}$ corresponding to a non-zero $\tilde{w}_{\mathsf{spu}}$ until the two classifiers disagree in which case we get Case 1. Note that since Assumption 6 does not hold, we are allowed to change $z_{\mathsf{spu}}$ arbitrarily.

Thus we have established that a classifier cannot be OOD optimal if it assigns a non-zero weight to the spurious feature. As a result, the classifier from the first part $g_{\mathsf{spu}}^*$ which assigned non-zero weight to spurious features cannot be OOD optimal without the Assumption 6. However, $g_{\mathsf{spu}}^*$ continues to be in the solution space of both ERM and IRM as it is still Bayes optimal across all the train environments, which is why both ERM and IRM fail. At this point the proof of the statement of theorem is complete. However, we give a characterization of optimal solutions in the next paragraph.

Now let us consider any classifier in $w \in \mathcal{W}_{\mathsf{inv}}$ (from equation (16)) written as $w = (w_{\mathsf{inv}}, 0)$. For such a classifier by definition it is true that for each $z_{\mathsf{inv}} \in \cup_{e\in\mathcal{E}_{tr}} \mathcal{Z}_{\mathsf{inv}}^e$, $\mathsf{I}\big(w_{\mathsf{inv}} \cdot z_{\mathsf{inv}}\big) = \mathsf{I}\big(w_{\mathsf{inv}}^* \cdot z_{\mathsf{inv}}\big)$. From Assumption 5 it follows that for each $z_{\mathsf{inv}} \in \cup_{e\in\mathcal{E}_{all}} \mathcal{Z}_{\mathsf{inv}}^e$, $\mathsf{I}\big(w_{\mathsf{inv}} \cdot z_{\mathsf{inv}}\big) = \mathsf{I}\big(w_{\mathsf{inv}}^* \cdot z_{\mathsf{inv}}\big)$ and thus the classifier continues to achieve an error of $q$ on all the test environments. Thus we can conclude that $\mathsf{I} \circ w \circ S^{-1}$ is OOD optimal. Therefore, all the elements in the set $\mathcal{W}_{\mathsf{inv}}^\dagger$ (from eq. (27)) are OOD optimal.

$\square$

**Remark on invertibility of $S$.** The proof extends to the case when we can invert and recover entire $Z_{\mathsf{inv}}^e$ and also recover at least one component of the spurious features $Z_{\mathsf{spu}}^e$.

**Remark on failure of ERM and IRM under continuous random variable assumption.** In the proof, we showed that if the test environment $e$ places all the mass on the solution to Case 1, then the

---

[15]The continuous random variable case can give rise to some pathological shifts. We show later in the proof of Theorem 4 as to why we do not need to worry about these pathological shifts.

algorithm fails. In the setting, where we are only allowed to work with continuous random variables, can we continue to make the claim for impossibility? The answer is yes. The reason is quite simple, we can instead of using the solution to Case 1 construct a small ball around that region, where the classifiers continue to disagree.

**Remark on multi-class classification.** We extend the result to the above SEM in Assumption 10. The reason ERM and IRM fail in this case is two fold – a) there exists a hyperplane that perfectly separates the support of the invariant features with a finite margin and b) support of spurious features are allowed to change. In the multi-class case, we can use the same reasoning – if there is a hyperplane that perfectly separates for adjacent classes, ERM and IRM continue to fail as long as the support of spurious features is allowed to change.

## A.6  Proof of Theorem 4: IB-IRM and IB-ERM vs. IRM and ERM

We now lay down some properties of the entropy of discrete random variables and in parallel also lay down the properties of differential entropy of continuous random variables. Recall that a discrete random variable has a non-zero probability at each point in its support and a continuous random variable has a zero probability (and a positive density) at each point in the support.

The entropy or the Shannon entropy of a discrete random variable $X \sim \mathbb{P}_X$ with support $\mathcal{X}$ is defined as

$$H(X) = - \sum_{x \in \mathcal{X}} \mathbb{P}_X(X = x) \log\big(\mathbb{P}_X(X = x)\big). \tag{39}$$

The differential entropy of a continuous random variable $X \sim \mathbb{P}_X$ with support $\mathcal{X}$ is given as follows

$$h(X) = - \int_{x \in \mathcal{X}} \log\big(d\mathbb{P}_X(x)\big) d\mathbb{P}_X(x), \tag{40}$$

where $d\mathbb{P}_X(x)$ is the Radon-Nikodym derivative of $\mathbb{P}_X$ w.r.t the Lesbegue measure.

**Lemma 5.** *If $X$ and $Y$ are discrete scalar valued random variables that are independent, then*

$$H(X + Y) \geq \max\Big\{H(X), H(Y)\Big\}.$$

**Proof of Lemma 5.** Define $Z = X + Y$.

$$
\begin{aligned}
H(Z|X) &= - \sum_{x \in \mathcal{X}} \mathbb{P}_X(x) \sum_{z \in \mathcal{Z}} \mathbb{P}_{Z|X}(Z = z|X = x) \log\Big(\mathbb{P}_{Z|X}(Z = z|X = x)\Big) \\
&= - \sum_{x \in \mathcal{X}} \mathbb{P}_X(x) \sum_{z \in \mathcal{Z}} \mathbb{P}_{Y|X}(Y = z - x|X = x) \log\Big(\mathbb{P}_{Y|X}(Y = z - x|X = x)\Big) \\
&= - \sum_{x \in \mathcal{X}} \mathbb{P}_X(x) \sum_{z \in \mathcal{Z}} \mathbb{P}_{Y|X}(Y = z - x|X = x) \log\Big(\mathbb{P}_{Y|X}(Y = z - x|X = x)\Big) \text{ (use } X \perp Y) \\
&= - \sum_{x \in \mathcal{X}} \mathbb{P}_X(x) \sum_{z \in \mathcal{Z}} \mathbb{P}_Y(Y = z - x) \log\Big(\mathbb{P}_Y(Y = z - x)\Big) \\
&= H(Y)
\end{aligned}
\tag{41}
$$

$$
\begin{aligned}
I(Z; X) &= H(Z) - H(Z|X) = H(X + Y) - H(Y) \\
I(Z; Y) &= H(Z) - H(Z|Y) = H(X + Y) - H(X)
\end{aligned}
\tag{42}
$$

From equation (42) and the property of mutual information that $I(Z; X) \geq 0, I(Z; Y) \geq 0$ it follows that

$$H(X + Y) \geq H(Y), \; H(X + Y) \geq H(X) \implies H(X + Y) \geq \max\{H(X), H(Y)\}. \tag{43}$$

This completes the proof.  $\square$

**Lemma 6.** *If $X$ and $Y$ are continuous scalar valued random variables that are independent, then*

$$h(X + Y) \geq \max\Big\{h(X), h(Y)\Big\}.$$

**Proof of Lemma 6.** Define $Z = X + Y$.

$$
\begin{aligned}
h(Z|X) &= \mathbb{E}_{\mathbb{P}_X}\Big[\mathbb{E}_{\mathbb{P}_{Z|X}}\Big[\log\Big(d\mathbb{P}_{Z|X}(Z = z|X = x)\Big)\Big]\Big] \\
&= \mathbb{E}_{\mathbb{P}_X}\Big[\mathbb{E}_{\mathbb{P}_{Y|X}}\Big[\log\Big(d\mathbb{P}_{Y|X}(Y = z - x|X = x)\Big)\Big]\Big] \text{ (use } X \perp Y) \\
&= h(Y)
\end{aligned}
\tag{44}
$$

Note that $I(Z;X) \geq 0 \implies h(Z) \geq h(Z|X)$. Combining this with the above equation (44) we get
$$h(X + Y) \geq h(Y). \tag{45}$$
From symmetry it follows that $h(X + Y) \geq h(X)$. This completes the proof. $\qquad\square$

**Lemma 7.** *If $X$ and $Y$ are discrete scalar valued random variables that are independent with the supports satisfying $2 \leq |\mathcal{X}| < \infty$, $2 \leq |\mathcal{Y}| < \infty$, then*
$$H(X + Y) > \max\Big\{ H(X), H(Y) \Big\}.$$

**Proof of Lemma 7.** Suppose $|\mathcal{X}| = \{x_{\min}, \ldots, x_{\max}\}$ and $\mathcal{Y} = \{y_{\min}, \ldots, y_{\max}\}$. The smallest value of $X + Y$ is $x_{\min} + y_{\min}$ and the largest value is $x_{\max} + y_{\max}$. Suppose that the inequality in the claim is not true in which case from Lemma 5 it follows $H(X+Y) = H(X)$ or $H(X+Y) = H(Y)$. Suppose $H(X + Y) = H(X)$, then from equation (42) it follows that $I(X + Y; Y) = 0 \implies X + Y \perp Y$. Observe that if $Z = x_{\max} + y_{\max} \implies Y = y_{\max}$. Therefore, $\mathbb{P}(Y = y_{\max}|Z = x_{\max} + y_{\max}) = 1$. However, $\mathbb{P}(Y = y_{\max}) \neq 1$ as the support of $Y$ has at least two elements. This contradicts $X + Y \perp Y$. As a result, $H(X+Y) \neq H(X)$. We can symmetrically show that $H(X+Y) \neq H(Y)$. Combining this with Lemma 5, it follows that $H(X + Y) > \max\{H(X), H(Y)\}$. $\qquad\square$

**Lemma 8.** *If $X$ and $Y$ are continuous scalar valued random variables that are independent and have a bounded support, then*
$$h(X + Y) > \max\Big\{ h(X), h(Y) \Big\}$$

**Proof of Lemma 8.** The steps of the proof are similar to Lemma 7. Suppose the inequality in the claim is not true in which case from Lemma 6 it follows that either $h(X+Y) = h(X)$ or $h(X+Y) = h(Y)$. Suppose $h(X + Y) = h(X)$ which implies $I(X + Y; Y) = 0 \implies X + Y \perp Y$. The support of $X$ can be written in the form of union of intervals. Suppose we consider the rightmost interval and we write it as $[x_{\max} - \Delta, x_{\max}]$. Similarly for $Y$, we write the rightmost interval as $[y_{\max} - \Delta, y_{\max}]$. [16] Define an event $\mathcal{M} : x_{\max} + y_{\max} - \delta \leq X + Y \leq x_{\max} + y_{\max}$. If $\mathcal{M}$ occurs, then $Y \geq y_{\max} - \delta$ and $X \geq x_{\max} - \delta$.
$$\begin{aligned} \mathbb{P}_X(X \leq x_{\max} - \delta|\mathcal{M}) = 0 \\ \mathbb{P}_Y(Y \leq y_{\max} - \delta|\mathcal{M}) = 0 \end{aligned} \tag{46}$$
If $\delta < \Delta$ we know that
$$\begin{aligned} \mathbb{P}_X(X \leq x_{\max} - \delta) > 0 \\ \mathbb{P}_Y(Y \leq y_{\max} - \delta) > 0 \end{aligned} \tag{47}$$
If $X + Y \perp Y$ then $\mathbb{P}_Y(Y \leq y_{\max} - \delta) = \mathbb{P}_Y(Y \leq y_{\max} - \delta|\mathcal{M})$, which is not the case from the above equations (46) and (47). Thus $X + Y \not\perp Y \implies I(X+Y; Y) > 0 \implies h(X+Y) > h(X)$. We can say the same for $Y$ and conclude that $h(X + Y) > h(Y)$. This completes the proof. $\qquad\square$

Theorem 4 has two versions – one for discrete random variables, and the other for continuous random variables. We discuss the discrete random variable case first as its easier to understand and then move to the continuous random variable case.

### A.6.1  Discrete random variables

In this section, we assume that in each $e \in \mathcal{E}_{all}$, random variables $Z_{\mathsf{inv}}^e$, $Z_{\mathsf{spu}}^e$, $N^e, W^e$ in Assumption 8 are discrete. We formulate the optimization in terms of Shannon entropy as follows.

$$\begin{aligned} &\min_{w \in \mathbb{R}^{k \times r}, \Phi \in \mathbb{R}^{r \times d}} \frac{1}{|\mathcal{E}_{tr}|} \sum_e H^e\big(w \cdot \Phi\big) \\ &\text{s.t.} \quad \frac{1}{|\mathcal{E}_{tr}|} \sum_e R^e\big(w \cdot \Phi\big) \leq r^* \\ &\qquad w \in \arg\min_{\tilde{w} \in \mathbb{R}^{k \times r}} R^e(\tilde{w} \cdot \Phi) \end{aligned} \tag{48}$$

Note that the only difference between equation (48) and the equation (6) is that the objective here is Shannnon entropy, while the objective in the other case is the differential entropy.

---

[16] We use same $\Delta$ for both $X$ and $Y$ because can take the smaller of the rightmost intervals for $X$ and $Y$.

**Theorem 8.** *IB-IRM and IB-ERM vs IRM and ERM*

***Fully informative invariant features (FIIF).*** *Suppose each $e \in \mathcal{E}_{all}$ follows Assumption 2. Assume that the invariant features are strictly separable, bounded, and satisfy support overlap (Assumptions 3,5 and 7 hold). Also, for each $e \in \mathcal{E}_{tr}$ $Z_{spu}^e \leftarrow AZ_{inv}^e + W^e$, where $A \in \mathbb{R}^{o \times m}$, $W^e \in \mathbb{R}^o$ is discrete, bounded noise, with zero mean (and each component takes at least two distinct values). Each solution to IB-IRM (eq. (6), with $\ell$ as 0-1 loss, and $r^{th} = q$), and IB-ERM solves the OOD generalization (eq. (1)) but ERM and IRM (eq.(3)) fail.*

In the above Theorem 8, we only state the first part of the Theorem 4, the reason is that the proof of the second part proof is exactly the same in both discrete and continuous random variable case and we describe the proof for the second part in the continuous random variable section next.

**Proof of Theorem 8.** First, let us discuss why IRM and ERM fail in the above setting. We argue that the failure, in this case, follows directly from the second part of Theorem 3. To directly use the second part of Theorem 3, we need Assumptions 2-5 and 7 to hold. In the statement of the above theorem, Assumption 2, 3, 5, and 7 already hold. We are only required to show that Assumption 4 holds. Since $Z_{inv}^e$ and $W^e$ are bounded on training environments we can argue that $Z_{spu}^e$ is also bounded in training environments ($\|Z_{spu}^e\| \leq \|A\|\|Z_{inv}^e\| + \|W^e\|$). We can now directly use the second part of Theorem 3 because Assumptions 2-5 and 7 hold. Since Assumption 6 is not required to hold, both ERM and IRM will fail as their solution space continue to contain classifiers that rely on spurious features. To further elaborate on why ERM and IRM fail, recall that in the second part of Theorem 3, we relied on Lemma 4. In Lemma 4, we had shown that if latent invariant features are strictly separable, and latent spurious features are bounded, then there exist classifiers that rely on spurious features and yet are Bayes optimal on all the training environments. In this case, we have latent invariant features that are strictly separable and spurious features that are bounded, which is why we can use Theorem 3. We now move to the part, where we establish why IB-IRM and IB-ERM succeed.

Consider a solution to equation (48) and call it $\Phi^\dagger$. Consider the prediction made by this model

$$\Phi^\dagger \cdot X^e = \Phi^\dagger \cdot S(Z_{inv}^e, Z_{spu}^e) = \Phi_{inv} \cdot Z_{inv}^e + \Phi_{spu} \cdot Z_{spu}^e. \tag{49}$$

We first show that $\Phi_{spu}$ is zero. We prove this by contradiction. Assume $\Phi_{spu} \neq 0$ and use the condition in the theorem to simplify the expression for the prediction as follows

$$\begin{aligned}
&\Phi_{inv} \cdot Z_{inv}^e + \Phi_{spu} \cdot Z_{spu}^e \\
&= \Phi_{inv} \cdot Z_{inv}^e + \Phi_{spu} \cdot (AZ_{inv}^e + W^e) \\
&= \Phi_{inv} \cdot Z_{inv}^e + \Phi_{spu} \cdot (AZ_{inv}^e + W^e) \\
&= \Big[\Phi_{inv} + \Phi_{spu} \cdot A\Big] \cdot Z_{inv}^e + \Phi_{spu} \cdot W^e.
\end{aligned} \tag{50}$$

We will show that $\Phi^+ = \left(\Big[\Phi_{inv} + \Phi_{spu} \cdot A\Big], 0\right)S^{-1} = \Big[\Phi_{inv} + \Phi_{spu} \cdot A\Big]S_{inv}^\dagger$, where $S_{inv}^\dagger$ corresponds to the first $m$ rows of the matrix $S^{-1}$, can continue to achieve an error of $q$ and has a lower entropy than $\Phi^\dagger$. Recall that $\Phi^\dagger$ achieves an average error across the training environments of $q$ (because $r^{th} = q$ the average cannot fall below $q$ as in that case at least one environment would have a lower error than $q$ which is not possible), which implies each environment also achieves an error of $q$.

Consider an environment $e \in \mathcal{E}_{tr}$. Since the error $\Phi^\dagger$ is $q$ it implies that for each training environment $e$

$$\mathsf{I}(w_{inv}^* \cdot Z_{inv}^e) = \mathsf{I}(\Phi_{inv} \cdot Z_{inv}^e + \Phi_{spu} \cdot Z_{spu}^e) \tag{51}$$

holds over all the points in the support of environment $e$. Suppose the above claim was not true, i.e. suppose the set $\mathsf{I}(w_{inv}^* \cdot Z_{inv}^e) \neq \mathsf{I}(\Phi_{inv} \cdot Z_{inv}^e + \Phi_{spu} \cdot Z_{spu}^e)$ occurs with a for some point in the support (suppose that point occurs with probability $\theta$). Let us compute the error

$$\begin{aligned}
R^e(\Phi^\dagger) &= \mathbb{E}\Big[\big(\mathsf{I}(w_{inv}^* \cdot Z_{inv}^e) \oplus N^e \oplus \mathsf{I}(\Phi_{inv} \cdot Z_{inv}^e + \Phi_{spu} \cdot Z_{spu}^e)\big)\Big] \\
&= \theta\mathbb{E}[1 \oplus N^e] + (1-\theta)\mathbb{E}[N^e] > q
\end{aligned} \tag{52}$$

If the above is true, then that contradicts the claim that $\Phi^\dagger$ achieves an error of $q$. Thus the statement in equation (51) has to hold at all points in the training support of the invariant features. Let $\mathcal{W}^e$ be the support of $W^e$. In each training environment, if we consider a $z^e_{\mathsf{inv}} \in \mathcal{Z}^e_{\mathsf{inv}}$, then $\forall w^e \in \mathcal{W}^e$, the following holds – if $\mathsf{I}(w^*_{\mathsf{inv}} \cdot z^e_{\mathsf{inv}}) = 1$, then

$$
\begin{aligned}
&\Phi_{\mathsf{inv}} \cdot z^e_{\mathsf{inv}} + \Phi_{\mathsf{spu}} \cdot (A z^e_{\mathsf{inv}} + w^e) \geq 0 \\
\Longrightarrow\ &\Phi_{\mathsf{inv}} \cdot z^e_{\mathsf{inv}} + \Phi_{\mathsf{spu}} \cdot (A z^e_{\mathsf{inv}}) \geq -\Phi_{\mathsf{spu}} \cdot w^e \\
\Longrightarrow\ &\left(\Phi_{\mathsf{inv}} + \Phi_{\mathsf{spu}} \cdot A\right) \cdot z^e_{\mathsf{inv}} \geq \max_{w^e \in \tilde{\mathcal{W}}^e} -\Phi_{\mathsf{spu}} \cdot w^e \\
\Longrightarrow\ &\left(\Phi_{\mathsf{inv}} + \Phi_{\mathsf{spu}} \cdot A\right) \cdot z^e_{\mathsf{inv}} \geq 0 \\
\Longrightarrow\ &\Phi^+ X^e \geq 0.
\end{aligned}
\tag{53}
$$

Similarly, we can argue that if $\mathsf{I}(w^*_{\mathsf{inv}} \cdot z^e_{\mathsf{inv}}) = 0$, then

$$
\begin{aligned}
&\left(\Phi_{\mathsf{inv}} + \Phi_{\mathsf{spu}} \cdot A\right) \cdot z^e_{\mathsf{inv}} < 0 \\
&\Phi^+ X^e < 0.
\end{aligned}
\tag{54}
$$

In the above simplification equation (53), we use $\max_{w^e} -\Phi_{\mathsf{spu}} \cdot w^e \geq 0$. Consider any component of $-\Phi_{\mathsf{spu}}$; if the sign of the component is positive (negative), then set the corresponding component of $w^e$ to be positive (negative). As a result, $-\Phi_{\mathsf{spu}} \cdot w^e \geq 0$. In this argument, we only relied on the assumption that $w^e$ can take both signs in the set $\mathcal{W}^e$. Suppose $\mathcal{W}^e$ had either positive or negative values only then this would imply that the mean of $w^e$ is strictly positive or negative, which cannot be true because $W^e$ is zero mean. From equation (53) and (54), we can conclude that $\Phi^+$ achieves the same error of $q$ in all the training environments.

Observe that we can write $\Phi^\dagger \cdot X^e = \Phi^+ \cdot X^e + \Phi_{\mathsf{spu}} \cdot W^e$. We state two properties that we use to show that entropy $\Phi^+$ is smaller than $\Phi^\dagger$:

a) $\Phi_{\mathsf{spu}} \cdot W^e \perp \Phi^+ \cdot X^e$ ($\Phi^+ \cdot X^e = \left[\Phi_{\mathsf{inv}} + \Phi_{\mathsf{spu}} \cdot A\right] \cdot Z^e_{\mathsf{inv}}$ and $Z^e_{\mathsf{inv}} \perp W^e$),

b) $\Phi^+ \cdot X, \Phi_{\mathsf{spu}} \cdot W^e$ are discrete random variables with finite support of size at least two.

We justify why b) is true in the above. $\Phi^+ \cdot X^e$ is a bounded random variable ($Z^e_{\mathsf{spu}}$ is bounded as $Z^e_{\mathsf{inv}}$ and $W^e$ are bounded. Thus $X^e$ is also bounded). $\Phi^+ \cdot X^e$ has at least two elements in its support this follows from equation (53) and (54). $\Phi_{\mathsf{spu}} \cdot W^e$ is bounded since $W^e$ is bounded and takes at least two values because each component of $W^e$ takes at least two distinct values.

From a), b), and Lemma 7 it follows that $\Phi^+ \cdot X^e$ is a classifier with lower entropy. We already established that $\Phi^+$ achieves the same error as $\Phi^\dagger$ for all the training environments. $\Phi^+$ achieves an error of $q$ for all the training environments simultaneously. Since $q$ is the smallest value for the error that is achievable, the invariance constraint in equation (71) is automatically satisfied. Therefore, $\Phi^+$ is strictly preferable to $\Phi^\dagger$. Thus the solution $\Phi^\dagger$ cannot rely on the spurious features and $\Phi_{\mathsf{spu}} = 0$.

Thus any solution $\Phi^\dagger$ to equation (48) has to satisfy $\Phi^\dagger \cdot S = (\Phi_{\mathsf{inv}}, 0)$ and $\Phi^\dagger \cdot S$ also satisfies

$$
\mathsf{I}(w^*_{\mathsf{inv}} \cdot Z^e_{\mathsf{inv}}) = \mathsf{I}(\Phi_{\mathsf{inv}} \cdot Z^e_{\mathsf{inv}}).
\tag{55}
$$

Recall that in the second part of Theorem 3's proof we showed that if a solution does not rely on spurious features and satisfies equation (65) for all the points in the support, then under the support overlap assumptions such a solution is OOD optimal as well. Since we assume support overlap assumption holds for the invariant features, we use the same argument from the second part of Theorem 3 and it follows that the solution to equation (48) also solves equation (1). $\square$

### A.6.2 Continuous random variables

In this section, we assume that in each $e \in \mathcal{E}_{all}$, the random variables $Z^e_{\mathsf{inv}}, Z^e_{\mathsf{spu}}, N^e, W^e$ in Assumption 2 are continuous.

**Lower bounding the differential entropy objective:** In general, the differential entropy can be unbounded below. Following the work of Kirsch et al. (2020), we add an independent noise term to

the predictor to ensure that the entropy is lower bounded. Suppose $w \cdot \Phi$ is the output of the predictor and the entropy of the predictor for the data in environment $e$ as $h^e(w \cdot \Phi)$. Consider a prediction made by the classifier $w \cdot \Phi(X^e)$; we add noise $\kappa^e$ (continuous, bounded random variable with a finite entropy) to this prediction to get $w \cdot \Phi(X) + \kappa^e$. The differential entropy after noise addition as $h^e(w \cdot \Phi(X^e) + \kappa^e)$. Observe that $h^e(w \cdot \Phi(X^e) + \kappa^e) \geq h(\kappa^e)$. In the rest of the discussion, we just write $h^e(w \cdot \Phi(X^e) + \kappa^e)$ as $h^e(w \cdot \Phi)$ to make the notation less cumbersome. We constrain $\mathcal{H}_\Phi$ ($\mathcal{H}_w$) in the optimization in equation (6) to a set $\tilde{\mathcal{H}}_\Phi = \{\Phi \in \mathbb{R}^{r \times d} \mid 0 < \phi_{\mathsf{inf}} \leq \|\Phi\| \leq \phi_{\mathsf{sup}}\}$ ($\tilde{\mathcal{H}}_w = \{w \in \mathbb{R}^{k \times r} \mid 0 < w_{\mathsf{inf}} \leq \|w\| \leq w_{\mathsf{sup}}\}$) instead of $\mathcal{H}_\Phi = \mathbb{R}^{r \times d}$ ($\mathcal{H}_w = \mathbb{R}^{k \times r}$). The reason to do this is that while the 0-1 loss does not change with scaling of the predictor but the entropy can change a lot. The lower bound on the norm of the classifier ensures that the optimization does not shrink it to zero in trying to minimize the entropy. We restate the optimization in equation (6) after accounting for the pathologies of differential entropy that we described above:

$$\min_{w \in \tilde{\mathcal{H}}_w, \Phi \in \tilde{\mathcal{H}}_\Phi} \frac{1}{|\mathcal{E}_{tr}|} \sum_e h^e(w \cdot \Phi)$$

$$\text{s.t.} \quad \frac{1}{|\mathcal{E}_{tr}|} \sum_e R^e(w \cdot \Phi) \leq r^{\mathsf{th}} \tag{56}$$

$$w \in \arg \min_{\tilde{w} \in \tilde{\mathcal{H}}_w} R^e(\tilde{w} \cdot \Phi)$$

We restate Theorem 4 for convenience.

### Theorem 9. *IB-IRM and IB-ERM vs IRM and ERM*

• *Fully informative invariant features (FIIF).* *Suppose each $e \in \mathcal{E}_{all}$ follows Assumption 2. Assume that the invariant features are strictly separable, bounded, and satisfy support overlap (Assumptions 3, 5 and 7 hold). Also, for each $e \in \mathcal{E}_{tr}$ $Z^e_{\mathsf{spu}} \leftarrow AZ^e_{\mathsf{inv}} + W^e$, where $A \in \mathbb{R}^{o \times m}$, $W^e \in \mathbb{R}^o$ is continuous, bounded, and zero mean noise. Each solution to IB-IRM (eq. (6), with $\ell$ as 0-1 loss, and $r^{\mathsf{th}} = q$), and IB-ERM solves the OOD generalization (eq. (1)) but ERM and IRM (eq.(3)) fail.*

• *Partially informative invariant features (PIIF).* *Suppose each $e \in \mathcal{E}_{all}$ follows Assumption 1 and $\exists \, e \in \mathcal{E}_{tr}$ such that $\mathbb{E}[\epsilon^e Z^e_{\mathsf{spu}}] \neq 0$. If $|\mathcal{E}_{tr}| > 2d$ and the set $\mathcal{E}_{tr}$ lies in a linear general position (a mild condition defined in the Appendix), then each solution to IB-IRM (eq. (6), with $\ell$ as square loss, $\sigma^2_\epsilon < r^{\mathsf{th}} \leq \sigma^2_Y$, where $\sigma^2_Y$ and $\sigma^2_\epsilon$ are the variance in the label and noise across $\mathcal{E}_{tr}$) and IRM (eq.(3)) solves OOD generalization (eq. (1)) but IB-ERM and ERM fail.*

**Proof of Theorem 9.** First, let us discuss why IRM and ERM fail in the above setting. We argue that the failure, in this case, follows directly from the second part of Theorem 3. To directly use the second part of Theorem 3, we need Assumptions 2-5 and 7 to hold. In the statement of the above theorem, Assumption 2, 3, 5, and 7 already hold. We are only required to show that Assumption 4 holds. Since $Z^e_{\mathsf{inv}}$ and $W^e$ are bounded on training environments we can argue that $Z^e_{\mathsf{spu}}$ is also bounded in training environments ($\|Z^e_{\mathsf{spu}}\| \leq \|A\|\|Z^e_{\mathsf{inv}}\| + \|W^e\|$). We can now directly use the second part of Theorem 3 because Assumptions 2-5 and 7 hold. Since Assumption 6 is not required to hold, both ERM and IRM will fail as their solution space continue to contain classifiers that rely on spurious features. [17]

Consider a solution to IB-IRM (eq. (56)) and call it $\Phi^\dagger$. Consider the prediction made by this model

$$\Phi^\dagger \cdot X^e = \Phi^\dagger \cdot S(Z^e_{\mathsf{inv}}, Z^e_{\mathsf{spu}}) = \Phi_{\mathsf{inv}} \cdot Z^e_{\mathsf{inv}} + \Phi_{\mathsf{spu}} \cdot Z^e_{\mathsf{spu}}. \tag{57}$$

We first show that $\Phi_{\mathsf{spu}}$ is zero. We prove this by contradiction. Assume $\Phi_{\mathsf{spu}} \neq 0$ and use the condition in the theorem to simplify the expression for the prediction as follows.

$$\begin{aligned}
&\Phi_{\mathsf{inv}} \cdot Z^e_{\mathsf{inv}} + \Phi_{\mathsf{spu}} \cdot Z^e_{\mathsf{spu}} \\
&= \Phi_{\mathsf{inv}} \cdot Z^e_{\mathsf{inv}} + \Phi_{\mathsf{spu}} \cdot (AZ^e_{\mathsf{inv}} + W^e) \\
&= \Phi_{\mathsf{inv}} \cdot Z^e_{\mathsf{inv}} + \Phi_{\mathsf{spu}} \cdot (AZ^e_{\mathsf{inv}} + W^e) \\
&= \Big[\Phi_{\mathsf{inv}} + \Phi_{\mathsf{spu}} \cdot A\Big] \cdot Z^e_{\mathsf{inv}} + \Phi_{\mathsf{spu}} \cdot W^e.
\end{aligned} \tag{58}$$

---

[17] In the remark following the proof of Theorem 3, we had discussed the failure of ERM and IRM continues to hold even when we are restricted to use continuous random variables.

We will show that $\Phi^+ = \left(\left[\Phi_{\mathsf{inv}} + \Phi_{\mathsf{spu}} \cdot A\right], 0\right) S^{-1} = \left[\Phi_{\mathsf{inv}} + \Phi_{\mathsf{spu}} \cdot A\right] S_{\mathsf{inv}}^\dagger$, where $S_{\mathsf{inv}}^\dagger$ corresponds to the first $m$ rows of the matrix $S^{-1}$, can continue to achieve an error of $q$ and has a lower entropy than $\Phi^\dagger$. Recall that $\Phi^\dagger$ achieves an average error across the training environments of $q$ (because $r^{\mathsf{th}} = q$ the average cannot fall below $q$ as in that case at least one environment would have a lower error than $q$ which is not possible), which implies each environment also achieves an error of $q$.

Consider an environment $e \in \mathcal{E}_{tr}$. Since the error $\Phi^\dagger$ is $q$ it implies that for each training environment

$$\mathsf{I}(w_{\mathsf{inv}}^* \cdot Z_{\mathsf{inv}}^e) = \mathsf{I}(\Phi_{\mathsf{inv}} \cdot Z_{\mathsf{inv}}^e + \Phi_{\mathsf{spu}} \cdot Z_{\mathsf{spu}}^e), \tag{59}$$

holds with probability 1. Suppose the above claim was not true, i.e. suppose the set $\mathsf{I}(w_{\mathsf{inv}}^* \cdot Z_{\mathsf{inv}}^e) \neq \mathsf{I}(\Phi_{\mathsf{inv}} \cdot Z_{\mathsf{inv}}^e + \Phi_{\mathsf{spu}} \cdot Z_{\mathsf{spu}}^e)$ occurs with a non-zero probability say $\theta$. Let us compute the error

$$\begin{aligned}
R^e(\Phi^\dagger) &= \mathbb{E}\left[\left(\mathsf{I}(w_{\mathsf{inv}}^* \cdot Z_{\mathsf{inv}}^e) \oplus N^e \oplus \mathsf{I}(\Phi_{\mathsf{inv}} \cdot Z_{\mathsf{inv}}^e + \Phi_{\mathsf{spu}} \cdot Z_{\mathsf{spu}}^e)\right)\right] \\
&= \theta\mathbb{E}[1 \oplus N^e] + (1-\theta)\mathbb{E}[N^e] > q
\end{aligned} \tag{60}$$

If the above is true, then that contradicts the claim that $\Phi^\dagger$ achieves an error of $q$. Thus the statement in equation (59) has to hold with probability 1. Let $\mathcal{W}^e$ denote the support of $W^e$ in environment $e$. We can restate the above observation as – there exists sets $\tilde{\mathcal{Z}}_{\mathsf{inv}}^e \subseteq \mathcal{Z}_{\mathsf{inv}}^e$ and a set $\tilde{\mathcal{W}}^e \subseteq \mathcal{W}^e$ such that $\mathbb{P}(\tilde{\mathcal{Z}}_{\mathsf{inv}}^e \times \tilde{\mathcal{W}}^e) = 1$ [18] and for each element in $\tilde{\mathcal{Z}}_{\mathsf{inv}}^e \times \tilde{\mathcal{W}}^e$

$$\mathsf{I}(w_{\mathsf{inv}}^* \cdot Z_{\mathsf{inv}}^e) = \mathsf{I}(\Phi_{\mathsf{inv}} \cdot Z_{\mathsf{inv}}^e + \Phi_{\mathsf{spu}} \cdot Z_{\mathsf{spu}}^e) \tag{61}$$

Consider a training environment $e \in \mathcal{E}_{tr}$. For each $z_{\mathsf{inv}}^e \in \tilde{\mathcal{Z}}_{\mathsf{inv}}^e$, the following conditions hold $\forall w^e \in \tilde{\mathcal{W}}^e$ – if $\mathsf{I}(w_{\mathsf{inv}}^* \cdot z_{\mathsf{inv}}^e) = 1$, then

$$\begin{aligned}
&\Phi_{\mathsf{inv}} \cdot z_{\mathsf{inv}}^e + \Phi_{\mathsf{spu}} \cdot (Az_{\mathsf{inv}}^e + w^e) \geq 0 \\
\implies &\Phi_{\mathsf{inv}} \cdot z_{\mathsf{inv}}^e + \Phi_{\mathsf{spu}} \cdot (Az_{\mathsf{inv}}^e) \geq -\Phi_{\mathsf{spu}} \cdot w^e \\
\implies &\left(\Phi_{\mathsf{inv}} + \Phi_{\mathsf{spu}} \cdot A\right) \cdot z_{\mathsf{inv}}^e \geq \max_{w^e \in \tilde{\mathcal{W}}^e} -\Phi_{\mathsf{spu}} \cdot w^e \\
\implies &\left(\Phi_{\mathsf{inv}} + \Phi_{\mathsf{spu}} \cdot A\right) \cdot z_{\mathsf{inv}}^e \geq 0 \\
\implies &\Phi^+ X^e \geq 0.
\end{aligned} \tag{62}$$

Similarly, we can argue that if $\mathsf{I}(w_{\mathsf{inv}}^* \cdot z_{\mathsf{inv}}^e) = 0$, then

$$\begin{aligned}
&\left(\Phi_{\mathsf{inv}} + \Phi_{\mathsf{spu}} \cdot A\right) \cdot z_{\mathsf{inv}}^e < 0 \\
&\Phi^+ X^e < 0.
\end{aligned} \tag{63}$$

In the above simplification in equation (62), we use $\max_{w^e} -\Phi_{\mathsf{spu}} \cdot w^e \geq 0$. Consider any component of $-\Phi_{\mathsf{spu}}$; if the sign of the component is positive (negative), then set the corresponding component of $w^e$ to be positive (negative). As a result, $-\Phi_{\mathsf{spu}} \cdot w^e \geq 0$. In this argument, we only relied on the assumption that $w^e$ can take both signs in the set $\tilde{\mathcal{W}}^e$. Suppose $w^e$ can only take either positive or negative values in $\tilde{\mathcal{W}}^e$ this would imply that the mean of $w^e$ is strictly positive or negative, which cannot be true because $W^e$ is zero mean. From equation (62), (63), and $\mathbb{P}(\tilde{\mathcal{Z}}_{\mathsf{inv}}^e \times \tilde{\mathcal{W}}^e) = 1$, we can conclude that $\Phi^+$ achieves the same error of $q$ in all the training environments.

Observe that we can write $\Phi^\dagger \cdot X^e = \Phi^+ \cdot X^e + \Phi_{\mathsf{spu}} \cdot W^e$. We state two properties that we use to show that entropy $\Phi^+$ is smaller than $\Phi^\dagger$:

a) $\Phi_{\mathsf{spu}} \cdot W^e \perp \Phi^+ \cdot X^e$ ($\Phi^+ \cdot X^e = \left[\Phi_{\mathsf{inv}} + \Phi_{\mathsf{spu}} \cdot A\right] \cdot Z_{\mathsf{inv}}^e$ and $Z_{\mathsf{inv}}^e \perp W^e$),

b) $\Phi_{\mathsf{inv}}^+ \cdot X, \Phi_{\mathsf{spu}} \cdot W^e$ are continuous bounded random variables,

We justify why b) is true in the above. $\Phi_{\mathsf{inv}}^+ \cdot X^e$ is a bounded random variable ($Z_{\mathsf{spu}}^e$ is bounded as $Z_{\mathsf{inv}}^e$ is bounded and as a result $X^e$ is bounded as well). Observe that $\Phi_{\mathsf{inv}}^+ \neq 0$, this follows

---

[18]Owing to the independence of the noise we also have $\mathbb{P}(\tilde{\mathcal{Z}}_{\mathsf{inv}}^e) = 1, \mathbb{P}(\tilde{\mathcal{W}}^e) = 1$.

from equation (62) and (63). $\Phi_{\mathsf{inv}}^{+} \cdot X^e$ is a continuous random variable as well. Suppose $\Phi_{\mathsf{inv}}^{+} \cdot X^e$ was not continuous, which implies for some constant $b$, $\Phi_{\mathsf{inv}}^{+} \cdot X^e = b$ with a finite probability. If $\Phi_{\mathsf{inv}}^{+} \cdot X^e = b$ with a finite probability, then $X$ cannot be a continuous random vector (as there exists a hyperplane which occurs with a non-zero probability).

From a), b), and Lemma 8 it follows that

$$h^e(\Phi^{+} \cdot X^e) < h^e(\Phi^{\dagger} \cdot X^e) \tag{64}$$

Note that the above equation (64) is true independent of whether we added a bounded noise to keep the entropy bounded from below. Therefore, so far we have established that $\Phi^{+}$ is a classifier with lower entropy and the same error as $\Phi^{\dagger}$. Observe that $\Phi^{+}$ achieves an error of $q$ for all the training environments simultaneously. Since $q$ is the smallest value for the error that is achievable, the invariance constraint in equation (71) is automatically satisfied with $\Phi^{\dagger}$ as the classifier and the representation as the identity. Thus $\Phi^{+}$ is a strictly preferable solution $\Phi^{\dagger}$, which contradicts the optimality of $\Phi^{\dagger}$. Therefore, it follows that $\Phi_{\mathsf{spu}} = 0$

Thus any solution $\Phi^{\dagger}$ to equation (56) has to satisfy $\Phi^{\dagger} \cdot S = (\Phi_{\mathsf{inv}}, 0)$ and $\Phi^{\dagger} \cdot S$ also satisfies

$$\mathsf{I}(w_{\mathsf{inv}}^{*} \cdot Z_{\mathsf{inv}}^e) = \mathsf{I}(\Phi_{\mathsf{inv}} \cdot Z_{\mathsf{inv}}^e) \tag{65}$$

with probability one. From the second part of Theorem 3's proof we know if a solution satisfies two properties a) does not rely on spurious features, and b) satisfies equation (65) for all the points in the support, then under the support overlap of invariant features such a solution is OOD optimal (solves equation (1)) as well. In this case, we have also assumed support overlap assumption holds for the invariant features. We have established that the solution does not rely on spurious features. Also, we have shown that equation (65) holds not pointwise but with probability one. We can still use the same argument from the second part of Theorem 3 and it follows that the solution to equation (56) also solves equation (1). Next, we show why it suffices for the equation (65) to hold with probability one.

Since the equation (65) does not hold pointwise at all the points in the support and can be violated over a set of probability zero we need to be careful about some pathological shifts at test time that place a finite mass in the region where equation (1) is violated. We now argue using arguments based on standard measure theory (Ash and Doléans-Dade, 2000) that such pathological shifts cannot occur under the assumptions made in this setting.

Recall that we defined $\tilde{\mathcal{Z}}_{\mathsf{inv}}^e \times \tilde{\mathcal{W}}^e$ to be the set where equation (65) holds pointwise. $\mathbb{P}(\tilde{\mathcal{Z}}_{\mathsf{inv}}^e \times \tilde{\mathcal{W}}^e) = 1$. Owing to the independence $Z^e \perp W^e$, we have $\mathbb{P}(\tilde{\mathcal{Z}}_{\mathsf{inv}}^e) = 1$, $\mathbb{P}(\tilde{\mathcal{W}}^e) = 1$. It can be shown that the Lebesgue measure $\mu$ of the set $\mathcal{Z}_{\mathsf{inv}}^e \setminus \tilde{\mathcal{Z}}_{\mathsf{inv}}^e$ is zero, i.e., $\mu(\mathcal{Z}_{\mathsf{inv}}^e \setminus \tilde{\mathcal{Z}}_{\mathsf{inv}}^e) = 0$. If the Lebesgue measure was positive, i.e., $\mu(\mathcal{Z}_{\mathsf{inv}}^e \setminus \tilde{\mathcal{Z}}_{\mathsf{inv}}^e) > 0$, then the probability of this set would also be non-zero, i.e., $\mathbb{P}(\mathcal{Z}_{\mathsf{inv}}^e \setminus \tilde{\mathcal{Z}}_{\mathsf{inv}}^e) > 0$. The main insight to show this follows from the observation that the probability density is positive on the set $\mathcal{Z}_{\mathsf{inv}}^e \setminus \tilde{\mathcal{Z}}_{\mathsf{inv}}^e$ since the set is part of the support of $Z_{\mathsf{inv}}^e$.

A formal argument to show $\mu(\mathcal{Z}_{\mathsf{inv}}^e \setminus \tilde{\mathcal{Z}}_{\mathsf{inv}}^e) > 0 \implies \mathbb{P}(\mathcal{Z}_{\mathsf{inv}}^e \setminus \tilde{\mathcal{Z}}_{\mathsf{inv}}^e) > 0$ goes as follows.

Assume the contrary, i.e., $\mathbb{P}(\mathcal{Z}_{\mathsf{inv}}^e \setminus \tilde{\mathcal{Z}}_{\mathsf{inv}}^e) = 0$. Let the density be denoted as $f_{Z_{\mathsf{inv}}^e}$. Define the set $\mathcal{P}_k = \{z_{\mathsf{inv}} \in \mathcal{Z}_{\mathsf{inv}}^e \setminus \tilde{\mathcal{Z}}_{\mathsf{inv}}^e \mid f_{Z_{\mathsf{inv}}^e}(z) > \frac{1}{k}\}$.

$$\mathcal{Z}_{\mathsf{inv}}^e \setminus \tilde{\mathcal{Z}}_{\mathsf{inv}}^e = \cup_{k=1}^{\infty} \mathcal{P}_k \tag{66}$$

$\mathcal{P}_k \uparrow \mathcal{Z}_{\mathsf{inv}}^e \setminus \tilde{\mathcal{Z}}_{\mathsf{inv}}^e \implies \mu(\mathcal{P}_k) \to \mu(\mathcal{Z}_{\mathsf{inv}}^e \setminus \tilde{\mathcal{Z}}_{\mathsf{inv}}^e)$. Since $\mu(\mathcal{Z}_{\mathsf{inv}}^e \setminus \tilde{\mathcal{Z}}_{\mathsf{inv}}^e) > 0$, $\exists$ some $s$ for which $\mu(\mathcal{P}_s) > 0$.

Define $g_s$

$$g_s(x) = \begin{cases} \frac{1}{s} & \text{if } x \in \mathcal{P}_k \\ 0 & \text{otherwise} \end{cases} \tag{67}$$

$$\mathbb{P}(\mathcal{Z}_{\mathsf{inv}}^e \setminus \tilde{\mathcal{Z}}_{\mathsf{inv}}^e) = \int_{\mathcal{Z}_{\mathsf{inv}}^e \setminus \tilde{\mathcal{Z}}_{\mathsf{inv}}^e} f_{Z_{\mathsf{inv}}^e} d\mu \geq \int_{\mathcal{Z}_{\mathsf{inv}}^e \setminus \tilde{\mathcal{Z}}_{\mathsf{inv}}^e} g_s d\mu \geq \frac{1}{s} \mu(\mathcal{P}_s) > 0 \tag{68}$$

$\mu(\mathcal{Z}_{\mathsf{inv}}^e \setminus \tilde{\mathcal{Z}}_{\mathsf{inv}}^e) > 0 \implies \mathbb{P}(\mathcal{Z}_{\mathsf{inv}}^e \setminus \tilde{\mathcal{Z}}_{\mathsf{inv}}^e) > 0 \implies \mathbb{P}(\tilde{\mathcal{Z}}_{\mathsf{inv}}^e) < 1$ which is a contradiction. Therefore, $\mu(\mathcal{Z}_{\mathsf{inv}}^e \setminus \tilde{\mathcal{Z}}_{\mathsf{inv}}^e) = 0$.

We now describe how our assumptions already eliminate the possibility of distribution shifts that happen in such a way that the a finite mass of the distribution resides in the region $\mathcal{Z}_{\mathsf{inv}}^e \setminus \tilde{\mathcal{Z}}_{\mathsf{inv}}^e$. Recall we assume that $\forall e \in \mathcal{E}_{all}$, $Z_{\mathsf{inv}}^e$ is a continuous random variable. Since the probability of continuous random is absolutely continuous w.r.t the Lebesgue measure it follows that for each $e \in \mathcal{E}_{all}$, $\mu(\mathcal{Z}_{\mathsf{inv}}^e \setminus \tilde{\mathcal{Z}}_{\mathsf{inv}}^e) = 0 \implies \mathbb{P}(\mathcal{Z}_{\mathsf{inv}}^e \setminus \tilde{\mathcal{Z}}_{\mathsf{inv}}^e) = 0$. Thus all distribution shifts would place a zero mass in the region of disagreement.

This completes the first part of the proof.

The second part of the theorem follows directly from the analysis of linear regression SEM in Arjovsky et al. (2019). The conditions in the second part of the theorem cover the conditions that are required in Theorem 1. Under those conditions there can be two invariant predictors one is the trivial invariant predictor that maps every input to zero. The other is the ideal invariant predictor that focuses on the causes. The constraint $r^{\mathsf{th}}$ is set to a low enough value such that only the ideal invariant predictor gets selected. Observe that the risk achieved by the trivial zero invariant predictor is $\frac{1}{|\mathcal{E}_{tr}|} E[(Y^e)^2] = \sigma_Y^2$ and the risk achieved by the ideal $\frac{1}{|\mathcal{E}_{tr}|} E[(N^e)^2] = \sigma_N^2$. If $\sigma_N^2 < r^{\mathsf{th}} < \sigma_Y^2$, then the only predictor that is selected is the ideal invariant predictor.

We now describe why ERM fails in this case. In the theorem, we assume that $\exists\ e$ where $v = \mathbb{E}[\epsilon^e Z_{\mathsf{spu}}^e] \neq 0$, which implies $\mathbb{E}[\epsilon^e X^e] \neq 0$. We show why this is the case next.

$$\mathbb{E}[\epsilon^e X^e] = \mathbb{E}[\epsilon^e S(Z_{\mathsf{inv}}^e, Z_{\mathsf{spu}}^e)] = \mathbb{E}[S\epsilon^e(Z_{\mathsf{inv}}^e, Z_{\mathsf{spu}}^e)] = S(0, v) \neq 0; \text{since } S \text{ is invertible} \quad (69)$$

The rest of the proof follows from Proposition 17 in (Ahuja et al., 2021b). If $r^{\mathsf{th}}$ is set low enough to assume the same risk achieved by ERM, then IB-ERM and ERM are identical and IB-ERM also fails.

$\square$

**Remark on invertibility of $S$.** The entire proof extends to the case when $S$ is not invertible but $Z_{\mathsf{inv}}^e$ can still be recovered. Note that at no point in the proof we required to have full $S$ to be invertible.

**Remark on regularized ERM, IRM.** Note that while we showed that the ERM and IRM fail, the failures extend to $\ell_1$ or $\ell_2$ regularized models as well. We would like to also mention that it may seem that information bottleneck and sparsity constraints such as $\ell_1$ have similarity. We want to point out that there is a major difference between the two. In our model, we observe scrambled data. As a result, even if there is sparsity in the latent space, that does not translate to the observed space. $\ell_1$ constraints operate in the input space and that is why they cannot fetch the same outcome as information bottleneck constraints.

**Remark on multi-class classification.** The proof presented in this section extends to multi-class setting described in Assumption 10. The simplification in equation (53) along with the lemmas (Lemma 6, Lemma 7) help establish why low-entropy representation based classifier discourages the use of spurious features. We can adapt the analysis in equation (53) to the multi-class case (Assumption 10) and follow the same line of reasoning to justify why IB-IRM and IB-ERM succeed.

### A.7 Derivation of the final objective in equation (7)

In this section, we give a step-by-step description of derivation of the objective in equation (7). We rewrite the IB-IRM optimization below in equation (70).

$$
\begin{aligned}
\min_{\Phi \in \mathbb{R}^k} \ & \frac{1}{|\mathcal{E}_{tr}|} \sum_e h^e (w \cdot \Phi) \\
\text{s.t.} \ & \frac{1}{|\mathcal{E}_{tr}|} \sum_e R^e (w \cdot \Phi) \leq r^{\mathsf{th}}, \\
& 1 \in \arg\min_{\tilde{w} \in \mathbb{R}} R^e (\tilde{w} \cdot \Phi).
\end{aligned}
\tag{70}
$$

In the above we assumed that the classifiers are scalar. We state a new optimization that we show is equivalent to the optimization in equation (70).

$$
\begin{aligned}
\min_{\Phi \in \mathbb{R}^k} \ & \frac{1}{|\mathcal{E}_{tr}|} \sum_e h^e (\Phi) \\
\text{s.t.} \ & \frac{1}{|\mathcal{E}_{tr}|} \sum_e R^e (\Phi) \leq r^{\mathsf{th}}, \\
& 1 \in \arg\min_{\tilde{w} \in \mathbb{R}} R^e (\tilde{w} \cdot \Phi).
\end{aligned}
\tag{71}
$$

It can be shown that the two forms of optimization in equation (70) and equation (71) are equivalent. First, we would like to show that the set of feasible classifiers $w \cdot \Phi$ for the first optimization in equation (71) and $\Phi$ in the second optimization in equation (71) are the same.

Suppose $w^*, \Phi^*$ is a feasible solution to the constraints in equation (70). Construct $\Phi^\dagger = w^* \cdot \Phi^*$. $\Phi^\dagger$ satisfies the constraint $\frac{1}{|\mathcal{E}_{tr}|} \sum_e R^e (\Phi^\dagger) \leq r^{\mathsf{th}}$. Suppose for some environment $e$, $1 \notin \arg\min_{\tilde{w}} R^e (\tilde{w} \cdot \Phi^\dagger) \implies \exists \, w \neq 1$ such that $R^e (w \cdot \Phi^\dagger) < R^e (\Phi^\dagger)$. If this is the case, then $w \times w^*$ improves over $w^*$ and contradicts the optimality of $w^*$ in equation (70). This establishes that $\Phi^\dagger$ satisfies the constraints in equation (70). This shows that the set of feasible classifiers for the first optimization in equation (70) are a subset of the feasible classifiers in the second optimization (71).

Suppose $\Phi^*$ is a feasible solution to the constraints in equation (71). Take any scalar $w$ and corresponding representation $\Phi^*/w$. The combined classifier $w \cdot (\Phi^*/w)$ satisfies the first constraint. Suppose $w \notin \arg\min_{\tilde{w} \in \mathbb{R}} R^e (\tilde{w} \cdot \frac{\Phi}{w})$, this implies that $\exists \, w^+ \neq w$ such that $R^e (\frac{w^+}{w} \cdot \Phi^*) < R^e (\Phi^*)$. If this was true, then that contradicts the optimality of 1 in equation (71). This shows that the set of feasible classifiers for the second optimization in equation (71) are a subset of the feasible classifiers in the first optimization (70).

From the above discussion, it is clear that the two formulations result in the same set of feasible $w \cdot \Phi$, which are finally fed into the same entropy minimization objective. Thus the two optimizations are equivalent. To get to the penalized objective in equation (7) from the equation (71) there are two key steps: i) converting the invariance constraint into the gradient-based penalty, i.e., the IRMv1 penalty from (Arjovsky et al., 2019), ii) converting the differential entropy term into a constraint on the variance. For ii), as we explained in the manuscript, minimization of variance is equivalent to minimizing an upper bound on the entropy. Also, note that since variance has a lower bound, we can directly work with $\Phi$ and do not need to add a noise term like earlier, which was done to ensure that differential entropy is lower bounded. Below we break down the steps to arrive at the objective. We first start with a weighted combination of the terms in equation (6).

$$
\sum_e \Big( R^e (\Phi) + \lambda \| \nabla_{w, w=1.0} R^e (w \cdot \Phi) \|^2 + \nu h^e (\Phi) \Big).
\tag{72}
$$

where $\| \nabla_{w, w=1.0} R^e (w \cdot \Phi) \|^2$ is the norm of the gradient computed w.r.t scalar classifier $w$ at 1.0. Note that in general the gradient can be computed w.r.t a fixed vector as well. In our experiments, we

found that using entropy conditioned on the environment or entropy unconditioned on the environment works equally well. Thus, we introduce the unconditional entropy $h(\Phi)$. We assume that all the environments occur with an equal probability.

$$h(\Phi) = -\mathbb{E}_{X \sim \mathbb{P}}[\log(d\mathbb{P}(\Phi(X)))] \tag{73}$$

where $d\mathbb{P}(\Phi(X))$ is the probability density of predictions (unconditional on the environment), $\mathbb{P} = \frac{1}{|\mathcal{E}_{tr}|}\sum_{e \in \mathcal{E}_{tr}} \mathbb{P}^e$ is the uniform mixture of data from all environments. Note here $X$ denotes an input sample and we do not know the environment it comes from unlike the sample $X^e$. The entropy of predictions computed in environment $e$ is given as

$$h^e(\Phi) = -\mathbb{E}_{X^e \sim \mathbb{P}^e}[\log(d\mathbb{P}^e(\Phi(X^e)))], \tag{74}$$

where $d\mathbb{P}^e$ is the probability density of the predictions in environment $e$. The conditional entropy over predictions conditioned on a random environment is given as

$$h(\Phi|E) = -\frac{1}{|\mathcal{E}_{tr}|}\sum_{e \in \mathcal{E}_{tr}} \mathbb{E}[\log(d\mathbb{P}^e(\Phi(X^e)))]. \tag{75}$$

Conditioning reduces entropy $h(\Phi) \geq h(\Phi|E)$ and thus we propose an upper bound on the objective in equation (72) below

$$\sum_e \Big(R^e(\Phi) + \lambda\|\nabla_{w,w=1.0}R^e(w \cdot \Phi)\|^2 + \nu h(\Phi)\Big). \tag{76}$$

Finally, instead of $h(\Phi)$ we use variance in predictions $\Phi$ denoted as $\mathsf{Var}(\Phi) = \mathbb{E}_{X \sim \mathbb{P}}[(\Phi(X) - \mathbb{E}[\Phi(X)])^2]$ to get

$$\sum_e \Big(R^e(\Phi) + \lambda\|\nabla_{w,w=1.0}R^e(w \cdot \Phi)\|^2 + \gamma\mathsf{Var}(\Phi)\Big). \tag{77}$$

## A.8 Proof of Theorem 5: impact of IB on the learning speed

In this section, we present a detailed analysis of 2D case in equation (4) leading up to the proof of Theorem 5. For convenience, we will restate the equation (4). Also, instead of assuming the binary values are from the set $\{0, 1\}$ we would shift them to $\{-1, 1\}$; we do this purely for making notation clearer.

$$Y^e \leftarrow \mathsf{sgn}\left(X^e_{\mathsf{inv}}\right), \text{ where } X^e_{\mathsf{inv}} \in \{-1, 1\} \text{ is Bernoulli}\left(\frac{1}{2}\right),$$

$$X^e_{\mathsf{spu}} \leftarrow X^e_{\mathsf{inv}}W^e, \text{ where } W^e \in \{-1, 1\} \text{ is Bernoulli}\left(1 - p^e\right) \text{ with selection bias } p^e > \frac{1}{2}, \tag{78}$$

**Connection between the discrete and the continuous case.** Before discussing the proof of Theorem 5, we provide an explanation as to why can we use the variance penalty as a proxy for the 2D example (eq. (78)), where the random variables are discrete (recall that variance is monotonically related to upper bound on the differential entropy of continuous random variables). We present a variation of equation (78), where the input feature values are continuous. For each $e \in \mathcal{E}_{tr}$ we have

$$\begin{aligned} X^e_{\mathsf{inv}} &\leftarrow C^e + U^e, \\ Y^e &\leftarrow \mathsf{sgn}(X^e_{\mathsf{inv}}), \end{aligned} \tag{79}$$

where $C^e \in \{-1, 1\}$ with equal probability for $-1$ and $1$ and $U^e$ is a uniform random variable with range $[-\delta, \delta]$ with $\delta < \frac{1}{2}$. Similarly, with probability $1 - p^e$,

$$X^e_{\mathsf{spu}} \leftarrow C^e + M^e,$$

and with probability $p^e$,

$$X^e_{\mathsf{spu}} \leftarrow -C^e + M^e,$$

where $M^e$ is a uniform random variable with range $[-\delta, \delta]$.

Suppose $\ell$ is exponential loss and the predictor has two dimensions $w_{\mathsf{inv}}$ and $w_{\mathsf{spu}}$. For the above problem description, we write the ERM objective ($\lambda = 0, \gamma = 0$ in equation (7)) and we get the following

$$R_{\mathsf{ERM}}(w_{\mathsf{inv}}, w_{\mathsf{spu}}) =$$
$$\frac{1}{|\mathcal{E}_{tr}|} \sum_{e \in \mathcal{E}_{tr}} \left( p^e e^{-(w_{\mathsf{inv}}+w_{\mathsf{spu}})} \mathbb{E}[e^{-w_{\mathsf{inv}}U^e} e^{-w_{\mathsf{spu}}M^e}] + (1 - p^e) e^{-(w_{\mathsf{inv}}-w_{\mathsf{spu}})} \mathbb{E}[e^{-w_{\mathsf{inv}}U^e} e^{w_{\mathsf{spu}}M^e}] \right)$$

$$\mathbb{E}[e^{-w_{\mathsf{inv}}U^e} e^{-w_{\mathsf{spu}}M^e}] = \mathbb{E}[e^{-w_{\mathsf{inv}}U^e}] \mathbb{E}[e^{-w_{\mathsf{spu}}M^e}]$$

$$\mathbb{E}[e^{-w_{\mathsf{inv}}U^e}] = \left( \int_{-\delta}^{\delta} e^{-w_{\mathsf{inv}}u} du \right) \frac{1}{2\delta} = \frac{e^{w_{\mathsf{inv}}\delta} - e^{-w_{\mathsf{inv}}\delta}}{2w_{\mathsf{inv}}\delta} \approx \frac{(1 + w_{\mathsf{inv}}\delta) - (1 - w_{\mathsf{inv}}\delta)}{2w_{\mathsf{inv}}\delta} = 1 \tag{80}$$

If $\delta$ is small, then we can approximate the loss as if the each of the feature values were discrete and only assumed one of the four possible values in $\{-1, 1\} \times \{-1, 1\}$.

$$R_{\mathsf{ERM}}(w_{\mathsf{inv}}, w_{\mathsf{spu}}) \approx p e^{-(w_{\mathsf{inv}}+w_{\mathsf{spu}})} + (1 - p) e^{-(w_{\mathsf{inv}}-w_{\mathsf{spu}})} \tag{81}$$

where $p = \frac{1}{|\mathcal{E}_{tr}|} p^e$. On the same lines, we expand the IB-ERM objective as follows

$$R_{\mathsf{IB-ERM}}(w_{\mathsf{inv}}, w_{\mathsf{spu}}) \approx p e^{-(w_{\mathsf{inv}}+w_{\mathsf{spu}})} + (1 - p) e^{-(w_{\mathsf{inv}}-w_{\mathsf{spu}})} + \gamma [w_{\mathsf{inv}}, w_{\mathsf{spu}}] \Sigma [w_{\mathsf{inv}}, w_{\mathsf{spu}}]^{\mathsf{T}} \tag{82}$$

where $\Sigma = \begin{pmatrix} 1 + \delta^2 & 2p - 1 \\ 2p - 1 & 1 + \delta^2 \end{pmatrix}$. Since $\delta$ is small, we approximate $\Sigma$ as $\begin{pmatrix} 1 & 2p - 1 \\ 2p - 1 & 1 \end{pmatrix}$.

**Theorem on impact of information bottleneck.** We would compare the rate of convergence of continuous-time gradient descent for $R_{\mathsf{IB-ERM}}$ and $R_{\mathsf{ERM}}$.

**Theorem 10.** *Suppose each $e \in \mathcal{E}_{tr}$ follows the 2D case from equation (4). Set $\lambda = 0$, $\gamma > 0$ in equation (7) to get the IB-ERM objective with $\ell$ as exponential loss. Continuous-time gradient descent on this IB-ERM objective achieves $|\frac{w_{\mathsf{spu}}(t)}{w_{\mathsf{inv}}(t)}| \leq \epsilon$ in time less than $\frac{W_0(\frac{1}{2\gamma})}{2(1-p)\epsilon}$ ($W_0(\cdot)$ denotes the principal branch of the Lambert W function), while in the same time the ratio for ERM $|\frac{w_{\mathsf{spu}}(t)}{w_{\mathsf{inv}}(t)}| \geq \ln(\frac{1+2p}{3-2p})/\ln\left(1 + \frac{W_0(\frac{1}{2\gamma})}{2(1-p)\epsilon}\right)$, where $p = \frac{1}{|\mathcal{E}_{tr}|}\sum_{e \in \mathcal{E}_{tr}} p^e$.*

**Proof of Theorem 10.** We simplify the ERM and the IB-ERM objective in equation (7) for the 2D case.

$$R_{\mathsf{ERM}}(w_{\mathsf{inv}}, w_{\mathsf{spu}}) = pe^{-(w_{\mathsf{inv}}+w_{\mathsf{spu}})} + (1-p)e^{-(w_{\mathsf{inv}}-w_{\mathsf{spu}})}$$

$$R_{\mathsf{IB-ERM}}(w_{\mathsf{inv}}, w_{\mathsf{spu}}) = pe^{-(w_{\mathsf{inv}}+w_{\mathsf{spu}})} + (1-p)e^{-(w_{\mathsf{inv}}-w_{\mathsf{spu}})} + \gamma[w_{\mathsf{inv}}, w_{\mathsf{spu}}]\Sigma[w_{\mathsf{inv}}, w_{\mathsf{spu}}]^{\mathsf{T}}$$

where $w_{\mathsf{inv}}, w_{\mathsf{spu}} \in \mathbb{R}$ are the weights for invariant and spurious features, $p = \frac{1}{|\mathcal{E}_{tr}|}\sum_{e \in \mathcal{E}_{tr}} p^e$ $\Sigma$ as $\begin{pmatrix} 1 & 2p-1 \\ 2p-1 & 1 \end{pmatrix}$. We first find the equilibrium point of the continuous-time gradient descent for $R_{\mathsf{IB-ERM}}$.

$$\frac{\partial R_{\mathsf{IB-ERM}}(w_{\mathsf{inv}}, w_{\mathsf{spu}})}{\partial w_{\mathsf{inv}}} = -pe^{-(w_{\mathsf{inv}}+w_{\mathsf{spu}})} - (1-p)e^{-(w_{\mathsf{inv}}-w_{\mathsf{spu}})} + 2\gamma(w_{\mathsf{inv}} + (2p-1)w_{\mathsf{spu}})$$

$$\frac{\partial R_{\mathsf{IB-ERM}}(w_{\mathsf{inv}}, w_{\mathsf{spu}})}{\partial w_{\mathsf{spu}}} = -pe^{-(w_{\mathsf{inv}}+w_{\mathsf{spu}})} + (1-p)e^{-(w_{\mathsf{inv}}-w_{\mathsf{spu}})} + 2\gamma((2p-1)w_{\mathsf{inv}} + w_{\mathsf{spu}})$$

$$(83)$$

$$\frac{\partial R_{\mathsf{IB-ERM}}(w_{\mathsf{inv}}, w_{\mathsf{spu}})}{\partial w_{\mathsf{inv}}} + \frac{\partial R_{\mathsf{IB-ERM}}(w_{\mathsf{inv}}, w_{\mathsf{spu}})}{\partial w_{\mathsf{spu}}} = -2pe^{-(w_{\mathsf{inv}}+w_{\mathsf{spu}})} + 4\gamma p(w_{\mathsf{inv}} + w_{\mathsf{spu}}) = 0$$

$$\implies \frac{1}{2\gamma}e^{-(w_{\mathsf{inv}}+w_{\mathsf{spu}})} = w_{\mathsf{inv}} + w_{\mathsf{spu}}$$

$$\implies w_{\mathsf{inv}} + w_{\mathsf{spu}} = W_0\left(\frac{1}{2\gamma}\right)$$

$$(84)$$

$$\frac{\partial R_{\mathsf{IB-ERM}}(w_{\mathsf{inv}}, w_{\mathsf{spu}})}{\partial w_{\mathsf{inv}}} - \frac{\partial R_{\mathsf{IB-ERM}}(w_{\mathsf{inv}}, w_{\mathsf{spu}})}{\partial w_{\mathsf{spu}}} = -2(1-p)pe^{-(w_{\mathsf{inv}}-w_{\mathsf{spu}})} + 4\gamma(1-p)(w_{\mathsf{inv}} - w_{\mathsf{spu}}) = 0$$

$$\implies \frac{1}{2\gamma}e^{-(w_{\mathsf{inv}}-w_{\mathsf{spu}})} = w_{\mathsf{inv}} - w_{\mathsf{spu}}$$

$$\implies w_{\mathsf{inv}} - w_{\mathsf{spu}} = W_0\left(\frac{1}{2\gamma}\right)$$

$$(85)$$

Therefore, the equilibrium point is $w_{\mathsf{inv}} = W_0\left(\frac{1}{2\gamma}\right)$ and $w_{\mathsf{spu}} = 0$. Having established that the equilibrium point of the differential equation coincides with ideal predictor, we now analyze the convergence of the trajectory. Let $w_{\mathsf{inv}} + w_{\mathsf{spu}} = x$ and $w_{\mathsf{inv}} - w_{\mathsf{spu}} = y$.

$$\frac{\partial x}{\partial t} = -\left(\frac{\partial R_{\mathsf{IB-ERM}}(w_{\mathsf{inv}}, w_{\mathsf{spu}})}{\partial w_{\mathsf{inv}}} + \frac{\partial R_{\mathsf{IB-ERM}}(w_{\mathsf{inv}}, w_{\mathsf{spu}})}{\partial w_{\mathsf{spu}}}\right) = 2p(e^{-x} - 2\gamma x) \qquad (86)$$

$$\frac{\partial y}{\partial t} = 2(1-p)(e^{-y} - 2\gamma y) \qquad (87)$$

Let us call $x^* = W_0\left(\frac{1}{2\gamma}\right)$; $x^*$ is equilibrium point for both $x(t)$ and $y(t)$. Denote $w_{\mathsf{inv}}(t) = \frac{x(t)+y(t)}{2}$ and $w_{\mathsf{spu}}(t) = \frac{x(t)-sy(t)}{2}$. Let us assume that $x(0) = 0$ and $y(0) = 0$. We would first like to argue that the solution to the above differential equations exist and are unique given the initial conditions

$x(0) = 0$ and $y(0) = 0$. Since $(e^{-x} - 2\gamma x)$ is Lipschitz continuous in $x$ on $\mathbb{R}$ the solution to the differential equation exists and is unique for any finite interval $t \in [0, T]$ (Simmons, 2016). With $T$ set to a sufficiently large value, we now show that the solution to the ODE converges to $x^*$.

Define an energy function $V(z) = z^2$ and define $V(x - x^*) = (x - x^*)^2$

$$\frac{\partial V(x - x^*)}{\partial t} = 2(x - x^*)\frac{\partial x}{\partial t} = 4p(x - x^*)(e^{-x} - 2\gamma x) \tag{88}$$

Observe that $\frac{\partial V(x-x^*)}{\partial t} < 0$ for all $x \neq x^*$ and $\frac{\partial V(x-x^*)}{\partial t} = 0$ when $x = x^*$. Therefore, from Lyapunov's asymptotic global stability theorem (Khalil, 2009) we obtain that $x(t)$ would converge to $x^*$.

Observe that for $x < x^*$, $\frac{\partial x}{\partial t} > 0$ and moreover $2p(e^{-x} - 2\gamma x)$ is a monotonically decreasing function. For all $x < x^* - \epsilon$, we can bound the rate at which $x$ increases is bounded below by $2p(e^{-x^*+\epsilon} - 2\gamma(x^* - \epsilon)) \approx 2p(e^{-x^*}(1 + \epsilon) - 2\gamma x^* + 2\gamma\epsilon) = 2p\epsilon(e^{-x^*} + 2\gamma)$. Let us call $\gamma^* = \epsilon(e^{-x^*} + 2\lambda)$. The rate at which $x$ increases is greater than $2p\epsilon\gamma^*$ and the rate at which $y$ increases is greater than $2(1 - p)\epsilon\gamma^*$. Thus the time to convergence for $x$ is atmost $\frac{x^*}{2p\epsilon}$. Similarly, the time to convergence for $y$ is atmost $\frac{x^*}{2(1-p)\epsilon}$. Since $p > \frac{1}{2}$ the time to convergence for $y(t)$ is more than the time taken for the convergence of $x(t)$.

If $|x(t) - x^*| \leq \epsilon$ and $|y(t) - x^*| \leq \epsilon$, then $|w_{\mathsf{spu}}(t)| = |\frac{x(t)-y(t)}{2}| = |\frac{x(t)-x^*+x^*-y(t)}{2}| \leq \frac{|x(t)-x^*|}{2} + \frac{|y(t)-x^*|}{2} \leq \epsilon$.

If $|x(t) - x^*| \leq \epsilon$ and $|y(t) - x^*| \leq \epsilon$, then $|w_{\mathsf{inv}}(t) - x^*| = |\frac{x(t)+y(t)}{2} - x^*| = |\frac{x(t)-x^*+y(t)-x^*}{2}| \leq \frac{|x(t)-x^*|}{2} + \frac{|y(t)-x^*|}{2} \leq \epsilon$.

As a result, if $|x(t) - x^*| \leq \epsilon$ and $|y(t) - x^*| \leq \epsilon$, then

$$\frac{|w_{\mathsf{spu}}(t)|}{|w_{\mathsf{inv}}(t)|} \leq \frac{\epsilon}{x^* - \epsilon} \approx \frac{\epsilon}{x^*} \tag{89}$$

Therefore, to get the ratio $\frac{|w_{\mathsf{spu}}(t)|}{|w_{\mathsf{inv}}(t)|} \leq \frac{\epsilon}{x^*}$ the time taken is at most $\frac{x^*}{2(1-p)\epsilon}$.

In comparison in the same amount of time the ratio $|\frac{w_{\mathsf{spu}}(t)}{w_{\mathsf{inv}}(t)}|$ achieved by gradient descent on $R_{\mathsf{ERM}}$ is at least $\frac{\ln(\frac{1+2p}{3-2p})}{\ln(1+\frac{x^*}{2(1-p)\epsilon})}$. The expression for lower bound on the ratio $|\frac{w_{\mathsf{spu}}(t)}{w_{\mathsf{inv}}(t)}|$ is derived by substituting the time taken, i.e., $\frac{x^*}{2(1-p)\epsilon}$, in the expression for the lower bound derived in Section B.3 in Nagarajan et al. (2021)). $\qquad\square$

**Remark on max-margin classifiers.** In the 2D example, the max-margin classifier seems to solve the problem. In general max-margin classifier would not work. In the more general setting, if there is noise in the labels, which is allowed by the SEM in Assumption 8, and the data is scrambled, which is also the case in Assumption 8, there is no guarantee that max-margin classifier would not rely on the spurious features.

## A.9 Illustrating both invariance and information bottleneck acting in conjunction

In this section, we present a case to illustrate why the invariance principle and the information bottleneck are needed simultaneously. The model we present follows a DAG that combines the DAGs in Figure 2a) and Figure 2b).

**Example extending the 2D case from equation** (4). For all the environments $e \in \mathcal{E}_{tr}$

$$
\begin{aligned}
Y^e &\leftarrow X^e_{\text{inv}} \oplus N^e \\
X^{1,e}_{\text{spu}} &\leftarrow Y^e \oplus W^e \\
X^{2,e}_{\text{spu}} &\leftarrow X^e_{\text{inv}} \oplus V^e
\end{aligned}
\tag{90}
$$

where all the variables in the above SEM are binary $\{0,1\}$ random variables. $N^e \sim \text{Bernoulli}(q)$, $V^e \sim \text{Bernoulli}(a)$; the distribution of noise $N^e$ and $V^e$ are the same across the environments. $W^e \sim \text{Bernoulli}(u^e)$ where $u^e$ is an environment dependent probability. For all the environments $e \in \mathcal{E}_{all}$, we assume that the distribution of $X^e_{\text{inv}}$, $N^e$, and $V^e$ does not change. The labelling function to generate $Y^e$ is also the same. The distribution of $X^{1,e}_{\text{spu}}$ can change arbitrarily. In this example, observe that $\mathbb{E}[Y^e|X^e]$ varies across the training environments. We show the simplification below.

$$
\mathbb{E}[Y^e|X^e] = \mathbb{E}\Big[X^e_{\text{inv}} \oplus N^e \big| (X^e_{\text{inv}}, X^{1,e}_{\text{spu}}, X^{2,e}_{\text{spu}})\Big]
\tag{91}
$$

If $X^e_{\text{inv}} = 0, X^e_{\text{spu}} = 0$, then $\mathbb{E}[Y^e|X^e] = \mathbb{P}(N^e = 1|X^e_{\text{inv}} = 0, X^{1,e}_{\text{spu}} = 0)$. We show that $\mathbb{P}(N^e = 1|X^e_{\text{inv}} = 0, X^{1,e}_{\text{spu}} = 0)$ varies across the environments.

$$
\begin{aligned}
\mathbb{P}(N^e = 1|X^e_{\text{inv}} = 0, X^{1,e}_{\text{spu}} = 0) &= \frac{\mathbb{P}(N^e = 1, X^e_{\text{inv}} = 0, X^{1,e}_{\text{spu}} = 0)}{\mathbb{P}(N^e = 1, X^e_{\text{inv}} = 0, X^{1,e}_{\text{spu}} = 0) + \mathbb{P}(N^e = 0, X^e_{\text{inv}} = 0, X^{1,e}_{\text{spu}} = 0)} \\
&= \frac{\mathbb{P}(N^e = 1, X^e_{\text{inv}} = 0)u^e}{\mathbb{P}(N^e = 1, X^e_{\text{inv}} = 0)u^e + \mathbb{P}(N^e = 0, X^e_{\text{inv}} = 0)(1 - u^e)}
\end{aligned}
\tag{92}
$$

Note that the above equation (92) describes the probability computed by the Bayes optimal classifier that relies on input feature dimensions are used. Observe that the above probability in equation (92) can only be equal across two environments if $u^e$ was the same. Therefore, if $|\mathcal{E}_{tr}| \geq 2$ and the probability $u^e$ varies across the environments, then the invariance constraint restrict us from using the identity representation. However, $\mathbb{E}[Y^e|X^e_{\text{inv}}, X^{2,e}_{\text{spu}}]$ is invariant and so is $\mathbb{E}[Y^e|X^e_{\text{inv}}]$. Based on the same arguments that we discussed in the main manuscript, we can show that one can construct classifiers that output probability distributions that minimize cross-entropy (maximize likelihood) and continue to depend on $X^{2,e}_{\text{spu}}$ as follows

$$
\begin{aligned}
\hat{\mathbb{P}}(Y^e = 1|X^e_{\text{inv}}, X^{2,e}_{\text{spu}}) &= (1-q)\mathsf{I}\Big(w_{\text{inv}}X^e_{\text{inv}} + w_{\text{spu}}X^e_{\text{spu}} - \frac{(w_{\text{inv}} + w_{\text{spu}})}{2}\Big) + \\
&\quad q\Big(1 - \mathsf{I}\Big(w_{\text{inv}}X^e_{\text{inv}} + w_{\text{spu}}X^e_{\text{spu}} - \frac{(w_{\text{inv}} + w_{\text{spu}})}{2}\Big)\Big).
\end{aligned}
\tag{93}
$$

If $w_{\text{inv}} > |w_{\text{spu}}|$, then above classifier $\hat{\mathbb{P}}(Y^e = 1|X^e_{\text{inv}}, X^{2,e}_{\text{spu}})$ matches the true probability distribution conditional on the invariant feature $\mathbb{P}(Y^e = 1|X^e_{\text{inv}})$ on all the training environments and it thus forms a valid invariant predictor with representation that focuses on $X^e_{\text{inv}}, X^{2,e}_{\text{spu}}$. Since the classifier relies on $X^{2,e}_{\text{spu}}$, the classifier fails as the support of spurious features can change. If we place an entropy constraint, then the representation that focuses only on $X^e_{\text{inv}}$ is strictly prefered to one that focuses on both $X^e_{\text{inv}}, X^{2,e}_{\text{spu}}$ and continues to achieve the same cross-entropy loss. Thus in this example, IRM fails as its solution space contains classifiers that rely on spurious features but IB-IRM would succeed. In the above example, ERM and IB-ERM (with $r^{\text{th}}$ set to match the loss of ERM) will rely on $X^{1,e}_{\text{spu}}$ on top of $X^e_{\text{inv}}$ as conditioning on $X^{1,e}_{\text{spu}}$ in addition to $X^e_{\text{inv}}$ further reduces the conditional entropy thus reducing the cross-entropy loss.

Let us consider a generalization of the above example.

**Assumption 11.** *Each environment $e \in \mathcal{E}_{all}$ follows*

$$Y^e \leftarrow \mathsf{I}\big(w_{\mathsf{inv}}^* \cdot X_{\mathsf{inv}}^e\big) \oplus N^e \tag{94}$$

*$N^e$ is binary noise, and $X_{\mathsf{inv}}^e$ are binary features. Both $N^e$ and $X_{\mathsf{inv}}^e$ have identical distributions across all the environments $\mathcal{E}_{all}$*

Divide the spurious features into two parts $X_{\mathsf{spu}}^e = (X_{\mathsf{spu}}^{1,e}, X_{\mathsf{spu}}^{2,e})$.

**Assumption 12.** *Each environment $e \in \mathcal{E}_{tr}$ follows*

$$\begin{aligned}
X_{\mathsf{spu}}^{1,e} &\leftarrow Y^e \mathbf{1} \oplus W^e \\
X_{\mathsf{spu}}^{2,e} &\leftarrow X_{\mathsf{inv}}^e \oplus V^e
\end{aligned} \tag{95}$$

*where $\mathbf{1} \in \mathbb{R}^{o'}$ is a vector of ones, $W^e \in \mathbb{R}^{o'}$ is a binary 0-1 vector with each component drawn i.i.d. from* Bernoulli$(u^e)$ *vector, $V^e$ is also a binary 0-1 vector with each component drawn i.i.d. from* Bernoulli$(a)$ *vector. The distribution of $W^e$ changes across environments and no two training environments have the same $u^e$. The distribution of $V^e$ is identical across all the training environments. Also, assume that there are at least two training environments, i.e., $|\mathcal{E}_{tr}| \geq 2$.*

**Assumption 13.** *$\mathcal{H}_\Phi$ is a set of diagonal matrices, where each element in the matrix is $0$ or $1$ ($\mathcal{H}_\Phi$ act as matrices that seletct subset of input features). $\mathcal{H}_w$ is set of all probability distributions on $\mathbb{R}^d$. $\ell$ is the cross-entropy loss.*

We use the Shannon entropy formulation of IB-IRM in this case as all the random variables involved are discrete. Moreover, we carry out entropy minimization for the representation directly and not the predictor. The IB-IRM optimization is given as follows.

$$\begin{aligned}
&\min_{\Phi \in \mathcal{H}_\Phi} \frac{1}{|\mathcal{E}_{tr}|} \sum_e H^e(\Phi) \\
&\text{s.t. } \frac{1}{|\mathcal{E}_{tr}|} \sum_e R^e\big(w \circ \Phi\big) \leq r^{\mathsf{th}} \\
&\quad w \in \arg \min_{\tilde{w} \in \mathcal{H}_w} R^e(\tilde{w} \circ \Phi)
\end{aligned} \tag{96}$$

**Theorem 11.** *Suppose the data follows Assumption 11, Assumption 12. Suppose $\mathcal{H}_w$ and $\mathcal{H}_\Phi$ follow Assumption 13. If invariant features are strictly separable, i.e., Assumption 7 holds, then IRM fails but IB-IRM succeeds.*

**Proof of Theorem 11.** We carry out the analysis for different types of representations separately.

Case 1: Consider a representation that selects a subset $\tilde{X}_1^e$ of $(X_{\mathsf{inv}}^e, X_{\mathsf{spu}}^{2,e})$ and a subset $\tilde{X}_2^e$ of $X_{\mathsf{spu}}^{1,e}$.

$$\begin{aligned}
\mathbb{P}(Y^e = 1 | \tilde{X}_1^e = 0, \tilde{X}_2^e = 0) &= \frac{\mathbb{P}(Y^e = 1, \tilde{X}_1^e = 0, \tilde{X}_2^e = 0)}{\mathbb{P}(Y^e = 1, \tilde{X}_1^e = 0, \tilde{X}_2^e = 0) + \mathbb{P}(Y^e = 0, \tilde{X}_1^e = 0, \tilde{X}_2^e = 0)} \\
&= \frac{\mathbb{P}(Y^e = 1, \tilde{X}_1^e = 0)(u^e)^{o'}}{\mathbb{P}(Y^e = 1, \tilde{X}_1^e = 0)(u^e)^{o'} + \mathbb{P}(Y^e = 1, \tilde{X}_1^e = 0)(1 - u^e)^{o'}}
\end{aligned} \tag{97}$$

Since $\mathbb{P}(Y^e = 1 | \tilde{X}_1^e = 0, \tilde{X}_2^e = 0)$ is strictly monotonic in $u^e$, this probability cannot be same across two environments. Hence, any $\tilde{X}_1^e, \tilde{X}_2^e$ cannot lead to an invariant predictor across the two environments.

Case 2: Consider a representation that selects a subset $\tilde{X}^e$ of $X_{\mathsf{spu}}^{1,e}$.

$$\begin{aligned}
\mathbb{P}(Y^e = 1 | \tilde{X}^e = 0) &= \frac{\mathbb{P}(Y^e = 1, \tilde{X}^{1,e} = 0)}{\mathbb{P}(Y^e = 1, \tilde{X}^{1,e} = 0) + \mathbb{P}(Y^e = 0, \tilde{X}^{1,e} = 0)} \\
&= \frac{\mathbb{P}(Y^e = 1)(u^e)^{o'}}{\mathbb{P}(Y^e = 1)(u^e)^{o'} + \mathbb{P}(Y^e = 0)(1 - u^e)^{o'}}
\end{aligned} \tag{98}$$

For the above class of representations also, we can use the same argument as the one discussed in Case 1 and show that the above probability cannot be the same across two environments.

Case 3: At this point, our only option is to consider representations that select subsets of $(X_{\mathsf{inv}}^e, X_{\mathsf{spu}}^{2,e})$. Each subset of $(X_{\mathsf{inv}}^e, X_{\mathsf{spu}}^{2,e})$ satisfies invariance. Among this set all the subsets that lead to lowest cross-entropy are selected by IRM. Among those sets IRM does not exclude the inclusion of spurious covariates $X_{\mathsf{spu}}^{2,e}$. However, when we impose entropy minimization objective, then $X_{\mathsf{spu}}^{2,e}$ will never be selected as entropy can be strictly reduced by not including these covariates in the set without sacrificing invariance or cross-entropy. To explicitly show a construction of the failure of IRM in this case, we can use the same construction as equation (93) but replacing the hyperplane in the indicator function with hyperplane constructed in Lemma 4.

$\square$

## A.10 Related works

### A.10.1 Invariance principles in causality

The foundations of invariance principles are rooted in the theory of causality (Pearl, 1995). There are several different forms in which the invariance principles or principles similar to it appear in the literature on causality. Modularity condition states that a variable $Y$ is caused by a set of variables $X_{\mathrm{Pa}(Y)}$ if and only if under all interventions other than those on $Y$ the conditional probability $\mathbb{P}(Y|X_{\mathrm{Pa}(Y)})$ remains invariant. Related and similar notions are *stability* (Pearl, 2009), *autonomy* (Schölkopf et al., 2012), *invariant causal prediction principle* (Peters et al., 2016; Heinze-Deml et al., 2018). These principles lead to a powerful insight – if we model all the environments (train and test) using interventions, then as long as these interventions do not affect the causal mechanism that generates the target variable $Y$, a classifier trained only on the transformation that extracts causal variables ($\Phi(X) = X_{\mathrm{Pa}(y)}$) to predict $Y$ is invariant under interventions.

### A.10.2 Invariance principles in OOD generalization

In recent years, there has been a surge in the works inspired from causality, examples of some notable works are (Peters et al., 2016; Arjovsky et al., 2019), which seek to address OOD generalization failures. The invariance principle is at the heart of many of these works. For a better understanding, we divide these works into two categories – theory and methods, though some works belong to both.

**Theory.** In Rojas-Carulla et al. (2018) it was shown that the predictors trained on the causes are min-max optimal under a large class of distribution shifts modeled by the interventions. These findings were generalized in Koyama and Yamaguchi (2020). Given that we know that predictors that focus on the causes are min-max optimal under many distribution shifts, the central question then is – can we learn these predictors from a finite set of training distributions/environments? Arjovsky et al. (2019) showed how to achieve such causal predictors that generalize OOD from a finite set of training environments for linear regression tasks under very general assumptions. Rosenfeld et al. (2021b) considered linear classification tasks where invariant features were partially informative w.r.t the label and showed that under assumptions of support overlap for invariant and spurious features, it is possible to learn predictors that generalize OOD. In this work, we analyze classification tasks but different from Rosenfeld et al. (2021b) we consider both fully and partially informative features. We showed that support overlap of invariant features is necessarily needed for OOD generalization in classification tasks else OOD generalization, in general, is impossible. On the other hand, we showed that support overlap for spurious features is not necessary but in its absence standard methods such as ERM and IRM can fail.

Recent works (Rosenfeld et al., 2021b,a; Kamath et al., 2021; Gulrajani and Lopez-Paz, 2021) have also pointed to several limitations of invariance based approaches for addressing OOD generalization failures. In Rosenfeld et al. (2021b), the authors showed that if we use the IRMv1 objective, then for non-linear tasks the solutions from IRMv1 are no better than ERM in generalizing OOD. In Lu et al. (2021), the authors present a two-phased approach to addressing the difficulties faced by IRM in the non-linear regime. In the first phase, an identifiable variational autoencoder (Khemakhem et al., 2020) is used to extract the latent representations from the raw input data. In the second phase, causal discovery-based approaches are used to identify the causal parents of the label and then learn predictors based on the causal parents only. The entire analysis in Lu et al. (2021) is for the setting when the invariant features are partially informative about the label. Also, the analysis assumes that we have access to side information (possibly in the form of environment index) that can help disentangle all the latent features, i.e., all the latent features are independent conditioned on this side information. Having access to such information, in general, is a strong assumption. In Kamath et al. (2021), the authors show that if the label and feature space is finite and if the distribution shifts are captured by analytic functions, then the set of invariant predictors found from two environments exactly capture all the invariant predictors described by the analytic function. While this is a very interesting and important result, we would like to point out that the distribution shifts captured using analytic functions represent a small family of interventions that are otherwise allowed when learning predictors that focus on causes.

In this work, we focused on linear SEMs unlike the non-linear SEMs described above. The setting that we considered in this work has three salient features – a) classification when invariant features are fully informative, b) spurious features are correlated with invariant features, and c) arbitrary shifts

are allowed on the spurious feature distribution. This setting is important as many of the existing failures correspond to this setting. We are the first to give provable OOD generalization guarantees for this setting. Considering non-linear models is a natural next step. On this note, we would like to mention that we believe several of our results can be generalized to the setting when the mapping from the latents to the raw data is piecewise linear.

**Methods.** Following the original works ICP (Peters et al., 2016) and IRM (Arjovsky et al., 2019), there have been several interesting works — (Teney et al., 2020; Krueger et al., 2020; Ahuja et al., 2020; Jin et al., 2020; Chang et al., 2020; Ahuja et al., 2021a; Mahajan et al., 2020; Koyama and Yamaguchi, 2020; Müller et al., 2020; Parascandolo et al., 2021; Ahmed et al., 2021; Robey et al., 2021; Zhang et al., 2021) is an incomplete representative list — that build new methods inspired from IRM to address the OOD generalization problem. We would not go into the details of these different works. However, we believe it is important to talk about works that use conditional independence-based criterion to achieve invariance (Koyama and Yamaguchi, 2020; Huszár, 2019) as those objectives also involve mutual information. Invariance can be enforced using conditional independence as follows. Suppose the environment is given as a random variable $E$. In this case, if we can learn a representation $\Phi(X)$ such that $Y \perp E|\Phi(X)$, then the predictors learned on $\Phi$ are invariant predictors. This conditional independence constraint is formulated in the form of mutual information-based criterion in (Koyama and Yamaguchi, 2020; Huszár, 2019). In this work, we argue that often in classification tasks, there are many representations $\Phi$ that satisfy $Y \perp E|\Phi(X)$ and we have to learn the one that has the least entropy or otherwise OOD generalization is not possible.

### A.10.3 Theory of domain adaptation and domain generalization

In the previous section, we discussed works that were directly based on causality/invariance or inspired from it. We now briefly review other relevant works on domain adaptation and domain generalization that are not based on invariance principle from causality. Starting with the seminal works (Ben-David et al., 2007, 2010), there have been many other interesting works in the area of domain adaptation and domain generalization. (Muandet et al., 2013; Zhao et al., 2019; Albuquerque et al., 2019; Piratla et al., 2020; Matsuura and Harada, 2020; Deng et al., 2020; Pagnoni et al., 2018; Greenfeld and Shalit, 2020; Garg et al., 2021) is an incomplete representative list of works that build the theory of domain adaptation and generalization and construct new methods based on it. We recommend the reader to Redko et al. (2019) for further references.

In the case of domain adaptation, many of these works develop bounds on the loss over the target domain using train data and unlabeled target data. In the case of domain generalization, these works develop bounds on the loss over the target domains using training data from multiple domains. Other works (Ben-David and Urner, 2012; David et al., 2010) analyze the minimal conditions under which domain adaptation is possible. In David et al. (2010), the authors showed that the two most common assumptions, a) covariate shifts, and b) the presence of a classifier that achieves close to ideal performance simultaneously in train and test domains, are not sufficient for guaranteed domain adaptation. In this work, we established the necessary and sufficient conditions for domain generalization in linear classification tasks. We showed that under a) covariate shift assumption (SEM in Assumption 2 satisfies the covariate shift), and b) the presence of a common labelling function across all the domains (a much stronger condition than assuming the existence of a classifier that achieves low error across the train and test domains), domain generalization in linear classification is impossible. We showed that adding the requirement that the invariant features satisfy support overlap is both necessary and sufficient (our approach IB-IRM succeeds while IRM and ERM fail) in many cases to guarantee domain generalization.

There has been a long line of research focused on learning domain invariant feature representations (Ganin et al., 2016; Li et al., 2018; Zhao et al., 2020). In these works, the common assumption is that the there exist highly predictive representations whose distributions $\mathbb{P}(\Phi(X^e))$(or distributions conditional on the labels $\mathbb{P}(\Phi(X^e)|Y^e)$) do not change across environments. Note that this is a much stronger assumption than the one typically made in works based on invariance principle (Arjovsky et al., 2019), where the labelling function ($\mathbb{P}(Y^e|\Phi(X^e))$) does not change. For a detailed analysis of why the assumptions made in these works are too strong and can often fail refer to Arjovsky et al. (2019); Zhao et al. (2019).

### A.10.4 Other works on OOD generalization

In Nagarajan et al. (2021) the authors explained why ERM based models trained with gradient descent based approaches fail to generalize OOD in terms of two failure modes – a) gradient descent during training early on relies on shortcut features, b) overparametrized models exhibit geometric biases that cause the models to rely on spurious features. We now describe the line of work based on domain adaptation. For failure mode described in a), we showed in Theorem 5 how information bottleneck penalty can help. Sagawa et al. (2019) studied how overparametrized models can exacerbate the impact of selection biases, Xie et al. (2021) studied the role of auxilliary information and how it can help OOD generalization.

### A.10.5 Information bottleneck penalties and impact on generalization

Information bottleneck principle (Tishby et al., 2000) has been used to explain the success of deep learning models; the principle has also been used to build regularizers that can help build models that achieve better in-distribution generalization. We refer the reader to Kirsch et al. (2020), which presents an excellent summary of the existing works on information bottleneck in deep learning. Kirsch et al. (2020) also present a unified framework to view many of the information bottleneck objectives in the literature such as the deterministic information bottleneck (Strouse and Schwab, 2017) and the standard information bottleneck. Other works (Alemi et al., 2016; Arpit et al., 2019) have argued for how information bottleneck can help achieve robustness to adversarial examples, and also to OOD generalization failures. In Arpit et al. (2019), the authors argued that information bottleneck constraints help filter out features that are less correlated with the label. However, the principle of invariance argues for selecting the invariant features even if they have small but invariant correlation with the label over features that maybe strongly correlated but have a varying correlation. As we showed, considering both the principles of invariance and information bottleneck in conjunction is important to achieve OOD generalization (eq. (1)) in a wide range of settings – when the invariant features are fully informative about the label and also when they are partially informative about the label.