# OpenReview forum: "Invariance Principle Meets Information Bottleneck for Out-of-Distribution Generalization"
_NeurIPS.cc/2021/Conference — NeurIPS 2021 Spotlight_

### Official Review · Reviewer_YQ8U · 2021-07-16

**Rating:** 5
**Confidence:** 4

**Summary:**

- The authors first argue that IRM is insufficient to obtain good test performance (only using invariant features) on OOD data at test time when the OOD support is not in the ID support
- Then they prove for a particular distributional assumption on the spurious features, that (additionally) limiting the differential entropy of the representation can solve this issue and lead to OOD generalizing predictors
- In some toy experiments, they show that IB-IRM sometimes performs better

**Limitations And Societal Impact:**

Authors do have a section on limitations and motivate future work through that.
There's no potential negative societal impact for OOD detection per se ... (for certain applications it could, but this holds for almost all problems that one can think of solving).

**Main Review:**

Strengths:
- They argue that the information bottleneck in some situations can help to obtain the invariant predictor when invariance itself cannot (when there’s not enough support overlap)
- In their (few) experiments on modified colored MNIST datasets, including one where IRM does not improve much compared to ERM (CS-MNIST) and demonstrate that adding the IB regularizer leads to better OOD generalization

Weaknesses:

Theory
- The insufficiency results for IRM are not surprising, e.g. FIIF problem was already noted in Nagarajan et al
- Comparison with other works that show limits of IRM such as Srebro - their examples themselves seem to make a similar point as FIIF?

Since the theoretical contributions are insufficient, we now examine the methodological contributions
Methodology:
- major criticism: is this supposed to be a game-changer for practice? then it’s absolutely necessary to do OOD generalization experiments on real world datasets (such as WILDS). the current datasets are completely toy (versions of colored MNIST)
- since variances are huge compared to MSE for examples 1/1s, 3/3s with 6 environments, one can really only compare the performances of the methods using CS-MNIST and AC-MNIST and perhaps example 2/2s which is rather unsatisfying as empirical evidence for a method
- table 3 suggests that more environments worsens the benefits of adding IB even in an FIIF setting? more discussion/experiments on that would be necessary
- again regarding the point that the variances are huge for Example 1/1s with 6 vs. 3 environments: a comment on that in the main text would be useful - is that expected and why?

Update: Apologies, I actually was not able to understand the problem that I previously had when I had a brief new look at the paper (before even reading your response). I'm guessing I was thinking about some problems that confused my mindset when reading. Now the sufficiency result actually sounded trivial since in the case of spurious correlations AND support overlap all possible environments in E_all would also be spuriously correlated...

Thanks for discussion on relation to Kamath et al.

I agree Nagarajan did not provide formal results, but hinted at the problem so this is why I said it is not surprising.

Thanks for comment on six-environments to discuss the high variance. That comment should be added to paper.

Adding experiments on Coco addresses my main concern (I still think the methodology is the main contribution). However I would like to see more experimental details before giving the nod + experiments on one more dataset. The lack of real-world experimental evidence was the primal reason for the score and hence I still tend to reject the paper. The number of scenarios where IB-IRM or IB-ERM do better is still rather limited and currently only includes COCO in terms of non-toy dataset (without having been able to see the revision and experimental details) which is in my opinion still quite rather meager for publication at Neurips. How about WILDS? I believe if the authors run a more extensive experimental study this paper can be accepted at the next venue.

I didn't see this point addressed: "Table 3 suggests that more environments worsens the benefits of adding IB even in an FIIF setting? more discussion/experiments on that would be necessary"

**Time Spent Reviewing:**

7

---

> ### Author Response · Authors · 2021-08-10
> **Response to Reviewer YQ8U (Part 1)**
>
> Thanks a lot for your comments. It appears, however, that there are several misunderstandings regarding our contributions, which we hope to clarify below.
>
> 1. **On the correctness of sufficiency result stated in Theorem 3.** The result is correct and here we provide more detail. In the sufficiency result, we assume support overlap for both the invariant and the spurious features, which is what allows all the solutions of ERM (or IRM) to generalize even with one environment. We begin with the 2D example from the paper to give a simple justification for the sufficiency result and then we give insights on the key proof steps.
> Let us recall the 2D example first (from equation 4 in the paper). The data $(X_{\mathsf{inv}}, X_{\mathsf{spu}})$ can only take four possible values $(0,0)$, $(0,1)$, $(1,0)$, $(1,1)$ in all the environments (there is one train environment and the rest are test environments) owing to the support overlap assumption in Theorem 3. The label $Y=X_{\mathsf{inv}}$. Define $\mathcal{W}$ as the set of classifiers that solve ERM. Each of these classifiers achieves an error of zero in the training environment. We now consider a candidate classifier in $\mathcal{W}$ that maximally relies on the spurious feature and hence, as per reviewer's intuition, it should be the best candidate to fail. Set the weight of the invariant feature to $1+\epsilon$ and that of the spurious feature to $1$,  where $\epsilon$ is a very small positive value. The classifier output is $1$ if  $(1+\epsilon)X_{\mathsf{inv}} + X_{\mathsf{spu}} - 1-\frac{\epsilon}{2}>0$ and $0$ otherwise. Suppose that the correlation between the spurious feature and the label is given by $p^{e_{\mathsf{train}}}$ and let $p^{e_{\mathsf{train}}}=0.99$ (in Figure 2a of the paper we denote the correlation by the color intensity of the points). Suppose the correlation flips to $p^{e_{\mathsf{test}}}=0.01$ at test time, which is exactly the setting that we believe the reviewer has in mind. (To visualize this in the figure in the paper, the color intensity would flip.) Note that even if there is a flip, the points are still perfectly classified by  $(1+\epsilon)X_{\mathsf{inv}} + X_{\mathsf{spu}} - 1-\frac{\epsilon}{2}$ as the support does not change, i.e., the four points remain where they are w.r.t the classifier at test time. However, if the support of the spurious features changes, then ERM is insufficient, which is what we show in the second part of Theorem 3.
>
>    The idea that we highlight above extends to the general case presented in Theorem 3. We give a high-level summary of the key proof steps. At first, all the ERM solutions have to satisfy equation (35) in the Supplement, i.e.,  for all the points in the training support, i.e., $(Z_{\mathsf{inv}}, Z_{\mathsf{spu}}) \in \mathcal{Z}\mathsf{inv} \times \mathcal{Z}\mathsf{spu}$
> \begin{equation}
> \mathsf{I}(w_{\mathsf{inv}} \cdot Z_{\mathsf{inv}} + w_{\mathsf{spu}}\cdot Z_{\mathsf{spu}}) =  \mathsf{I}(w_{\mathsf{inv}}^{*} \cdot Z_{\mathsf{inv}} ),
> \end{equation}
>
>    where $w_{\mathsf{inv}}^{*}$ is the hyperplane corresponding to the true model in Assumption 2. By satisfying the above equality, every solution to ERM achieves an error value of $q$ (which is also the value that the Bayes optimal model achieves).
> Since the above equality is satisfied at all points in the support, every point is classified correctly with a probability $1-q$. At test time, the distribution over the support changes but no new points are added to the support and as a result the model error remains at $q$.
>
>    **Therefore, this result shows two important points counter to the reviewer's intuition: a) even if the classifier relies on spurious features to some extent, it can continue to be OOD optimal provided the support of the spurious features and invariant features does not change; b) the result does not require us to have multiple training environments. Given the counter-intuitive nature of the result, we hope the reviewer will find it sufficiently nontrivial.**
>
> 2. **Regarding theoretical contributions and comparisons with Nagarajan et al. and Kamath et al.** First we want to clarify that we compared with Nagarajan et al. (Theorem 5 precisely builds on the result of Nagarajan et al. and establishes how IB-IRM overcomes the difficulties highlighted in Nagarajan et al.) and Kamath et al. in the main manuscript and the supplement respectively.  We now decribe why our key insights are  important and not covered by Nagarajan et al. and Kamath et al..
>
> 	i) **On the impossibility result.** In invariant risk minimization, Arjovsky et al. had shown that for linear regression problems it is possible to learn models that generalize out-of-distribution as long as all the domains use the same labelling function to generate the outcome $Y$.  Since this is quite a strong result, it is quite natural to ask if same finding holds true in linear classification. In our impossibility result (Theorem 2), we showed that the same conditions are not enough for out-of-distribution generalization in linear classification. **Neither Nagarajan et al. nor Kamath et al. provide any such impossibility result.**
>
>       ii) **On the role of support overlap.** The reviewer says ``insufficiency results for IRM are not surprising". We would like to point out that the same classifier that relies on a spurious feature may be sufficient or insufficient depending on the support overlap assumptions. We believe this is an important and surprising result. 	**There are no formal results in Nagarajan et al. or Kamath et al. that show how support overlap assumptions play a crucial role in OOD generalization.** We show that a) if the support overlap assumption holds for both invariant and spurious features, then both ERM and IRM suffice, and surprisingly, relying on spurious features to some extent does not hurt, b) if the support overlap assumption holds only for invariant features, then ERM and IRM fail but IB-IRM succeeds, c) support overlap does not hold for invariant features, then the OOD generalization is impossible.
>
>       iii) **Further contrast with Nagarajan et al.**  While both our setting and Nagarajan et al. rely on the fully informative invariant feature  (FIIF) assumption, none of the results we mentioned above were shown in Nagarajan. In fact, **our last result (Theorem 5) even builds on one of the main findings in Nagarajan et al..** Nagarajan et al. operate under the special case of our FIIF data generation (Assumption 2 with $S$ as identity and no noise) and a special case of b) in the above bullet (they assume the distribution of invariant features does not change, not just the support). They  argue that even though in this special case there are many potential solutions of ERM, the gradient descent procedure is biased towards learning the ideal solution. However, the process of learning can be incredibly slow. In our Theorem 5, we show how IB-IRM can exponentially increase the rate of learning of gradient descent in this case.
>
>      iv) **Further contrast with Kamath et al.** **In Kamath et al., the entire analysis is carried out for PIIF case, which is fundamentally different from FIIF.** Secondly, and perhaps equally importantly the results in Kamath et al. point to different limitations of IRM than we studied. In the PIIF setting, Kamath et al. show that there can be many "ideal" looking invariant predictors, which cannot be differentiated from the risk values. In the FIIF setting, we on the other hand show that there can be many "ideal" looking invariant predictors, which cannot be differentiated from the risk values but can be differentiated from the lens of entropy (in the Appendix, we also show how some of our findings extend to the PIIF case).
>
> 	v) **OOD generalization guarantees in the FIIF case.**  **The inability of IRM to solve FIIF has been recognized as an important loophole in IRM by authors of IRM (https://www.youtube.com/watch?v=yFXPU2lMNdk&ab_channel=InstituteforAdvancedStudy between 31:20 to 32:10)**. For the non-trivial setting of FIIF where ERM and IRM fail, we were able to show how information bottleneck constraints can provably work.
>
>      To summarize, we believe our work is the first to provide impossibility results (Theorem 2), a sharp characterization of the role of support overlap (Theorem 3), and a way to overcome  some of the key failure modes, with the proposed method (Theorems 4 and 5).
> We are quite confident that the above results provide a sufficient theoretical contribution. However, if the reviewer still  feels the above arguments are not sufficiently convincing, we  would really appreciate a  detailed and substantiated counter-argument regarding the importance and novelty of our results.
>
> 3. **Why linear unit tests & colored MNIST variations?**  We would like to explain our rationale behind using linear unit tests and colored MNIST datasets.  Linear unit tests [Aubin et al.] were proposed from the same set of authors who proposed IRM post their work on domainbed (a large scale evaluation of methods on many domain generalization tasks) [Gulrajani et al.]. **The conclusion of [Aubin et al.] and many other works that also dealt with linear problems [Nagarajan et al., Kamath et al., Ahuja et al.] was that the challenge of OOD generalization is far from resolved in linear settings.** Variations of colored MNIST are natural non-linear generalizations of the linear unit tests. The community previously did not have methods that perform well on these tasks.  In light of these challenges, we strongly felt that solving linear unit tests and the two challenging variations of MNIST that capture both FIIF and PIIF settings is a solid demonstration of our theory and methods. We believe a natural next step from here is to scale our method to higher dimensions.
>
> **References**
>
> [Nagarajan et al.]  Understanding failure modes of OOD generalization.
>
> [Kamath et al.] Does IRM Capture Invariance?.

---

> > ### Author Response · Authors · 2021-08-10
> > **Response to Reviewer YQ8U (Part 2)**
> >
> > 4. **Some new results on real-world datasets.** We have conducted some experiments on more complex datasets with more complex models.  We use the COCO dataset with colored backgrounds [Lin et al., Ahmed et al.]; the dataset consists of images of objects on different backgrounds. The background colors are spuriously correlated with the label of the object.  **We use a wide resnet-based architecture and find that the OOD accuracy of ERM is at $22.70 \pm 1.04$ percent while by adding the bottleneck penalty it improves to $31.66 \pm 2.39 $  percent.** Note that due to lack of time we could not optimize hyper-parameters, which would mean that we can have further gains in this new experiment.
> >
> > 5.  **On the large variance in some of the six-environment experiments.**  Before I explain why the comparisons in the high-variance setting with six environments, we would like to clarify one point. If we focus on the three-environment case,  then that already shows that our theory matches experiments across all settings -- Example 1,2,3, their scrambled versions, and both colored MNISTS. Note that **in both the original IRM paper [Arjovsky et al.] and the recent linear unit tests paper [Aubin et al.], the main qualitative conclusions were drawn using the three-environment case (since it already presents a challenging setting)**.
> >
> >     Consider Example 1 and Example 1S from [Aubin et al.], the new environments that are sampled have labels that are drawn with a higher noise level.  Therefore, as the number of environments increases, we observe a larger error value and a larger variance.  The authors  in [Aubin et al.] compared the methods with an oracle (the ERM procedure carried out on data with no spurious correlations), which also exhibits a large variance. In [Aubin et al.]  the method that performs closest to the oracle and has similar variance as the oracle is deemed to be better than the rest. **In our findings for six environments, we follow [Aubin et al.] and use the same evaluation protocol.** In the supplement, we present the performance of the oracle to support our findings.
> >
> >
> > **We hope that we have adequately addressed your questions. We would be happy to provide further clarifications.**
> >
> > **References**
> >
> > [Arjovsky et al.] Invariant risk minimization, arxiv 2019.
> >
> > [Ahuja et al.] Empirical or invariant risk minimization? A sample complexity perspective. ICLR 2021.
> >
> > [Ahmed et al.] Systematic generalisation with group invariant predictions. ICLR 2021.
> >
> > [Lin et al.] Microsoft coco: Common objects in context. ECCV 2014.
> >
> > [Gulrajani et al.] In search of lost domain generalization.
> >
> > [Aubin et al.] Linear unit tests for invariance discovery.

---

> > > ### Author Response · Authors · 2021-08-23
> > > **Request for Response**
> > >
> > > We believe there are significant misunderstandings and mischaracterizations in your review. We would appreciate it if you can let us know if our response has changed your position. We look forward to hearing from you!

---

> ### Author Response · Authors · 2021-08-29
> **Post update: Regarding theoretical contributions + new experiment on Terra Incognita from WILDS and Domainbed**
>
> Thanks for your update. Firstly, we strongly believe that the work has novel and insightful theoretical contributions, which gives it enough merit to be accepted much like other theory papers in this area [Kamath et al., Nagarajan et al., Rosenfield et al.]. Secondly, in addition to the COCO dataset results, we have also conducted new experiments on the Terra Incognita dataset (part of both WILDS and Domainbed repository) and continue to get gains as the theory predicts (further details provided below). We hope these points would convince you to change your score.
>
> **Regarding theoretical contributions:** From your response, it seems that you are satisfied by the theoretical contributions of this work. We would also like to remind the reviewer that the work does not just prove the sufficiency of ERM and IRM. We prove other results as well. Most importantly, we prove that when both ERM and IRM fail, IB-IRM (or IB-ERM) succeeds. We succeed in settings (realizable problems) that are accepted by the authors of IRM as the main loophole in IRM.  **We hope that the reviewer realizes that these are all important results and were not remotely hinted at in [Nagarajan et al.] or [Kamath et al.].**
>
> **Results on Terra Incognita:** In addition to the COCO dataset, we ran experiments on the Terra Incognita dataset, which consists of images of animals taken from different camera traps.  We used the exact experimental setting that is described in the work [Gulrajani et al.] for Terra incognita dataset (Our code is built on their original repoistory https://github.com/facebookresearch/DomainBed).  We use Resnet-50 as was used in [Gulrajani et al.]. We provide results for two of the commonly used hyper-parameter tuning procedure from [Gulrajani et al.]
>
> 1. Train domain validation results: The accuracy of IRM is  $54.6 \pm 1.3$ percent. The accuracy of ERM is $49.8 \pm 4.4$ percent and by adding information bottleneck penalty it improves to $56.4 \pm 2.1$ percent.
>
> 2. Test domain validation results:  The accuracy of IRM is  $56.5 \pm 2.5$ percent. The accuracy of ERM is $59.4 \pm 0.9$ percent and by adding information bottleneck penalty it improves to $64.2 \pm 1.2$ percent.
>
> These results show that we get gains between $2-5$ percent in the above dataset. **We hope that the above new experiment along with the COCO dataset experiment convinces you to change the score, especially since you said you wanted to see one more dataset before you accept the work.**
>
> **Further details on COCO dataset experiment:**  We use Wide Resnet 28-10 [Zagoruyko et al.]. We append an extra group of 4 residual blocks with the same layer widths as in the previous group, and a widening factor of 4 instead of 10 to avoid memory overflow (base dimension is set to 64). We use an $\ell_2$ regularizer with coefficient of 5e−4. We train for 200 epochs with SGD + Momentum (0.9), using batch sizes of 384, with an initial learning rate of 0.1 which is cut by 10 every 30 epochs starting from epoch 120. We use an IB penalty coefficient $\gamma=1.0$.
>
> **Other points:** We would be happy to add the discussion in the paper explaining why there is higher variance in six environments.  Lastly, you raised the point about why the IB penalty worsens with more environments.  IB-ERM continues to perform well in both 3 and 6 environment cases (close to 2 percent error). It goes from $0$ percent to $2$ percent error and this is perhaps due to the new three environments having data points that have stronger shortcut signals.
>
>
> **References**
>
> [Zagoruyko et al.] Wide residual networks. arXiv preprint arXiv:1605.07146.
>
> [Gulrajani et al.] In search of lost domain generalization. ICLR 2021.
>
> [Kamath et al.] Does IRM capture invariance? AISTATS 2021.
>
> [Rosenfield et al.] Risks of Invariant Risk Minimization. ICLR 2021.
>
> [Nagarajan et al.] Understanding failure modes of out-of-distribution generalization. ICLR 2021.

---

> > ### Comment · Reviewer_YQ8U · 2021-09-05
> > **thanks**
> >
> > Thanks for the clarifications and new experiments. The Terra incognita experiments are nice, thanks. Due to the authors efforts I'm happy to raise the score to 5 and would also be ok if the paper were accepted. I do understand sufficiency was not the main theoretical result, but I still think the methodology is the main contribution and the theory does not give a lot of new insights given previous work and hence the emphasis in the writing part from my perspective should be a bit more on the experimental part.
> >
> > Regarding the 3 to 6 getting worse hypothesis, would be good if you could verify the hypothesis with some experiments.

---

> > > ### Author Response · Authors · 2021-09-05
> > > **Thank you!**
> > >
> > > Thank you for appreciating our efforts and raising your score. We will incorporate the suggested changes in the manuscript.

---

> > ### Comment · Reviewer_bprw · 2021-09-05
> > **A quick question on the results on Terra Incognita**
> >
> > As you used the exactly same experimental settings described in the work [Gulrajani et al.] for Terra incognita dataset, I am just wondering why the results of IRM and ERM in your experiments are obviously different (huge gap) from those reported in their work? See the detailed comparison below.
> >
> > + Train domain validation results
> >    +  **in their work**: The accuracy of IRM is $47.9 \pm 0.7$, and the accuracy of ERM is $47.2 \pm 0.4$.
> >    +  **in your work**: The accuracy of IRM is $54.6 \pm 1.3$, and the accuracy of ERM is $49.8 \pm 4.4$.
> >
> > + Test domain validation results
> >    +  **in their work**: The accuracy of IRM is $50.9 \pm 0.4$, and the accuracy of ERM is $52.7 \pm 0.2$.
> >    +  **in your work**: The accuracy of IRM is $56.5 \pm 2.5$, and the accuracy of ERM is $59.4 \pm 0.9$.

---

> > > ### Author Response · Authors · 2021-09-05
> > > **Clarification**
> > >
> > > Hi
> > >
> > >
> > > Thank you for your question. There are four environments in Terra incognita, which are called L100, L38, L43, L46. Three environments can be used for training and fourth is used for testing. In the experiments, we conducted we used L100 as test environment and the remaining environments for training. In the paper, https://arxiv.org/pdf/2007.01434.pdf, on page 22 Table B6 (first column), the authors report the performance for this exact setting.  There is not much difference in the results. Please see below.
> > >
> > > a) Train domain validation results:
> > >
> > >    in their work: The accuracy of IRM is $52.2 \pm 3.1$, and the accuracy of ERM is $50.8\pm 1.8$.
> > >
> > >    in our work: The accuracy of IRM is $ 54.6\pm 1.3$, and the accuracy of ERM is $49.8 \pm 4.4$.
> > >
> > > b) Test domain validation results
> > >
> > >   in their work: The accuracy of IRM is $56.8 \pm 2.0$, and the accuracy of ERM is $59.9 \pm 1.0$.
> > >
> > >   in our work: The accuracy of IRM is $56.5 \pm 2.5$, and the accuracy of ERM is $59.4 \pm 0.9$.
> > >
> > > The numbers that you stated are averaged performance over all possible test environment, train environment combination. We hope this answers your question.

---

> ### Author Response · Authors · 2021-08-31
> **Request for a response since the discussion phase ends soon.**
>
> In the updated review you said, "I would like to see more experimental details before giving the nod + experiments on one more dataset."
> We added new experiments on the Terra Incognita dataset (part of WILDS and Domainbed; see details below), which continue to show that our theory aligns with practice. The discussion phase ends soon. We would be grateful if you can update your review in light of our response below (https://openreview.net/forum?id=jlchsFOLfeF&noteId=hveQAx6A4Yy).

---

### Official Review · Reviewer_s4BJ · 2021-07-17

**Rating:** 6
**Confidence:** 3

**Summary:**

This paper investigates the conditions that are needed to provably generalize OOD in the linear classification tasks. Comparing with the linear regression tasks, linear classification tasks need more strong conditions to guarantee the OOD generalization.

The paper first proposes the overlap assumptions on both invariant and spurious feature and analyze the OOD generalization guarantee under these assumptions. The theorem shows that only both two types of features satisfy the support overlap assumption can we expect the IRM to solve the OOD generalization problems. Then the paper study the possibility of solving the problem only with the support overlap assumption on the invariant features. The proposed method uses both information bottleneck and invariance constraints. Theoretical results show that the proposed IB-IRM can successfully solve the problem in both fully and partially informative invariant features situations. The experiment results are consistent with the theoretical results.


**Limitations And Societal Impact:**

See the detailed questions in the main review, especially question 3 and 4.

**Main Review:**

This paper is well organized and easy to follow. The theoretical results are presented in a logical order and the conclusions are quite clear. Although all the proofs are in the appendix, authors provide the intuitions by examples and discussed the significance of the theorems.
With these theorems, the proposed IB-ERM is well-motivated. Some relaxations are involved from the original formulation in (6) to the final objective in (7). The influence of the IB term to optimization is discussed in Theorem 5.

Overall, this paper is well-written, with comprehensive theoretical analysis to support the proposed IB-ERM for linear classification.
Detailed comments/questions:
1.	Although we can construct tasks like CS-CMNIST and the model in assumption 2, is it true that, with high-probability, the oracle in real world is PIIF? Since the conditional independency is a special case of the conditional dependency. Then, what is the advantage of IB-IRM, comparing to other methods designed for PIIF, e.g., (Rosenfeld et al. 2021)?
2.	The proposed IB-IRM in eqn (6) takes IB as the objective and ERM and invariance as constraints. Will theorem 4 still holds if one take IB as an additional constraint in IRM? Can we have more insights to design the relaxed objective later?
3.	How did you tune the coefficient of the two penalty terms at the equation (7) in real experiments? Any ablation study to show the influence of these two penalties in a different situation?
4.	In sec 5, authors make discussion about the results on CMNIST tasks. How about the experimental results on linear unit tests? Why only make comparison on three and six training environments? Do we have any prior knowledge about which tasks are PIIF and which tasks are FIIP? IB-IRM is not the best across all the tasks.
5.	Minors: 1) put the DAG into Figure 2? 2) ref the definition of “support overlap invariant/spurious features” in table 1; 3) Is the observation in figure 3 general for different values of \lambda and \gamma? 4) In Theorem 3, “some of the ERM and IRM solution..” is not rigorous. “these exist”?


**Time Spent Reviewing:**

6

---

> ### Author Response · Authors · 2021-08-10
> **Response to reviewer s4BJ**
>
> Thank you for your comments! Below we provide responses to the questions.
>
> 1. **On FIIF vs PIIF in the real world:** In a talk on invariant risk minimization (https://www.youtube.com/watch?v=yFXPU2lMNdk&ab_channel=InstituteforAdvancedStudy from 29:50), Léon Bottou refers to most human-generated labels as realizable, which is the same as FIIF in this context, and nature-generated examples as non-realizable, which is same as PIIF in this context. Below we describe different real examples of FIIF and PIIF and for a further list refer to [Arjovsky].
>
>     *FIIF in the real world:* FIIF is supposed to capture phenomena where invariant features capture all or almost all the information about the label. The original IRM paper [Arjovsky et al.] also described image classification as FIIF (they refer to it as realizable problems). There is significant evidence that IRM does not beat ERM in FIIF problems [Aubin et al., Ahuja et al.]. Consider the case of digit classification in colored MNIST. In colored MNIST, the uncolored digit is used by the human to generate the labels and the color is present due to selection bias but does not provide any extra information about the digit label. The same phenomenon is true for images of cows on green pastures.
>
>     *PIIF in the real world:* PIIF is supposed to capture phenomena where invariant features capture some information about the label and a significant portion of the information is also carried by spurious features in the training environments. Consider the case when humans are given blurry images of cows to label. In such a case, the human may rely on the background color to guess the animal in the image. For instance, if the image is green and the animal is blurry, the human would most likely guess it is a cow rather than a camel.
>
> 2.  **What makes FIIF fundamentally different from PIIF?** **IRM can solve PIIF but it's inability to solve FIIF has been recognized as an important loophole in IRM by authors of IRM (https://www.youtube.com/watch?v=yFXPU2lMNdk&ab_channel=InstituteforAdvancedStudy between 31:15 to 32:10)**.
> In the PIIF settings, since the spurious features carry extra information about the label not contained in the invariant features, any ERM-based framework aimed at reducing the error would rely on the spurious features. The extent to which spurious features impact the error varies across environments. This very fact is exploited by IRM-based frameworks, which allows it to perform well in PIIF based settings. However, when it comes to FIIF settings, the spurious features do not provide extra information and thus cannot improve the error any further than what can be achieved by the invariant features. Therefore, in FIIF settings the signature of spuriousness is harder to detect. For further discussion on FIIF vs PIIF refer to [Nagarajan et al.][Ahuja et al.]. To summarize, we believe both FIIF and PIIF settings can occur in real world and impact the methods very differently, which is why we believe both invariance and information bottleneck constraints play an important role (as demonstrated in our theory and experiments).
>
> 3. **On the alternate constrained formulation with IB as constraint instead of objective and insights on the objective in equation (7):**. The results should hold in the alternate formulation and we can add a discussion on that in the paper. We provided the derivation of the objective in equation (7) in the supplement. We would be happy to move the key insights to the main text. Our derivation follows two key steps a) we approximate invariance constraint using the invariance penalty from [Arjovsky et al.], and b) we approximate differential entropy with its upper bound from [Kirsch et al.].
>
> 4. **On the details of tuning of penalties and ablation:** The description of the tuning procedure is in the supplement. We would be happy to move some of the key steps in the main text. There are two common procedures for tuning hyper-parameters that are used in the OOD generalization literature [Gulrajani et al.] a) train-domain validation procedure -- in this procedure we construct a validation set by mixing data from the training environments , b) oracle validation procedure -- in this procedure, we assume access to limited samples from the test distribution (not the test samples themselves)  and that we are allowed to do limited queries on this data. We carried out our CMNIST experiments with both these procedures. We present all the detailed results in the supplement.
>
>    We can provide a detailed ablation study on the impact of penalties in each setting. Also, we would like to highlight how our current experiments and theory already explain the role of penalties to a reasonable degree. We divided the datasets into two parts -- a) FIIF: fully informative invariant features, and b) PIIF: partially informative invariant features. Our theory (Theorem 4) tells us that in a) FIIF settings the information bottleneck constraints play a more important role than the invariance constraints, and in b) PIIF settings the invariance constraints play a more important role than the information bottleneck constraints. In our experiments, we observed a consistent behavior, i.e., the invariance penalty was much more crucial in Example 1, 3 (1S, 3S), and AC-CMNIST, which are all PIIF settings.   On the other hand, the information bottleneck penalty was more crucial in Example 2, 2S, and CS-CMNIST.
>
> 5. **Questions on experiments.**
>
>     a) **Discussion on linear unit tests:** We did discuss results for both CMNIST and linear unit test results under the two dataset categories. AC-CMNIST, Example 1/1S, Example 3/3S fall in the PIIF category. CS-CMNIST, Example 2/2S fall in the FIIF category.
>
>    b) **Why three and six environments?** The experimental setting in the linear unit tests that were proposed by [Aubin et al.] is such that as we add more environments, the error achieved by the models increases and so does the variance for a fixed number of experiments. Therefore, to achieve stable results with more environments we would need many more experiments, which is the reason we did not add more environments. If you see the results in [Aubin et al]., the main qualitative conclusions come from the three-environment setting. In their experiments with larger number of environments, we can see that the variance is quite large, thus pointing to the need for many more experiments.
>
>    c) **Prior knowledge of FIIF vs PIIF:** Following the discussion in Léon Bottou's talk (link above) , we believe many tasks with human-generated labels correspond to the FIIF category. However, tasks that involve non-human or nature-generated  labels are PIIF. Developing methods that can actually help discern what task falls in which category is a very interesting area for future work.
>
> 6. **On the minor comments:** We would add a reference to the definition of support overlap. The observation is quite general, and yes, we can add a plot with a range of values (in the supplement).  Thanks for pointing it out, we will change it to "there exists" in the statement of the theorem.
>
>
>
> **References**
>
> Aubin, B., Słowik, A., Arjovsky, M., Bottou, L., & Lopez-Paz, D. (2021). Linear unit-tests for invariance discovery. arXiv preprint arXiv:2102.10867.
>
> Arjovsky, M., Bottou, L., Gulrajani, I., & Lopez-Paz, D. (2019). Invariant risk minimization. arXiv preprint arXiv:1907.02893.
>
> Ahuja, K., Wang, J., Dhurandhar, A., Shanmugam, K., & Varshney, K. R. (2020). Empirical or invariant risk minimization? a sample complexity perspective. arXiv preprint arXiv:2010.16412.
>
> Nagarajan, V., Andreassen, A., & Neyshabur, B. (2020). Understanding the failure modes of out-of-distribution generalization. arXiv preprint arXiv:2010.15775.
>
> Kirsch, A., Lyle, C., & Gal, Y. (2020). Unpacking information bottlenecks: Unifying information-theoretic objectives in deep learning. arXiv preprint arXiv:2003.12537.

---

> > ### Comment · Reviewer_s4BJ · 2021-08-24
> > **Thanks for the response**
> >
> > Thanks for the clarifications. I suggest the authors modify the manuscript in the next version accordingly. Some discussion or experiments on non-linear case will enhance the impact of this work.

---

> > > ### Author Response · Authors · 2021-08-27
> > > **Thank you for the response**
> > >
> > > We would like to thank the reviewer for reading our response. We will incorporate the suggested changes in the manuscript.

---

### Official Review · Reviewer_kbVa · 2021-07-18

**Rating:** 6
**Confidence:** 4

**Summary:**

This work gives an indepth analysis to the sufficient conditions on the generalization of invariant risk minimization (IRM) in classification setting. Authors show that in order to guarantee good generalization, the support overlap condition of invariant features is required. Further, authors also show and validated that if the support overlap condition does not satisfied for spurious features, the combination of IRM objective and information bottleneck regularization (i.e., minimzing the mutual information $I(X; \Phi(X))$) is a reliable remedy.

**Limitations And Societal Impact:**

Authors clearly mentioned limitations and future work directions.

**Main Review:**

In general, this work made a few theoretical contributions. It specifies some conditions to guarantee good generalization of IRM in classification, especially the support overlap of invariant (or spurious) features. Some theorems (like Theorem 4) also partially explains why information bottleneck can improve generalization, which is less investigated before.

However, I still have some concerns:
1. Authors combine the objective of IRM and the minimizing of entropy of representation $H(\Phi(X))$. However, instead of directly optimizing the differential entropy (which is usually hard), authors made a Gaussian assumption and alternatively minimize the variance. I am not so sure how practical is this assumption in real-world data or applications. Because the experiments in this work is just on simple synthetic data or MNIST-like data.

Can you provide more justifications on this approximation or more empricial results on recent domain generalization benchmarks (i.e., the DomainBed [1] or WILDS [2]) in which both IRM and ERM are baselines.

[1] Gulrajani, Ishaan, and David Lopez-Paz. "In search of lost domain generalization." arXiv preprint arXiv:2007.01434 (2020).

[2] Koh, Pang Wei, et al. "Wilds: A benchmark of in-the-wild distribution shifts." International Conference on Machine Learning. PMLR, 2021.

2. Similar to the information bottleneck, I feel the hyperparameter $\gamma$ (in Eq. (7)) controls a trade-off and plays a significant role to the performance. Here, there is another hyperprameter $\lambda$ that controls the degree of invariance. Can you give a more indepth analysis on the trade-offs between these terms or how different values of $\gamma$ or $\lambda$ influence the performance of the combined objective?

3. I still feel there is a jump from your analysis in Section 3 to the introduction of information bottleneck regularization (i.e., $\min H(\Phi(X))$) in Section 4. Just focusing on your illustrative example, when $\Phi$ only picks invariant features, the entropy is minimum. However, it does not mean when the entropy is minimized, $\Phi$ exactly picks the invariant features. For example, the suprious features only also gives the minimum entropy. My question is how can you guarantee this regularization alone is sufficient for IB-IRM to only focus on invariant features, rather than other combinations of invariant and suprious features? or did I miss something?

Finally, some minor suggestions:
1. It would be much better if authors can give some illustrative examples on "fully" informative features and "partially" informative features.

**Time Spent Reviewing:**

6

---

> ### Author Response · Authors · 2021-08-10
> **Response to reviewer kbVa**
>
> Thank you for your comments! Below we provide responses to the questions.
>
> 1. **On the approximation of differential entropy using upper bounds:** Our approximation of differential entropy is based on the works [Kirsch et al.] (also cited in the paper). While the approximation using Gaussians do seem simple, it provides the same effects as the true information bottleneck constraints. See the results in [Kirsch et al.] on ImageNette (a subset of ImageNet) and CIFAR-10 with Resnet-18 architecture.  Also, note that in our experiments on colored MNIST  using ConvNets the dimension of the layer at which bottleneck constraints are applied is reasonably large (256 dimensions), in case the scaling of our approximation to higher dimensions was a concern. Also, following the reviewer's advice, we have conducted some additional experiments on a more complex dataset with a more complex model. Specifically, we use the COCO dataset with colored backgrounds [Lin et al., Ahmed et al.]; the dataset consists of images of objects on different backgrounds. The background colors are spuriously correlated with the label of the object. **We use a wide resnet-based architecture and find that OOD accuracy of ERM is at $22.70 \pm 1.04$
> percent and that by adding the bottleneck penalty it improves to $31.66 \pm 2.39 $ percent.** Note that due to lack of time we could not optimize hyper-parameters, which would mean that we can have further gains in this new experiment.
>
> 2. **On the role of $\gamma$ (information bottleneck penalty) and $\lambda$ (invariance penalty) and their tradeoffs:** We would be happy to add a detailed experiment on the above. However, we would like to explain that our current experiments and theory already provide insights into the qualitative nature of the results. We divided the datasets into two parts:  a) FIIF: fully informative invariant features, and b) PIIF: partially informative invariant features. Our theory (Theorem 4) tells us that in a) FIIF settings information bottleneck constraints play a more important role than the invariance constraints, and in b) PIIF settings invariance constraints play a more important role than the information bottleneck constraints. In our experiments, we observed a consistent behavior, i.e., the invariance penalty was much more crucial in Example 1, 3 (1S, 3S), and AC-CMNIST, which are all PIIF settings.   On the other hand, the information bottleneck penalty was more crucial in Example 2, 2S, and CS-CMNIST.
>
> 3. **On the confusion about why does information bottleneck regularization suffice:** We use three metrics --  performance of the model, the entropy (information bottleneck enforcer), and the invariance constraints to select a model. Let us focus on performance of the model and the entropy as these two suffice for this question. In equation (6), **we select the minimum entropy model among the models that have an error rate below a threshold $r^{\mathsf{th}}$**. Suppose we selected a model that relies on spurious features only: such a model would not meet the error threshold (as long as the constraints are set strict enough). To further the point, in Theorem 4 we showed that the optimization in equation (6) would result in models that only focus on invariant features provided the constraints on performance are sufficiently strict.
>
> 4. **On FIIF vs PIIF in real world.** In a talk on IRM (https://www.youtube.com/watch?v=yFXPU2lMNdk&ab_channel=InstituteforAdvancedStudy from 29:50), Léon Bottou refers to most human-generated labels as realizable, which is the same as FIIF in this context,  and nature-generated examples as non-realizable, which is same as PIIF in this context.  Below we describe different real examples of FIIF and PIIF and for a further list refer to [Arjovsky].
>
>     *FIIF in the real world:* FIIF is supposed to capture phenomena where invariant features capture all or almost all the information about the label. The original IRM paper [Arjovsky et al.] also described image classification as FIIF (they refer to it as realizable problems). There is significant evidence that IRM does not beat ERM in FIIF problems [Aubin et al., Ahuja et al.]. In the same talk  that we mentioned above, **Léon Bottou describes how IRM can solve PIIF but it's inability to solve FIIF is an important loophole in IRM (this is one of the key assues our work addresses)**.  Consider the case of digit classification in colored MNIST. In colored MNIST,  the uncolored digit is used by the human to generate the labels and the color is present due to selection bias but does not provide any extra information about the digit label. The same phenomenon is true for images of cows on green pastures.
>
>    *PIIF in the real world:* PIIF is supposed to capture phenomena where invariant features capture some information about the label and a significant portion of the information is also carried by spurious features in the training environments.  Consider the case when humans are given blurry images of cows to label. In such a case, the human may rely on the background color to guess the animal in the image. For instance, if the image is green and the animal is blurry, the human would most likely guess it is a cow rather than a camel.
>
>
> **We hope that we have adequately addressed your questions. We would be happy to provide further clarifications.**
>
> **References**
>
> Kirsch, A., Lyle, C., & Gal, Y. (2020). Unpacking information bottlenecks: Unifying information-theoretic objectives in deep learning. arXiv preprint arXiv:2003.12537.
>
> Ahmed, F., Bengio, Y., van Seijen, H., & Courville, A. (2020, September). Systematic generalisation with group invariant predictions. In International Conference on Learning Representations.
>
> Lin, T. Y., Maire, M., Belongie, S., Hays, J., Perona, P., Ramanan, D., ... & Zitnick, C. L. (2014, September). Microsoft coco: Common objects in context. In European conference on computer vision (pp. 740-755). Springer, Cham.
>
> Arjovsky, M., Bottou, L., Gulrajani, I., & Lopez-Paz, D. (2019). Invariant risk minimization. arXiv preprint arXiv:1907.02893.
>
> Arjovsky, M. (2020). Out of distribution generalization in machine learning (Doctoral dissertation, New York University).
>
> Ahuja, K., Wang, J., Dhurandhar, A., Shanmugam, K., & Varshney, K. R. (2020). Empirical or invariant risk minimization? a sample complexity perspective. arXiv preprint arXiv:2010.1
>
> Aubin, B., Słowik, A., Arjovsky, M., Bottou, L., & Lopez-Paz, D. (2021). Linear unit-tests for invariance discovery. arXiv preprint arXiv:2102.10867.

---

> > ### Comment · Reviewer_kbVa · 2021-08-27
> > **Thanks for your reply**
> >
> > Thanks for your reply. My concern is largely addressed. I would like to increase my score to 6.
> > However, I recommend authors to carefully revise the current manuscript, add new results and experimental details on COCO. It would be also better if authors can discuss some issues on Gaussian approximation in the revised manuscript.

---

> > > ### Author Response · Authors · 2021-08-27
> > > **Thank you for the response**
> > >
> > > We would like to thank the reviewer for reading our response and raising the score. We are happy to hear that we have largely addressed your concerns. We will surely incorporate the suggested changes -- adding COCO experiment details and results, adding discussion pertaining to the Gaussian approximation.

---

### Official Review · Reviewer_bprw · 2021-07-20

**Rating:** 7
**Confidence:** 4

**Summary:**

In this paper, the authors re-check the invariance principle and show that out-of-distribution generalization is much harder for linear classification tasks, requiring much stronger restrictions in the form of support overlap assumptions on the distribution shifts. They propose an approach combining both information bottleneck constraints and the invariance principle to addressing the OOD generalization in both regression and classification tasks.

**Limitations And Societal Impact:**

Yes.

**Main Review:**

I pretty like the theoretical analysis presented in this paper, from which we could get very insightful thoughts.

In this paper, the authors use 0-1 loss for binary classification $\mathcal{Y}=\{0, 1\}$ and square loss for regression $\mathcal{Y}=\mathbb{R}$. I am curious if all the results in this paper can be extended to the other losses and to multi-class classification tasks.

Although the authors admit in the final section that the proposed method mainly focuses on linear models. I would like to hear more from the authors about what the main challenge is when their theory is extended to nonlinear models.

**Time Spent Reviewing:**

two hours

---

> ### Author Response · Authors · 2021-08-10
> **Response regarding extensions**
>
> Thank you for your comments! Below we provide responses to the questions.
>
> 1. **Extension to multi-class classification tasks:** We used binary classification tasks for the clarity of exposition. The work generalizes to multi-class classification tasks,  and we will add a discussion on it in the paper.  Consider a natural extension of binary classification to a multi-class classification. In a standard binary setting, we classify points as positive or negative based on which side of a hyperplane the points are located at. In the multi-class setting with $k$ classes, we consider one hyperplane per class.
> The classifier is parametrized by a matrix $W_{\mathsf{inv}}\in \mathbb{R}^{k \times d}$, where each row corresponds to the hyperplane of the corresponding classs. The generative model in Assumption 2 is extended to multi-class case as follows
> \begin{equation}
> \begin{split}
>     & Y^e \leftarrow \arg\max(W_{\mathsf{inv}}\cdot Z_{\mathsf{inv}}^{e}), \\
>     & X^e  \leftarrow S(Z_{\mathsf{inv}}^{e}, Z_{\mathsf{spu}}^{e})
> \end{split}
> \end{equation}
> where $X^e$ represents the observed data, $Z_{\mathsf{inv}}^{e}$ is the latent invariant  feature vector, $Z_{\mathsf{spu}}^{e}$ is the latent spurious feature vector, $Y^{e}$ is the label generated from the $\arg\max$ operation over the rows of the matrix $W_{\mathsf{inv}}$. In the above model, we can also make the label generation noisy, i.e., add a noise term that switches the class to another class, uniformly at random, with some probability.  In order to extend our results to the above multi-class model, we need to extend Lemma 7 in the supplement from scalar random variables to vectors.  However, the result can be extended to  random vectors by following the exact same proof steps, which leads to generalizing our claims  to  the multi-class classification problems.
>
> 2. **Extension to the non-linear case:** In Assumption 1, we write the observations in terms of the latents, as  $X=S(Z)$, where $S$ is a linear function. Suppose  $S$ is a piecewise linear function. We can then divide the learning into two parts: a) learning a model that tries to find the true representation $Z$ (up to permutation and scaling errors) and  b) using this representation to learn a linear model for classification. Suppose we can solve part a). In part b), the key challenge is   to ensure that we learn a model that does not rely on the latent spurious  features. Following the footsteps of the first part of Theorem 4, we can resolve the challenge posed by part b) (provided the noise level in $W^{e}$ in Theorem 4 is not too large). The main challenge then is to solve part a). We propose a naive approach based on the works [Khemakhem, et al.][Lu et al.]. We can use techniques from non-linear ICA that guarantee that we can learn the representation $Z$ upto permutation and scaling. However, the existing techniques assume the latents are independent or conditionally independent given some side information. In our setting, we may treat the environment index as such side information. Assuming the environment provides conditional independence does allow us to extend the results, but this is quite a strong assumption. We are currently in the process of working on relaxations that do not make such assumptions and that we believe is a fruitful, and interesting future work.
>
> **References**
>
> Khemakhem, I., Kingma, D., Monti, R., & Hyvarinen, A. (2020, June). Variational autoencoders and nonlinear ica: A unifying framework. In International Conference on Artificial Intelligence and Statistics (pp. 2207-2217). PMLR.
>
> Lu, C., Wu, Y., Hernández-Lobato, J. M., & Schölkopf, B. (2021). Nonlinear invariant risk minimization: A causal approach. arXiv preprint arXiv:2102.12353.

---

### Decision · Program_Chairs · 2021-09-27

**Decision:**

Accept (Spotlight)

**Comment:**

After substantial discussion, including additions of new experimental results from the authors, reviewers have come to a consensus that this paper provides interesting and relevant theoretical results, as well as sufficient empirical evidence supporting the theory, to merit acceptance. We expect that this work will appeal to the ERM, IRM, and Information Bottleneck communities. We particularly think that the exponential speedup of convergence of the IB-based methods will be of theoretical and practical interest. We look forward to reading the updated version of the paper with the additions and revisions the authors have promised.